# Training-Free Bayesianization for Low-Rank Adapters of Large Language Models

**Haizhou Shi**[*1]    **Yibin Wang** [*2]    **Ligong Han** [3]    **Huan Zhang** [2]    **Hao Wang** [†1]

## Abstract

Estimating the uncertainty of responses from Large Language Models (LLMs) remains a critical challenge. While recent Bayesian methods have demonstrated effectiveness in quantifying uncertainty through low-rank weight updates, they typically require complex fine-tuning or post-training procedures. In this paper, we propose **T**raining-**F**ree **B**ayesianization (TFB), a simple yet theoretically grounded framework that efficiently transforms trained low-rank adapters into Bayesian ones without additional training. TFB systematically searches for the maximally acceptable level of variance in the weight posterior, constrained within a family of low-rank isotropic Gaussian distributions. Our theoretical analysis shows that under mild conditions, this search process is equivalent to KL-regularized variational optimization, a generalized form of variational inference. Through comprehensive experiments, we show that TFB achieves superior uncertainty estimation and generalization compared to existing methods while eliminating the need for complex Bayesianization training procedures. Code is available at `https://github.com/Wang-ML-Lab/bayesian-peft`.

## 1   Introduction

Despite recent advances in Large Language Models (LLMs) showing great capacity for generating responsive answers to human instructions [5, 64, 63, 44, 9, 3, 54, 55, 49, 8, 2, 48], the reliability of such large models remains a critical concern [59, 58], as untruthful yet confident answers could cause significant damage to individuals and society [17, 47, 66, 28]. The accurate estimation of uncertainty in LLMs has thus emerged as an urgent challenge. Current approaches mainly follow two paths: one focuses on directly asking the model to elicit its internal internal (verbalized) uncertainty [65, 52, 28], while the other employs complex fine-tuning techniques [28, 67, 62].

Both approaches suffer from inherent limitations. Verbalized uncertainty, while simple to implement, remains controversial in terms of its empirical reliability and ***theoretical soundness*** [27, 38]. On the other hand, low-rank adapters (LoRA [23]), which offer a parameter-efficient way to adapt LLMs by adding a small set of low-rank weight matrices, have emerged as a promising direction for fine-tuning models. However, while LoRA efficiently adapts large models to new tasks, it does not itself provide a mechanism for principled uncertainty estimation. In response, recent Bayesianization attempts [67, 62], integrate Bayesian methods with LoRA, but they still require complex training procedures and sophisticated hyperparameter tuning, ***limiting their practicality***. These constraints motivate the following research question:

*Can we "Bayesianize" LLM low-rank adapters in a **theoretically sound** yet **empirically simple** way?*

In this paper, we diverge from conventional fine-tuning and post-training approaches. Instead, we develop a Training-Free Bayesianization (TFB) technique applicable to *any* given low-rank LLM adapter. TFB constrains the family of full-weight approximate posteriors produced by LoRA adapters to low-rank isotropic Gaussian distributions. Given a trained LoRA adapter, it systematically searches for the maximally acceptable variance of the variational distribution of the weight posterior, without

---

[*]Equal Contribution. [1]Rutgers University. [2]University of Illinois Urbana-Champaign (UIUC). [3]Red Hat AI Innovation. [†]Correspondence to: Haizhou Shi <haizhou.shi@rutgers.edu>, Hao Wang <hw488@cs.rutgers.edu>.

39th Conference on Neural Information Processing Systems (NeurIPS 2025).

the need for complex fine-tuning procedures. TFB's search range and stopping criteria can be determined using *any* in-distribution "anchor dataset," e.g., a small subset of the training dataset. Note that (1) this eliminates the need for an additional calibration or validation dataset; (2) this flexibility extends to both supervised and unsupervised data, even regardless of whether it was used in the original LoRA training.

Despite its simplicity, we theoretically demonstrate that, TFB's process of finding the maximal variance of the low-rank isotropic Gaussian posterior is equivalent to generalized variational inference, under mild conditions.

We verify TFB's effectiveness through extensive empirical evaluation across various settings, datasets, LLM backbones, LoRA weights, and LoRA variants. Our comprehensive experiments demonstrate that this novel training-free Bayesianization framework consistently achieves superior generalization and more accurate uncertainty estimation. To summarize, the main contributions of this paper are:

- We propose Training-Free Bayesianization (TFB), the first framework to transform trained LoRAs into Bayesian ones without re-training, continued training, or gradient estimation.
- We establish theoretical connections between TFB and generalized variational inference, proving their equivalence under mild conditions.
- We develop an efficient implementation of TFB requiring only an anchor dataset for search, making it widely applicable across different application scenarios.
- Through comprehensive experiments, we demonstrate that TFB consistently improves uncertainty estimation for off-the-shelf LoRA adapters, and overall surpasses the state-of-the-art counterparts of Bayesian LoRA.

## 2  Related Work

**LLM Uncertainty Estimation.** To estimate the uncertainty of LLMs, the models are often employed to generate and evaluate their own uncertainty [40, 27]. However, such approaches typically rely on task-specific labels and require additional training. Semantic entropy [38] leverages the invariance of language stemming from shared meanings to estimate uncertainty, while mutual information is used to compute a lower bound on model uncertainty by sampling from the model's output distribution [66]. Despite their contributions, these methods fail to accurately capture true model uncertainty, as they do not model the probability distribution over the LLM parameters [25, 1, 15].

**Bayesian Low-Rank Adaptation.** The Bayesian framework provides a powerful approach for capturing and estimating uncertainty during fine-tuning by defining prior distributions and approximating posterior distributions over the model parameters [46, 21, 14, 56]. Recent research has explored combining Bayesian methods with LoRA to mitigate the additional computational overhead associated with modeling parameter distributions across the entire parameter space. Yang et al. [67] applies a Kronecker-factorized Laplace approximation to fine-tuned LoRA parameters. More recently, BLoB [62] advances the field by simultaneously estimating both the mean and covariance of LLM parameters within a single fine-tuning stage. Our proposed training-free Bayesianization represents a significant departure from these existing methods. Unlike approaches that require re-training [14, 61, 4, 62] or rely on continued training and gradient estimation [67], our method achieves uncertainty estimation without any additional training steps, substantially improving the simplicity and efficiency for Bayesian learning of LLMs.

## 3  Training-Free Bayesianization (TFB)

This section introduces our Training-Free Bayesianization (TFB). Sec. 3.1 introduces the problem setup. Sec. 3.2 and Sec. 3.3 present the two key parts of TFB: low-rank Gaussian variational distribution family and a novel approach for converting deterministic weights to probabilistic distributions without training. The complete algorithmic implementation is provided in Sec. 3.4, with theoretical foundations addressed in a separate section (Sec. 4).

**Notation.** Scalars, vectors, and matrices are denoted by lowercase letters, lowercase boldface letters, and uppercase boldface letters, respectively. For a matrix $\boldsymbol{X} = [\boldsymbol{x}_1, \cdots, \boldsymbol{x}_n] \in \mathbb{R}^{m \times n}$, we use $\text{vec}(\boldsymbol{X}) = [\boldsymbol{x}_1^\top, \boldsymbol{x}_2^\top, \cdots, \boldsymbol{x}_n^\top]^\top \in \mathbb{R}^{(mn) \times 1}$ to denote vectorization. $\otimes$ and $\circ$ denote the Kronecker and element-wise product, respectively. We use $\boldsymbol{0}_n \in \mathbb{R}^{n \times n}$ to denote a zero matrix.

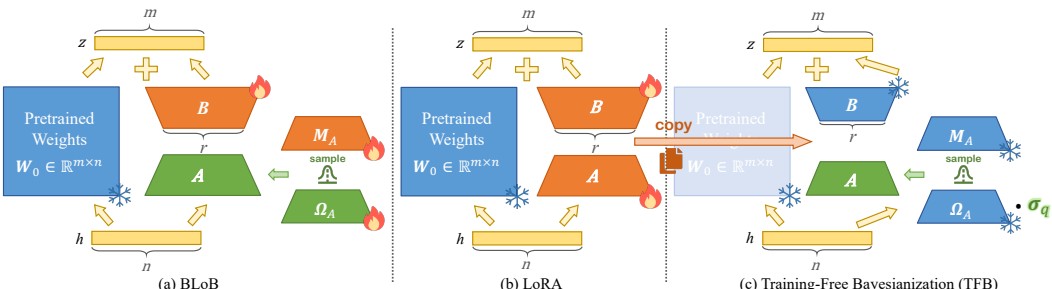

Figure 1: Overview of **T**raining-**F**ree **B**ayesianization (TFB, Ours, **right**), as well as comparison with existing methods such as LoRA **(middle)** and BLoB **(left)**.

## 3.1 Preliminaries

**Low-Rank Adaptation (LoRA).** Given a pre-trained neural network layer with weight matrix $\boldsymbol{W}_0$, Low-Rank Adaptation (LoRA) [23] confines weight updates to a low-rank subspace during fine-tuning, expressing the update as $\Delta \boldsymbol{W} = \boldsymbol{B}\boldsymbol{A}$, where $\Delta \boldsymbol{W} \in \mathbb{R}^{m \times n}$, $\boldsymbol{B} \in \mathbb{R}^{m \times r}$, and $\boldsymbol{A} \in \mathbb{R}^{r \times n}$. For input $\boldsymbol{h}$ and output $\boldsymbol{z}$ of the LoRA layer, the forward pass computation is then given by:

$$\boldsymbol{z} = \boldsymbol{W}_0\boldsymbol{h} + \Delta \boldsymbol{W}\boldsymbol{h} = \boldsymbol{W}_0\boldsymbol{h} + \boldsymbol{B}\boldsymbol{A}\boldsymbol{h}. \tag{1}$$

**LoRA Bayesianization with Low-Rank Gaussian Distribution.** BLoB [62], a pioneering work in low-rank Bayesianization for LLMs, empirically demonstrates that modeling $\boldsymbol{A}$'s elements with independent Gaussian variables suffices for effective uncertainty estimation in LoRA. Specifically, the probability density of each element of $\boldsymbol{A}$ follows $q(A_{ij}) = \mathcal{N}(A_{ij}|M_{ij}, \Omega_{ij}^2), \forall i \in [r], \forall j \in [n]$, where matrices $\boldsymbol{M}$ and $\boldsymbol{\Omega}$, sharing the dimensions of $\boldsymbol{A}$, represent the mean and standard deviation of the random variable $\boldsymbol{A}$, respectively. This formulation is equivalent to approximating the Bayesianized low-rank adapter's posterior in the full-weight space of $\boldsymbol{W}$ with a low-rank degenerate distribution:

$$q(\text{vec}(\boldsymbol{W})|\boldsymbol{B}, \boldsymbol{\theta}) = \mathcal{N}(\text{vec}(\boldsymbol{W})|\boldsymbol{\mu}_q, \boldsymbol{\Sigma}_q), \tag{2}$$

where $\boldsymbol{\theta} = \{\boldsymbol{M}, \boldsymbol{\Omega}\}$ denotes the set of parameters for modeling $\boldsymbol{A}$'s posterior distribution, $\boldsymbol{\mu}_q = \text{vec}(\boldsymbol{W}_0 + \boldsymbol{B}\boldsymbol{M})$ is its mean, and $\boldsymbol{\Sigma}_q = [\boldsymbol{I}_n \otimes \boldsymbol{B}][\text{diag}(\text{vec}(\boldsymbol{\Omega})^2)][\boldsymbol{I}_n \otimes \boldsymbol{B}^\top]$ is its low-rank degenerate covariance matrix. In this paper, we adopt a similar approach for modeling the variational distribution of the weight posterior, focusing exclusively on Bayesianizing the weight update matrix $\boldsymbol{A}$.

## 3.2 TFB's Variational Low-Rank Isotropic Gaussians

**Variational Distribution Family.** In TFB, we constrain the variational distributions of the weight posterior to a more compact family of Gaussians than BLoB: specifically, we employ full-space isotropic Gaussian distributions projected onto the low-rank space:

$$q(\text{vec}(\boldsymbol{W})|\boldsymbol{B}, \boldsymbol{\theta}) = \mathcal{N}(\text{vec}(\boldsymbol{W})|\boldsymbol{\mu}_q, \text{proj}(\sigma_q^2\boldsymbol{I})), \tag{3}$$

where $\boldsymbol{\mu}_q$ is defined as in Eqn. 2. Here, $\sigma_q^2\boldsymbol{I} \in \mathbb{R}^{mn \times mn}$ represents a full-rank isotropic covariance matrix with standard deviation $\sigma_q$, and $\text{proj}(\cdot)$ denotes a linear projection operator that maps the full-space covariance matrix onto the low-rank space (see Proposition D.1.1 for details). [2]

**TFB as Generalized Variational Inference.** The choice of low-rank isotropic Gaussian approximate posteriors serves both *theoretical* and *empirical* purposes: it provides a single-parameter family that enables converting the generalized variational inference into a variance maximization problem (more details in Sec. 3.3, Theorem 4.2, and Appendix F.1), and empirically outperforms alternative distribution families (Sec. 5.4). Below, we present a practically efficient implementation for Bayesianizing LoRA under the constraint specified in Eqn. 3, with detailed theoretical analysis provided in Theorem 4.1.

**TFB in Practice.** Consider a LoRA layer with weight updates $\boldsymbol{B} \in \mathbb{R}^{m \times r}, \boldsymbol{A} \in \mathbb{R}^{r \times n}$ and a standard deviation scale $\sigma_q > 0$. We begin by computing the compact Singular Value Decomposition (SVD) [33] of $\boldsymbol{B}$:

$$\boldsymbol{B} = \boldsymbol{U} \, \text{diag}(\boldsymbol{d}) \boldsymbol{V}^\top, \tag{4}$$

---

[2] $\text{proj}(\cdot)$ only depends on the rank $r$ of the trained LoRA.

where $U \in \mathbb{R}^{m \times r}$ and $V \in \mathbb{R}^{r \times r}$ are orthonormal matrices, and $d = [d_1, d_2, \cdots, d_r]^\top$ is the vector consisting of singular values with all positive entries[3]. We then transform the original weight matrices $\{B, A\}$ into an equivalent pair

$$\{B' = U \operatorname{diag}(d), A' = V^\top A\}, \tag{5}$$

maintaining the equality $\Delta W = BA = B'A'$. Following BLoB's Asymmetric Bayesianization scheme, we define the variational distribution for $A'$ using the mean matrix $M = A'$ and the standard deviation matrix $\Omega \in \mathbb{R}^{r \times n}$, such that

$$q(A'_{ij}) = \mathcal{N}(A'_{ij}|M_{ij}, \Omega^2_{ij}), \forall i \in [r], \forall j \in [n]. \tag{6}$$

Unlike BLoB, our $\Omega$ is not freely parameterized but instead derived from projecting the full-space matrix $\sigma_q I$ onto the low-rank weight space:

$$\Omega_{ij} = \sigma_q/d_i, \quad \forall i \in [r], \forall j \in [n], \tag{7}$$

where $d$ is defined in Eqn. 4. This solution can be expressed compactly as $\Omega = [\sigma_q/d, \cdots, \sigma_q/d]$, comprising $n$ repeated vectors. To summarize, our TFB

- **takes as input** a trained LoRA matrix pair $\{B = U \operatorname{diag}(d)V^\top, A\}$ and a predetermined standard deviation $\sigma_q$, and
- **outputs** a "Bayesianized" LoRA adapter $\{B', A'\}$, where $B' = U \operatorname{diag}(d)$, and $A'$ becomes a distribution $q(A') = \prod_{i \in [r], j \in [n]} \mathcal{N}(A'_{ij}|M_{ij}, \Omega^2_{ij})$, with $M = V^\top A$, and $\Omega = [\sigma_q/d, \cdots, \sigma_q/d]$.

Note that the formulation in Eqn. 7 significantly improves memory efficiency during inference, reducing the storage for standard deviation parameters from $O(rn)$ to $O(r)$. While alternative parameterization approaches are possible, they must be capable of generating the low-rank isotropic Gaussian noises as demonstrated in Theorem 4.1. We have selected the current method (implementation) to ensure maximum compatibility with existing codebases [62]. In TFB, we use a single $\sigma_q$ shared across all LoRA layers.

### 3.3 TFB as Variance Maximization

The previous section presents a straightforward Bayesianization scheme **for a predetermined value of $\sigma_q$**. In this section, we describe a practical method for determining $\sigma_q$.

**A General Bayesianization Framework.** Consider an in-distribution "anchor" dataset $\mathcal{D}$, an associated evaluation metric $l$, and a performance change tolerance $\epsilon$. TFB determines $\sigma_q$ by solving a constrained optimization problem:

$$\begin{aligned} \max \quad & \sigma_q \\ s.t. \quad & |l(\mathcal{D}|B', M, \Omega(\sigma_q)) - l(\mathcal{D}|B, A)| \leq \epsilon, \end{aligned} \tag{8}$$

where $l(\mathcal{D}|B, A)$ and $l(\mathcal{D}|B', M, \Omega(\sigma_q)) = \mathbb{E}_{E \sim \mathcal{N}(0, \Omega^2)}[l(\mathcal{D}|B', M + E)]$ denote the pre- and post-Bayesianization performance, respectively. This optimization maximizes the noise scale $\sigma_q$ applied to model weights $M$ while ensuring that the resulting performance change remains within an acceptable threshold $\epsilon$.

**Anchor Dataset $\mathcal{D}$ and Evaluation Metric $l$.** Our *general* TFB framework accommodates various choices of anchor dataset $\mathcal{D}$ and evaluation metric $l$ based on practical requirements. Below, we consider two key scenarios (with $N$ being slightly overloaded in its notation).

*For supervised dataset $\mathcal{D} = \{x_n, y_n\}_{n=1}^N$:* The Negative-Log Likelihood (NLL) serves as a natural evaluation metric in Eqn. 8: $l_{\text{nll}}(\mathcal{D}|\theta) = -\frac{1}{N} \sum_{n=1}^N \log P_\theta(y_n|x_n)$, as it theoretically corresponds to minimizing the KL-regularized variational objective (more details in Sec. 4). The anchor dataset $\mathcal{D}$ can be either the original training set used for the LoRA model or an independent calibration dataset, as commonly employed in calibration-based methods [16, 70]. Alternative evaluation metrics such as accuracy or F1 score are also readily applicable. In our experimental setup, to ensure fair comparisons across uncertainty estimation baselines, we use the original training data as $\mathcal{D}$ (maintaining the same information access as baselines) and employ NLL as the evaluation metric. Additional results with accuracy as $l$ can be found in Appendix F.5.

---

[3]By stating $d \succ 0$, we assume $B$ has the full column rank $r$, which usually holds for LLM adaptation.

*For unsupervised dataset* $\mathcal{D} = \{\boldsymbol{x}_n\}_{n=1}^N$: One approach is to generate pseudo-labels $\widehat{y}$ using the model before Bayesianization, effectively converting the problem to the supervised case with $\mathcal{D} = \{\boldsymbol{x}_n, \widehat{y}_n\}_{n=1}^N$. TFB can also directly incorporate purely unsupervised metrics such as the expected embedding norm $l_{\text{emb}}(\mathcal{D}|\boldsymbol{\theta}) = \mathbb{E}_{\boldsymbol{x}\sim\mathcal{D}}[\|\text{emb}(\boldsymbol{x}|\boldsymbol{\theta})\|]$, where we are only concerned with properties of the representations themselves rather than any supervised signal. Hence our TFB offers substantially more flexibility compared to pure calibration methods, which typically rely on a labeled unseen calibration dataset. As a general framework, TFB also supports alternative evaluation metrics and statistical measures specifically designed for unsupervised data.

**Performance Change Tolerance** $\epsilon$. The selection of performance change tolerance $\epsilon$ is critical in TFB. While our experiments demonstrate that a fixed relative change rate, i.e., $\epsilon/p_0 = 0.3\%$ for NLL and $\epsilon/p_0 = 1\%$ for accuracy, where $p_0$ denotes the pre-Bayesianization performance, can achieve effective uncertainty estimation across various datasets and LoRA checkpoints, an adaptive $\epsilon$ can further improve the performance of TFB. Users can determine the appropriate value for $\epsilon$ by considering multiple factors simultaneously, among which the most important is the given LoRA checkpoint. For instance, an overfitted LoRA can typically accommodate a larger tolerance $\epsilon$ when using the training dataset (or its subset) as the anchor dataset. Additional properties of the data, model, and adaptation tasks can inform the choice of $\epsilon$ as well.

### 3.4 TFB: Final Algorithm

**Final TFB Algorithm: Automatically Determining** $\sigma_q$. Our final algorithm, presented in Algorithm 1 and Fig. 1, employs binary search to determine the optimal $\sigma_q^*$ within an initial range $[\sigma_{q_{\min}}, \sigma_{q_{\max}}]$. After identifying the optimal $\sigma_q^*$, we Bayesianize all LoRA layers using this value.

**Prediction.** For prediction, we average multiple outputs produced by samples from TFB's posterior:

$$P_{\boldsymbol{\theta}}(y|\boldsymbol{x}) = \mathbb{E}_{q(\boldsymbol{W}|\boldsymbol{\theta})}[P(y|\boldsymbol{x}, \boldsymbol{W})] \approx \tfrac{1}{N} \sum\nolimits_{n=1}^{N} P(\boldsymbol{y}|\boldsymbol{x}, \boldsymbol{W}_n), \quad \boldsymbol{W}_n \sim q(\boldsymbol{W}|\boldsymbol{\theta}), \tag{9}$$

where $q(\boldsymbol{W}|\boldsymbol{\theta})$ denotes the variational distribution defined in Eqn. 3, and we set the number of test-time samples to $N = 10$, following BLoB's protocol [62].

**Remark on TFB's Efficiency.** While TFB with binary search is efficient in terms of both time and memory (Appendix 5.3), and yields near-optimal solution of $\sigma_q$[4], more efficient parallel searching technique can be applied in practice. For instance, in Appendix F.10, we conduct a grid search across 8 different $\sigma_q$ values in parallel, construct an approximate function $\widehat{\sigma}_q(p)$ through piecewise linear interpolation of the observed performance, and estimate $\sigma_q^* \approx \widehat{\sigma}_q(p_0 - \epsilon)$, where $p_0$ denotes the model's performance before TFB.

## 4 Theoretical Analysis

In this section, we discuss our theoretical analysis, with complete proofs provided in Appendix D. First, we demonstrate that our TFB's Bayesianization scheme, defined in Equations 4, 5, and 7, projects a full-rank isotropic Gaussian distribution onto the low-rank space. We then prove that Eqn. 8 is equivalent to generalized variational inference for LLMs' weights under specific, achievable conditions, offering solid theoretical grounding for TFB.

**Assumption 4.1.** *The evaluation metric* $l_{\mathcal{D}} : \mathbb{R}_+ \to \mathbb{R}_+$ *is the Negative Log Likelihood (NLL) evaluated on the data distribution* $\mathcal{D}$ *for the variational standard deviation* $\sigma_q$:

$$l_{\mathcal{D}}(\sigma_q) = -\mathbb{E}_{(\boldsymbol{x},y)\sim\mathcal{D}, \boldsymbol{W}\sim q(\cdot|\sigma_q)}[\log P(y|\boldsymbol{x}, \boldsymbol{W})]. \tag{10}$$

*Furthermore, we assume* $l_{\mathcal{D}}$ *is locally convex, i.e.,* $\exists \epsilon_0 > 0$ *s.t.* $l_{\mathcal{D}}''(\sigma_q) > 0, \forall \sigma_q \in [0, \epsilon_0)$.

**Remark.** *The local convexity of the loss function is not unrealistic [43]. For instance, a local minimum* $\boldsymbol{W}_0$ *of a twice-differentiable loss function* $l$ *will imply the local convexity around* $\boldsymbol{W}_0$, *as assumed in Laplace Approximation [53, 6].*

**Theorem 4.1** (**Equivalent Variational Distribution of the Full Weight** $W$ **in** TFB)**.** *With the pre-trained weight matrix* $\boldsymbol{W}_0 \in \mathbb{R}^{m\times n}$, *the low-rank weight update matrix* $\{\boldsymbol{B}' \in \mathbb{R}^{m\times r}, \boldsymbol{A}' \in \mathbb{R}^{r\times n}\}$ *transformed from the given matrices* $\{\boldsymbol{B}, \boldsymbol{A}\}$ *following Eqn. 4 and 5, suppose that the variational*

---

[4]While traditional search algorithms require monotonicity within the search range to guarantee optimal solutions, empirically a near-optimal $\sigma_q$ is sufficient for effective uncertainty estimation.

distribution of $\boldsymbol{A}'$ is Gaussian $q(\boldsymbol{A}'|\boldsymbol{\theta}) = \prod_{ij} \mathcal{N}(A_{ij}|M_{ij}, \Omega_{ij}^2)$, where $\boldsymbol{M} = [M_{ij} = A'_{ij}] \in \mathbb{R}^{r \times n}$ is its mean and $\boldsymbol{\Omega} = [\Omega_{ij}] \in \mathbb{R}^{r \times n}$ is the standard deviation calculated as in Eqn. 7. The equivalent variational distribution $q(\text{vec}(\boldsymbol{W})|\sigma_q)$ defined on the full weight $\boldsymbol{W}$ is

$$
\begin{aligned}
q(\text{vec}(\boldsymbol{W})|\sigma_q) &= \mathcal{N}(\text{vec}(\boldsymbol{W})|\boldsymbol{\mu}_q, \boldsymbol{\Sigma}_q), \\
\text{where} \quad \boldsymbol{\mu}_q &= \text{vec}(\boldsymbol{W}_0 + \boldsymbol{B}'\boldsymbol{M}), \\
\boldsymbol{\Sigma}_q &= \sigma_q^2 \boldsymbol{I}_n \otimes \begin{bmatrix} \boldsymbol{I}_r & \\ & \boldsymbol{0}_{m-r} \end{bmatrix}.
\end{aligned}
\tag{11}
$$

Theorem 4.1 establishes that for any given $\sigma_q$, our algorithm for regrouping $\boldsymbol{B}, \boldsymbol{A}$ and computing the standard deviation matrix $\boldsymbol{\Omega}$ successfully constrains the corresponding full-weight variational distributions to the family of low-rank isotropic Gaussian distributions. This lays the foundation for the equivalence between our TFB and generalized variational inference to approximate the posterior distribution of LLM parameters (details in Theorem 4.2).

While alternative families of Gaussian distributions parameterized by a single scale $\sigma_q$ are possible, our empirical results demonstrate that our approach achieves superior performance (Sec. 5.4).

**Theorem 4.2** (TFB **as Generalized Variational Inference**). *Suppose the evaluation metric $l_\mathcal{D}(\sigma_q)$ defined following Assumption 4.1 is locally convex within the range of $\sigma_q \in [0, \epsilon_0)$. Suppose the approximate distribution of $\boldsymbol{W}$ given $\sigma_q$ is defined following Theorem 4.1. Suppose we have the prior distribution $P(\text{vec}(\boldsymbol{W})) = \mathcal{N}(\text{vec}(\boldsymbol{W})|\boldsymbol{\mu}_p, \boldsymbol{\Sigma}_p)$, where $\boldsymbol{\mu}_p = \boldsymbol{\mu}_q = \text{vec}(\boldsymbol{W}_0 + \boldsymbol{B}'\boldsymbol{M})$, and $\boldsymbol{\Sigma}_p = \sigma_p^2 \boldsymbol{I}$ with $\sigma_p > \epsilon_0$. Then for $\forall \lambda > 0, \exists \widetilde{\epsilon}$, s.t. the following two optimization problems (i) Generalized Variational Inference [7, 22, 30, 34]*

$$
\min_{\sigma_q} \quad l_\mathcal{D}(\sigma_q) + \lambda \, \text{KL}[q(\boldsymbol{W}|\sigma_q) \parallel P(\boldsymbol{W})],
\tag{12}
$$

*and (ii) Training-Free Bayesianization (TFB)*

$$
\begin{aligned}
\max \quad & \sigma_q \\
\text{s.t.} \quad & l_\mathcal{D}(\sigma_q) \leq \widetilde{\epsilon},
\end{aligned}
\tag{13}
$$

*are equivalent, i.e., the two optimization problems have the same optimal solution, where $\lambda$ is the regularization coefficient of the KL-divergence.*

This theorem provides the primary theoretical foundation for TFB. It demonstrates that under specific conditions – namely, local convexity within $[0, \epsilon_0)$ and prior standard deviation $\sigma_p > \epsilon_0$ – maximizing the scale $\sigma_q$ of the standard deviation matrix is equivalent to generalized variational inference [34], which approximates the posterior distribution of LLM parameters. Notably, when $\lambda = 1/|\mathcal{D}|$ is set to the reciprocal of the dataset size, generalized variational inference reduces to variational inference.

**Remark.** *TFB maintains theoretical soundness (through its equivalence to variational optimization) while offering practical simplicity, as it eliminates the need to explicitly specify the prior distribution's standard deviation $\sigma_p$. The condition $\sigma_p > \epsilon_0$ is naturally satisfied by common choices such as the standard normal distribution ($\sigma_p = 1$) or uniform distribution ($\sigma_p \to +\infty$).*

## 5 Experiments

We evaluate TFB through comprehensive experiments.

### 5.1 Settings

**Models, Datasets, and Evaluation.** We use the latest open-source `Meta-Llama-3.1-8B` as our primary LLM backbone while also providing additional results on other recent LLM architectures in Sec. 5.5, including `llama-2-7b-hf`, `Meta-Llama-3-8B`, and `Mistral-7B-v0.3` from the Llama [12] and Mistral [26] families.

For in-distribution experiments, we evaluate model performance on six commonsense reasoning tasks: Winogrande-Small (**WG-S**) and Winogrande-Medium (**WG-M**) [50], ARC-Challenge (**ARC-C**) and ARC-Easy (**ARC-E**) [11], Open Book Question Answering (**OBQA**) [42], and BoolQ [10]. Furthermore, we use models fine-tuned on OBQA [42] to evaluate their generalization ability on out-of-distribution datasets: college-level chemistry (**Chem**) and physics (**Phy**) subsets of MMLU [20]. Label spaces and prompt templates are detailed in Appendix E.1.

Table 1: **Performance of different methods applied to LoRA on Llama3.1-8B pre-trained weights,** where Accuracy (**ACC**) and Expected Calibration Error (**ECE**) are reported in percentages. **"TF?"** denotes whether a method is **T**raining-**F**ree. The evaluation is done across six common-sense reasoning tasks with a shared hyper-parameter setting after fine-tuning of 5 epochs. We use $N = 10$ samples during inference in all sampling-based methods including **BLoB [62]** and TFB. Rows with shading indicate training-free Bayesianization methods that use a pre-trained LoRA as their mean. Cells highlighted in **BLUE** indicate improved performance achieved by TFB compared to the weight mean. "↑" and "↓" indicate that higher and lower values are preferred, respectively. **Boldface** and underlining denote the best and the second-best performance, respectively.

| Metric | Method | TF? | In-Distribution Datasets | | | | | | Out-of-Distribution Datasets (OBQA→X) | | | |
| | | | | | | | | | Small Shift | | Large Shift | |
| | | | WG-S | ARC-C | ARC-E | WG-M | OBQA | BoolQ | ARC-C | ARC-E | Chem | Phy |
|---|---|---|---|---|---|---|---|---|---|---|---|---|
| ACC (↑) | MCD | ✗ | 78.03±0.61 | 81.64±1.79 | 91.37±0.38 | 83.18±0.84 | 87.20±1.02 | 89.93±0.16 | 81.42±1.38 | 87.27±0.84 | 47.92±2.25 | **46.53±0.49** |
| | ENS | ✗ | **78.82±0.52** | 82.55±0.42 | **91.84±0.36** | **83.99±0.74** | 87.37±0.67 | **90.50±0.14** | 79.62±0.57 | 86.56±0.60 | 49.65±3.22 | 44.44±1.96 |
| | LAP | BP | 76.05±0.92 | 79.95±0.42 | 90.73±0.08 | 82.83±0.85 | 87.90±0.20 | 89.36±0.52 | 81.08±1.20 | 87.21±1.20 | 48.26±3.93 | 46.18±1.30 |
| | MonteCLoRA | ✗ | 69.20±0.18 | 78.38±0.89 | 90.79±0.62 | 74.79±0.23 | 84.13±0.31 | 89.17±0.30 | 79.63±0.87 | 86.58±0.49 | **50.00±1.04** | 42.01±2.41 |
| | BLoB | ✗ | 76.45±0.37 | 82.32±1.15 | 91.14±0.54 | 82.01±0.56 | 87.57±0.21 | 89.65±0.15 | 79.75±0.43 | 87.13±0.00 | 42.71±3.71 | 44.79±6.64 |
| | MLE | - | 77.87±0.54 | 81.08±0.48 | 91.67±0.36 | 82.30±0.53 | 87.90±0.87 | 89.58±0.26 | 81.48±2.41 | 86.83±0.87 | 45.83±0.85 | 42.36±1.77 |
| | + TFB (Ours) | ✓ | 77.44±0.30 | 82.53±1.00 | 91.33±0.37 | 82.53±0.56 | **88.53±0.57** | 89.75±0.25 | 79.76±1.24 | 85.52±0.56 | 44.33±4.03 | 37.00±2.16 |
| | MAP | - | 76.90±0.97 | 81.08±2.48 | 91.61±0.44 | 82.59±0.28 | 85.73±0.19 | 90.09±0.28 | 79.98±0.87 | 86.58±0.79 | 43.40±4.98 | 38.54±3.40 |
| | + TFB (Ours) | ✓ | 76.43±0.72 | 82.80±1.42 | 91.39±0.37 | 82.64±0.58 | 86.00±0.16 | 89.96±0.18 | 80.61±1.24 | 86.30±0.89 | 45.33±2.87 | 35.67±4.11 |
| | BLoB-Mean | ✗ | 77.72±0.12 | 82.60±0.60 | 91.64±0.55 | 83.92±0.48 | 88.00±0.00 | 89.86±0.05 | 82.06±1.15 | **88.54±0.31** | 39.93±5.20 | 39.93±4.02 |
| | + TFB (Ours) | ✓ | 77.81±0.36 | **83.33±0.19** | 91.76±0.48 | 83.81±0.39 | 87.80±0.16 | **90.11±0.28** | 82.93±1.54 | 87.64±0.51 | 39.67±7.32 | 37.33±6.65 |
| ECE (↓) | MCD | ✗ | 16.13±0.54 | 13.69±1.11 | 6.73±0.71 | 13.05±0.99 | 9.76±0.71 | 7.95±0.17 | 13.63±1.18 | 9.27±0.60 | 30.91±3.57 | 33.08±1.40 |
| | ENS | ✗ | 14.72±0.17 | 13.45±1.19 | 6.59±0.45 | 11.17±0.92 | 8.17±0.86 | 7.35±0.55 | 11.37±1.82 | 7.21±1.13 | 18.92±6.03 | 26.80±3.23 |
| | LAP | BP | **4.18±0.11** | 9.26±3.08 | 5.27±0.51 | **3.50±0.78** | 8.93±0.34 | **1.93±0.22** | 7.83±1.49 | 7.80±1.99 | **14.49±0.57** | 13.17±2.14 |
| | MonteCLoRA | ✗ | 18.29±0.27 | 12.22±0.75 | 7.23±0.71 | 15.97±0.45 | 9.79±0.07 | 7.09±0.52 | 10.65±0.53 | 8.18±0.26 | 23.21±0.17 | 30.39±4.76 |
| | BLoB | ✗ | 9.93±0.22 | **5.41±1.17** | 2.70±0.87 | 4.28±0.64 | 2.91±0.92 | 2.58±0.25 | **5.61±0.40** | 2.48±0.43 | 16.67±0.87 | **12.78±4.18** |
| | MLE | - | 17.02±0.46 | 16.35±0.68 | 7.00±0.53 | 13.83±0.65 | 9.77±0.81 | 8.69±0.21 | 14.45±2.19 | 10.78±0.50 | 32.46±2.60 | 38.41±4.44 |
| | + TFB (Ours) | ✓ | 12.98±0.37 | 11.63±0.68 | 5.14±0.14 | 10.01±0.70 | 7.20±0.47 | 7.39±0.26 | 6.54±0.53 | 5.69±1.64 | 14.63±1.46 | 19.68±3.27 |
| | MAP | - | 18.71±0.74 | 15.77±1.60 | 6.62±0.64 | 14.26±0.92 | 12.19±0.55 | 8.40±0.25 | 16.46±0.44 | 11.36±0.58 | 34.79±3.76 | 38.50±2.18 |
| | + TFB (Ours) | ✓ | 14.95±0.65 | 11.27±2.53 | 5.76±0.63 | 10.97±1.19 | 9.70±0.69 | 6.86±0.31 | 13.25±0.95 | 9.22±0.91 | 27.21±2.62 | 35.91±4.12 |
| | BLoB-Mean | ✗ | 15.43±0.15 | 12.41±1.52 | 4.91±0.28 | 9.37±1.33 | 6.44±0.15 | 6.26±0.29 | 11.22±0.38 | 6.34±0.71 | 26.65±3.06 | 25.40±5.40 |
| | + TFB (Ours) | ✓ | 8.16±0.48 | 6.48±0.36 | **2.44±0.50** | 3.83±0.43 | **2.67±0.18** | 3.10±0.59 | 6.69±1.63 | 3.61±0.87 | 18.45±6.75 | 20.53±6.27 |
| NLL (↓) | MCD | ✗ | 0.83±0.01 | 0.99±0.10 | 0.45±0.06 | 0.64±0.03 | 0.62±0.08 | 0.49±0.01 | 1.03±0.02 | 0.61±0.03 | 1.91±0.18 | 2.02±0.15 |
| | ENS | ✗ | 0.75±0.02 | 0.80±0.11 | 0.38±0.05 | 0.55±0.02 | 0.45±0.05 | 0.42±0.05 | 0.72±0.07 | 0.44±0.03 | 1.40±0.18 | 1.50±0.13 |
| | LAP | BP | 0.56±0.00 | 1.18±0.02 | 1.04±0.01 | 0.51±0.00 | 0.94±0.00 | 0.43±0.00 | 1.17±0.01 | 1.11±0.00 | **1.27±0.01** | **1.28±0.00** |
| | MonteCLoRA | ✗ | 0.82±0.02 | 0.71±0.03 | 0.51±0.04 | 0.74±0.02 | 0.55±0.02 | 0.36±0.02 | 0.68±0.03 | 0.49±0.01 | 1.43±0.00 | 1.44±0.06 |
| | BLoB | ✗ | 0.58±0.00 | **0.51±0.03** | 0.23±0.01 | 0.43±0.01 | 0.34±0.01 | 0.26±0.01 | 0.56±0.02 | 0.35±0.02 | 1.34±0.04 | 1.35±0.10 |
| | MLE | - | 0.88±0.01 | 1.20±0.11 | 0.46±0.04 | 0.68±0.01 | 0.61±0.06 | 0.52±0.01 | 1.07±0.06 | 0.72±0.06 | 1.91±0.16 | 2.25±0.21 |
| | + TFB (Ours) | ✓ | 0.68±0.03 | 0.85±0.02 | 0.33±0.03 | 0.53±0.01 | 0.46±0.04 | 0.42±0.00 | 0.66±0.02 | 0.44±0.01 | 1.39±0.11 | 1.49±0.05 |
| | MAP | - | 0.99±0.07 | 1.12±0.23 | 0.46±0.03 | 0.74±0.07 | 0.79±0.02 | 0.52±0.01 | 1.19±0.04 | 0.83±0.06 | 1.97±0.13 | 2.32±0.10 |
| | + TFB (Ours) | ✓ | 0.77±0.05 | 0.80±0.15 | 0.38±0.03 | 0.57±0.05 | 0.61±0.03 | 0.40±0.01 | 0.96±0.08 | 0.66±0.06 | 1.69±0.16 | 2.12±0.08 |
| | BLoB-Mean | ✗ | 0.74±0.02 | 0.73±0.04 | 0.29±0.03 | 0.47±0.03 | 0.37±0.02 | 0.32±0.02 | 0.67±0.07 | 0.39±0.03 | 1.53±0.13 | 1.54±0.15 |
| | + TFB (Ours) | ✓ | **0.55±0.01** | **0.53±0.04** | 0.23±0.02 | **0.40±0.01** | **0.33±0.02** | 0.27±0.01 | **0.52±0.05** | 0.35±0.02 | 1.36±0.13 | 1.46±0.11 |

To assess uncertainty estimation, we measure Expected Calibration Error (**ECE** [45]) and Negative Log-Likelihood (**NLL**) on the test dataset. We also report Accuracy (**ACC**) to ensure models maintain strong performance. Additional evaluation details are provided in Appendix E.2.

**Baselines.** We compare TFB with state-of-the-art uncertainty estimation methods for LoRA-adapted LLMs, including ensemble-based method: Deep Ensemble (**ENS**) [39, 4, 61], variational inference methods: Monte-Carlo Dropout (**MCD**) [14], Monte Carlo-enhanced LoRA (**MonteCLoRA**) [51], Bayesian LoRA by Backprop (**BLoB**) [62], and post-training method: Laplace-LoRA (**LAP**) [67]. For reference, we also include two standard Parameter-Efficient Fine-Tuning (PEFT) baselines: Maximum Likelihood Estimation (**MLE**) [23] and Maximum A Posteriori (**MAP**). All baselines are implemented following the protocols established in BLoB, detailed in Appendix E.4.

TFB **Implementation.** TFB can be directly applied to trained LoRA adapters without additional training. As indicated by the **"TF?"** column in Table 1, TFB is **T**raining-**F**ree and requires only LLM inference (✓), while the other methods need full retraining (✗) or gradient estimation with Backpropagation (BP). We evaluate TFB on three off-the-shelf LoRA checkpoints: **MLE**, **MAP**, and the mean component of **BLoB** (obtained by discarding BLoB's standard deviation matrix $\mathbf{\Omega}$). More details are included in Appendix E.4.

Table 2: **A comparison of running time and maximum GPU memory cost between** TFB **and** BLoB **during the process of Bayesianizatioin.** The experiments are conducted on a single NVIDIA A100 GPU. The subscripts in the table calculate the relative cost of a method compared to that of LoRA, a non-Bayesian baseline method. **RED** and **GREEN** represent **worse** and **better** efficiency, respectivley. Note that varying batch sizes do not impact the performance of TFB, as Algorithm 1 is independent of gradient and batch size

| Method | Batch Size | Datasets | | | | | | | | | | | |
| --- | --- | --- | --- | --- | --- | --- | --- | --- | --- | --- | --- | --- | --- |
| | | WG-S | | ARC-C | | ARC-E | | WG-M | | OBQA | | BoolQ | |
| | | Time (s) | Mem. (MB) | Time (s) | Mem. (MB) | Time (s) | Mem. (MB) | Time (s) | Mem. (MB) | Time (s) | Mem. (MB) | Time (s) | Mem. (MB) |
| LoRA | 4 | 338 | 12,894 | 632 | 19,762 | 1,238 | 18,640 | 1,339 | 13,164 | 2,692 | 17,208 | 6,489 | 29,450 |
| BLoB | 4 | 371 (1.10x) | 13,194 (1.02x) | 685 (1.08x) | 21,736 (1.10x) | 1,360 (1.10x) | 20,700 (1.11x) | 1,476 (1.10x) | 13,194 (1.00x) | 3,257 (1.21x) | 18,046 (1.05x) | 7,251 (1.12x) | 30,578 (1.04x) |
| TFB (Ours) | 4 | 1,203 (3.56x) | 10,372 (0.80x) | 1,257 (1.99x) | 11,966 (0.61x) | 1,246 (1.01x) | 11,202 (0.60x) | 1,237 (0.92x) | 10,344 (0.79x) | 1,238 (0.46x) | 10,376 (0.60x) | 1,452 (0.22x) | 16,340 (0.55x) |
| TFB (Ours) | 8 | 628 (1.86x) | 10,666 (0.83x) | 731 (1.16x) | 15,286 (0.77x) | 702 (0.57x) | 12,598 (0.68x) | 634 (0.47x) | 10,662 (0.81x) | 642 (0.24x) | 12,116 (0.70x) | 1,015 (0.16x) | 22,146 (0.75x) |
| TFB (Ours) | 12 | 446 (1.31x) | 12,064 (0.93x) | 599 (0.94x) | 18,204 (0.92x) | 540 (0.43x) | 14,310 (0.76x) | 441 (0.32x) | 11,370 (0.86x) | 487 (0.18x) | 13,410 (0.77x) | 908 (0.13x) | 25,220 (0.85x) |

## 5.2 TFB **Improves Accuracy and Uncertainty Estimation across Distributional Shifts**

Table 1 shows results on comprehensive metrics for various methods applied to LoRA on Llama3.1-8B pre-trained weights. More empirical results on Llama2-7B can be found in Appendix F.10.

**In-Distribution Results.** The addition of TFB maintains competitive accuracy while substantially improving model calibration across in-distribution datasets. For ECE, TFB yields notable improvements when applied to different base methods: MLE+TFB reduces ECE to 5.14% on ARC-E (from 7.00%); similarly MAP+TFB and BLoB-Mean+TFB reduce ECE to 9.70% on OBQA (from 12.19%) and 3.83% on WG-M (from 9.37%), respectively. For NLL, TFB consistently produces better-calibrated predictions, with BLoB-Mean+TFB achieving strong performance across datasets: 0.23 on ARC-E (from 0.29), 0.33 on OBQA (from 0.37), and 0.27 on BoolQ (from 0.32). These improvements in both ECE and NLL demonstrate TFB's effectiveness in enhancing model calibration while preserving accuracy on in-distribution tasks.

**Out-of-Distribution Results.** For out-of-distribution datasets, which represent a more challenging evaluation scenario, TFB continues to show benefits, though the performance gaps are generally smaller. In both Small Shift and Large Shift scenarios, TFB-enhanced methods maintain relatively strong performance, particularly in the Small Shift cases (ARC-C and ARC-E). However, there's a noticeable performance drop in the Large Shift scenarios (Chem and Phy), which is expected given the significant domain difference. Even in these challenging cases, TFB-enhanced methods tend to maintain better calibration (lower ECE scores) compared to their base counterparts, suggesting improved reliability in out-of-distribution settings.

## 5.3 **Computational Efficiency of** TFB

We compare the computational efficiency of TFB and BLoB during the process of Bayesianization [62] in Table 2. We also report the computational cost of the standard LoRA fine-tuning as reference. All three methods are evaluated on the configurations detailed in Appendix E.4. For LoRA and BLoB, the evaluation of running time and maximum GPU memory is based on fine-tuning for 5 epochs. TFB uses a fixed number of 500 training examples to search for $\sigma_q^*$ across all datasets, and performs binary search for at most 5 rounds (sequentially).

As shown in the table, TFB can be slower on small datasets (e.g., WG-S with ∼600 samples, nearly 3× slower than BLoB under the same batch size). However, on larger datasets (e.g., BoolQ with ∼10,000 samples), TFB **is up to 5× faster while using only half the GPU memory.** Since TFB avoids gradient estimation, memory use is substantially reduced, allowing larger batch sizes; increasing from 4 to 12 already yields lower time and memory than BLoB on most datasets. Remarkably, TFB is even more efficient than standard LoRA fine-tuning, thanks to its training-free nature. Importantly, *our efficiency results cover the entire process of searching $\sigma$*, whereas baselines report only successful runs and *exclude hyperparameter tuning costs,* biasing the comparison in their favor. Despite this, TFB still achieves superior time and memory efficiency, highlighting the advantages of its training-free approach and flexibility to trade off speed and memory under resource constraints.

Table 3: **Performance of** TFB **with different variational distribution families applied to BLoB-Mean on Llama3.1-8B pre-trained weights. FR:** Full-rank isotropic Gaussian noises are applied to $\Delta W$; **C-STD:** Standard deviation matrix $\mathbf{\Omega} = [\Omega_{ij} = \sigma_q]$ is constant. The evaluation protocol strictly follows Table 1. **"Rk.":** Average ranking of each method when compared to all other approaches on in-distribution datasets. "↑" and "↓" indicate that higher and lower values are preferred, respectively. **Boldface** and underlining denote the best and the second-best performance, respectively.

| Metric | Method | In-Distribution Datasets | | | | | | | Out-of-Distribution Datasets (OBQA→X) | | | |
| | | | | | | | | | Small Shift | | Large Shift | |
| | | WG-S | ARC-C | ARC-E | WG-M | OBQA | BoolQ | Rk. (↓) | ARC-C | ARC-E | Chem | Phy |
|---|---|---|---|---|---|---|---|---|---|---|---|---|
| ACC (↑) | BLoB-Mean | $\underline{77.72}_{\pm0.12}$ | $82.60_{\pm0.60}$ | $\underline{91.64}_{\pm0.55}$ | $\mathbf{83.92}_{\pm0.48}$ | $88.00_{\pm0.80}$ | $89.86_{\pm0.05}$ | 2.50 | $82.06_{\pm1.15}$ | $\mathbf{88.54}_{\pm0.31}$ | $39.93_{\pm5.20}$ | $39.93_{\pm4.02}$ |
| | + TFB (FR) | $75.57_{\pm0.25}$ | $\underline{83.20}_{\pm0.65}$ | $91.58_{\pm0.67}$ | $82.19_{\pm1.09}$ | $\mathbf{88.73}_{\pm0.41}$ | $89.46_{\pm0.17}$ | 2.83 | $81.33_{\pm0.82}$ | $88.06_{\pm0.75}$ | $42.00_{\pm2.16}$ | $\mathbf{41.33}_{\pm5.44}$ |
| | + TFB (C-STD) | $76.35_{\pm0.08}$ | $\underline{83.20}_{\pm0.33}$ | $91.33_{\pm0.70}$ | $81.79_{\pm0.51}$ | $88.20_{\pm0.57}$ | $89.65_{\pm0.08}$ | 3.00 | $81.73_{\pm0.68}$ | $88.18_{\pm0.65}$ | $\mathbf{43.00}_{\pm1.41}$ | $39.33_{\pm3.86}$ |
| | + TFB (Final) | $\mathbf{77.81}_{\pm0.36}$ | $\mathbf{83.33}_{\pm0.19}$ | $\mathbf{91.76}_{\pm0.48}$ | $\underline{83.81}_{\pm0.39}$ | $87.80_{\pm0.16}$ | $\mathbf{90.11}_{\pm0.28}$ | 1.67 | $\mathbf{82.93}_{\pm1.54}$ | $\underline{87.64}_{\pm0.51}$ | $39.67_{\pm7.32}$ | $37.33_{\pm6.65}$ |
| ECE (↓) | BLoB-Mean | $15.43_{\pm0.15}$ | $12.41_{\pm1.52}$ | $4.91_{\pm0.28}$ | $9.37_{\pm1.33}$ | $6.44_{\pm0.15}$ | $6.26_{\pm0.29}$ | 4.00 | $11.22_{\pm0.38}$ | $6.34_{\pm0.71}$ | $26.65_{\pm3.06}$ | $25.40_{\pm5.40}$ |
| | + TFB (FR) | $10.42_{\pm0.29}$ | $7.45_{\pm0.88}$ | $\mathbf{2.01}_{\pm1.03}$ | $4.36_{\pm0.68}$ | $3.70_{\pm1.04}$ | $3.62_{\pm0.10}$ | 2.67 | $7.19_{\pm1.40}$ | $\underline{3.29}_{\pm1.03}$ | $\mathbf{17.78}_{\pm1.01}$ | $\mathbf{19.14}_{\pm4.01}$ |
| | + TFB (C-STD) | $9.23_{\pm0.20}$ | $\mathbf{5.98}_{\pm0.32}$ | $2.94_{\pm0.67}$ | $\underline{3.86}_{\pm0.45}$ | $\underline{3.17}_{\pm0.21}$ | $\mathbf{2.82}_{\pm0.62}$ | 1.83 | $\underline{6.89}_{\pm0.89}$ | $\mathbf{2.76}_{\pm0.88}$ | $18.27_{\pm2.52}$ | $19.45_{\pm3.46}$ |
| | + TFB (Final) | $\mathbf{8.16}_{\pm0.48}$ | $\underline{6.48}_{\pm0.36}$ | $\underline{2.44}_{\pm0.50}$ | $\mathbf{3.83}_{\pm0.43}$ | $\mathbf{2.67}_{\pm0.18}$ | $3.10_{\pm0.59}$ | 1.50 | $\mathbf{6.69}_{\pm1.63}$ | $3.61_{\pm0.87}$ | $18.45_{\pm6.75}$ | $20.53_{\pm6.27}$ |
| NLL (↓) | BLoB-Mean | $0.74_{\pm0.02}$ | $0.73_{\pm0.04}$ | $0.29_{\pm0.03}$ | $0.47_{\pm0.03}$ | $\underline{0.37}_{\pm0.02}$ | $0.32_{\pm0.02}$ | 3.67 | $0.67_{\pm0.07}$ | $0.39_{\pm0.03}$ | $1.53_{\pm0.13}$ | $1.54_{\pm0.15}$ |
| | + TFB (FR) | $0.60_{\pm0.01}$ | $0.53_{\pm0.03}$ | $0.23_{\pm0.02}$ | $0.43_{\pm0.01}$ | $\mathbf{0.33}_{\pm0.02}$ | $0.27_{\pm0.01}$ | 2.00 | $0.57_{\pm0.04}$ | $0.34_{\pm0.02}$ | $\mathbf{1.34}_{\pm0.07}$ | $1.42_{\pm0.09}$ |
| | + TFB (C-STD) | $\underline{0.57}_{\pm0.01}$ | $\mathbf{0.51}_{\pm0.02}$ | $\mathbf{0.22}_{\pm0.01}$ | $0.43_{\pm0.01}$ | $\mathbf{0.33}_{\pm0.01}$ | $\mathbf{0.26}_{\pm0.01}$ | 1.33 | $\underline{0.56}_{\pm0.04}$ | $\mathbf{0.33}_{\pm0.02}$ | $\mathbf{1.34}_{\pm0.08}$ | $\mathbf{1.41}_{\pm0.09}$ |
| | + TFB (Final) | $\mathbf{0.55}_{\pm0.01}$ | $\underline{0.53}_{\pm0.04}$ | $\underline{0.23}_{\pm0.02}$ | $\mathbf{0.40}_{\pm0.01}$ | $\mathbf{0.33}_{\pm0.02}$ | $0.27_{\pm0.01}$ | 1.50 | $\mathbf{0.52}_{\pm0.05}$ | $0.35_{\pm0.02}$ | $\underline{1.36}_{\pm0.13}$ | $1.46_{\pm0.11}$ |

## 5.4 TFB **Beyond the Low-Rank Isotropic Gaussians**

In this section, we consider two simple TFB variants with other families of Gaussians for modeling the variational distributions of $W$: (i) Full-Rank Isotropic Gaussian (**FR**, $\Sigma_q = \sigma_q^2 I$), and (ii) Constant Low-Rank Standard Deviation (**C-STD**, $\mathbf{\Omega} = [\Omega_{ij} = \sigma_q]$). Similar to our final TFB, both distributions are controlled by a single $\sigma_q$ parameter and fit the maximal variance search in Eqn. 8. For fair comparison, we adopt the same optimal $\sigma_q^*$ search protocol as described in Sec. 5.1. Table 3 shows the performances of TFB and its variants applied to the mean of BLoB (more in Table 11 of Appendix F.6).

These results show that our final TFB outperforms both variants **FR** and **C-STD** across multiple metrics on in-distribution datasets, with notable improvements in calibration (ECE reduced by up to 15.77%) and accuracy (e.g., 77.81% on WG-S). While **C-STD** shows better NLL scores, the improvement comes at the cost of a significantly degraded overall performance—particularly in accuracy, where it performs the worst—making it impractical for real-world applications. Although our final TFB maintains strong performance on datasets with smaller distributional shifts, its advantages diminish on datasets with larger shifts in the domains of Physics and Chemistry.

Table 4: Performance of different **LLM backbones** on the combined dataset.

| Method | ACC (↑) | ECE (↓) | NLL (↓) |
|---|---|---|---|
| Llama2-7B | $\mathbf{81.41}_{\pm0.64}$ | $4.50_{\pm0.37}$ | $\mathbf{0.43}_{\pm0.00}$ |
| + TFB (Ours) | $81.32_{\pm0.51}$ | $\mathbf{1.24}_{\pm0.22}$ | $\mathbf{0.43}_{\pm0.00}$ |
| Llama3-8B | $\mathbf{86.93}_{\pm0.09}$ | $4.28_{\pm0.54}$ | $\mathbf{0.34}_{\pm0.00}$ |
| + TFB (Ours) | $86.61_{\pm0.20}$ | $\mathbf{1.64}_{\pm0.64}$ | $\mathbf{0.34}_{\pm0.00}$ |
| Llama3.1-8B | $\mathbf{86.70}_{\pm0.08}$ | $4.74_{\pm0.28}$ | $0.35_{\pm0.00}$ |
| + TFB (Ours) | $86.45_{\pm0.33}$ | $\mathbf{1.05}_{\pm0.06}$ | $\mathbf{0.34}_{\pm0.00}$ |
| Mistral-7B-v0.3 | $\mathbf{86.88}_{\pm0.51}$ | $5.05_{\pm0.88}$ | $0.35_{\pm0.02}$ |
| + TFB (Ours) | $86.64_{\pm0.28}$ | $\mathbf{1.68}_{\pm0.53}$ | $\mathbf{0.33}_{\pm0.01}$ |

**Advantages of Final** TFB**'s Variational Low-Rank Isotropic Gaussians.** Compared to *TFB (FR)* and *TFB (C-STD)*, *TFB (Final)* offers additional advantages. It is computationally more efficient than **FR** with noise complexity of $O(rn)$ versus $O(mn)$. Furthermore, unlike **C-STD** whose variational distributions vary with different but equivalent LoRA matrix pairs (see Appendix F.6 for details), *TFB (Final)* produces consistent Bayesianization for all equivalent LoRAs satisfying $BA = \Delta W$.

## 5.5 TFB **Beyond the Llama3.1-8B Backbone**

We conduct comprehensive experiments across multiple LLM backbones to validate our approach. Our experiments span several models from the Llama family [55, 12], including `llama-2-7b-hf`, `Meta-Llama-3-8B`, and `Meta-Llama-3.1-8B`. While we initially considered `Llama-3.2-1B`, we ultimately discarded its results due to poor adaptation performance with the smaller model architecture. We also extend our analysis to include `Mistral-7B-v0.3` [26].

Following commonsense-170k [24, 60], we combine the 6 reasoning sub-tasks from the main experiments with one shared label space (**Combined**) and train the base LoRA adapters with MLE. Break-down statistics of each sub-dataset are available in Appendix F.7. Table 4 shows the results, demonstrating TFB's effectiveness: it dramatically reduces ECE across all models (e.g., from 4.74% to 1.05% for Llama3.1-8B) while maintaining strong ACC and NLL scores.

## 5.6 TFB Beyond the Naive LoRA

Our proposed TFB is general, compatible with various LoRA-based methods that use different initialization strategies [41], parameter sharing schemes [35], and optimization approaches [68]. We evaluate TFB on two representative LoRA variants (see Appendix F.8 for details):

- VeRA [35]: Uses shared low-rank matrices $B$ and $A$ across layers, with the layer-specific trainable scalar vector $d$ and the bias vector $b$.
- PiSSA [41]: Employs an alternative initialization while maintaining LoRA's training process.

Table 5: Performance of different **LoRA-like PEFTs** on the combined dataset.

| Method | ACC (↑) | ECE (↓) | NLL (↓) |
|---|---|---|---|
| LoRA | **86.70**±0.08 | 4.74±0.28 | 0.35±0.00 |
| + TFB (Ours) | 86.45±0.33 | **1.05**±0.06 | **0.34**±0.00 |
| VeRA | **84.93**±0.50 | 5.11±0.55 | 0.39±0.01 |
| + TFB (Ours) | 84.28±0.48 | **1.44**±0.44 | **0.38**±0.01 |
| PiSSA | **86.83**±0.51 | 4.26±0.14 | 0.35±0.00 |
| + TFB (Ours) | 86.61±0.43 | **1.17**±0.22 | **0.33**±0.00 |

Table 5 shows the results, demonstrating TFB's broad applicability: it substantially reduces ECE across all LoRA-like PEFT methods (e.g., from 5.11 to 1.44 for VeRA). Importantly, it maintains strong ACC and NLL with minimal performance degradation, validating the effectiveness of low-rank isotropic Gaussian distributions for variational inference of LLMs.

## 5.7 Improving Inference-Time Efficiency of TFB

Inspired by Harrison et al. [19], in this section, we study the compactibility of TFB and Last-Layer Bayesianization to speed up inference. Specifically, we employ the same standard deviation $\sigma_q^*$ as identified through the full-model TFB, but apply the Bayesianization to the last layer only. By limiting inference to sampling

Table 6: Performance of **Last-Layer** TFB (LL TFB) applied to the combined dataset.

| Method | #Sample (N) | ACC (↑) | ECE (↓) | NLL (↓) | Time (s) |
|---|---|---|---|---|---|
| MLE | - | 86.70±0.08 | 4.74±0.28 | 0.35±0.00 | 118 |
| + TFB | 10 | 86.45±0.33 | 1.05±0.06 | 0.34±0.00 | 1,114 |
| + LL TFB | 10 | 86.53±0.12 | 1.14±0.03 | 0.34±0.00 | 182 |
| + LL TFB | 100 | **86.75**±0.09 | **0.92**±0.10 | **0.33**±0.00 | 924 |

from the variational distribution of the last layer, the outputs of earlier layers can be reused, significantly improving inference speed. Although restricting Bayesianization to the last layer sacrifices some richness of the variational distribution family, this approach allows for a higher number of posterior samples, resulting in more accurate approximations.

Table 6 shows the results. When only $N = 10$ samples are used, the uncertainty estimation of last-layer Bayesianization performs worse than that of full-model Bayesianization. Nevertheless, the improved posterior estimation with $N = 100$ samples enables last-layer Bayesianization to achieve better performance. For a comprehensive analysis of its performance, please refer to Appendix F.9.

## 6 Conclusion

In this paper, we introduce Training-Free Bayesianization (TFB), a novel framework that transforms trained LoRA adapters into Bayesian ones without additional training. By systematically searching for the maximally acceptable variance in the weight posterior within a family of low-rank isotropic Gaussian distributions, TFB provides a practical solution to uncertainty estimation in LLMs. Our theoretical analysis shows that TFB's variance maximization process is equivalent to generalized variational inference under mild conditions. Our empirical results verify its superior performance across various settings and model configurations. Our framework's simplicity and effectiveness, requiring only an anchor dataset for search, makes it widely applicable across different domains. As LLMs continue to evolve, TFB represents a significant step toward more reliable and uncertainty-aware AI systems, paving the way for future research in adaptive and trustworthy machine learning. For **Limitations**, please refer to Appendix B.

## Acknowledgement

We thank all reviewers, AC, and SAC for their valuable comments. HW is supported by Amazon Faculty Research Award, Microsoft AI & Society Fellowship, NSF CAREER Award IIS-2340125, NIH grant R01CA297832, and NSF grant IIS-2127918.

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

# Appendix

In Appendix A, we present the full algorithmic description of TFB. In Appendix B, we present the limitations of TFB. Next, in Appendix C, we present a more detailed introduction to recent advances of Bayesian Low-Rank Adaptation. In Appendix D, we provide detailed proofs for all theorems presented in the main paper. In Appendix E, we describe our experimental methodology. Finally, in Appendix F, we present additional empirical results, including:

- a visual study demonstrating TFB functions as a general Bayesian Neural Network (Appendix F.1),
- test-time sample size analysis (Appendix F.2),
- anchor dataset size analysis (Appendix F.3),
- TFB with unlabeled test data as the anchor data (Appendix F.4),
- TFB with non-NLL evaluation metrics (Appendix F.5),
- TFB with other single-parameter variational distribution families (Appendix F.6),
- TFB with other LLM backbones than Llama3.1-8B (Appendix F.7),
- TFB with other PEFT methods than LoRA (Appendix F.8),
- improving inference-time efficiency of TFB with last-layer Bayesianization (Appendix F.9), and
- the full results on a widely used LLM architecture Llama2-7B (Appendix F.10).

## A  TFB: Algorithm

---
**Algorithm 1 Training-Free Bayesianization (TFB)**

---
**input** $\mathcal{D}$: Anchor Dataset;
  $\{\boldsymbol{B}, \boldsymbol{A}\}$: Low-Rank Component;
  $l$: Model Evaluation Metric;
  $\epsilon$: Performance Change Tolerance;
  $[\sigma_{q_{\min}}, \sigma_{q_{\max}}]$: search range of $\sigma_q$.
1: Evaluate the original performance: $p_0 \leftarrow l(\mathcal{D}|\boldsymbol{B}, \boldsymbol{A})$.
2: Singular Value Decomposition on $\boldsymbol{B}$:
  $\boldsymbol{U}, \mathrm{diag}(\boldsymbol{d}), \boldsymbol{V} \leftarrow \mathrm{SVD}(\boldsymbol{B})$.                          ▷ Eqn. 4.
3: Get an equivalent pair of the low-rank component:
  $\boldsymbol{B}' \leftarrow \boldsymbol{U}\,\mathrm{diag}(\boldsymbol{d}); \boldsymbol{A}' \leftarrow \boldsymbol{V}^\top \boldsymbol{A}$.                          ▷ Eqn. 5.
4: **while** $\sigma_q$ not converged **do**
5:   $\sigma_q \leftarrow (\sigma_{q_{\max}} + \sigma_{q_{\min}})/2$.
6:   Calculate the standard deviation matrix $\boldsymbol{\Omega}$ for $\boldsymbol{A}'$:
    $\Omega_{ij} = \sigma_q/d_i$.                          ▷ Eqn. 7.
7:   Evaluate the performance:
    $p \leftarrow l(\mathcal{D}|\boldsymbol{B}', \boldsymbol{A}', \boldsymbol{\Omega})$.
8:   **if** $|p - p_0| < \epsilon$ **then**
9:     $\sigma_{q_{\min}} \leftarrow \sigma_q$.
10:   **else**
11:     $\sigma_{q_{\max}} \leftarrow \sigma_q$.
12:   **end if**
13: **end while**
**output** $\{\boldsymbol{B}', \boldsymbol{A}', \boldsymbol{\Omega}\}$: Bayesianized Low-Rank Adapter.

---

## B  Broader Impact and Limitations

**Broader Impact.**  This research advances methods for making large language models more trustworthy and reliable through improved uncertainty estimation. While we focus on language models, the fundamental principles of our framework can enhance uncertainty quantification across the broader machine learning field. This wider applicability creates opportunities for improving model reliability and safety across diverse applications. To the best of our knowledge, there are no ethical or other concerns that need to be addressed.

**Limitations.** TFB is subject to several limitations. First, our approach relies on the availability of an anchor dataset for determining search range and stopping criteria. Although this dataset doesn't require supervision or prior use in LoRA training, its quality and representativeness could impact the effectiveness of uncertainty estimation. Second, by constraining the family of full-weight posteriors to low-rank isotropic Gaussian distributions, TFB may not capture more complex uncertainty patterns that could be present in the data. At first blush, this seems to imply a trade-off between computational efficiency and model expressiveness. However, in practice, the trade-off may not be necessary as TFB can often enjoy both computational efficiency and model expressiveness, getting the best of both worlds. Given TFB's proven effectiveness and inherent simplicity, we recommend implementing TFB as the initial approach when developing reliable LLMs with existing LoRA adapters. If TFB fails to meet specific requirements, practitioners can then consider alternative expensive Bayesian methods. Finally, while we have demonstrated the effectiveness of TFB in various settings, its performance in more complex generation tasks requires further investigation. Future work could explore extending the framework to handle more sophisticated language generation scenarios and broader applications.

## C  Related Work

**Bayesian Low-Rank Adaptation.** The Bayesian framework provides a powerful approach for capturing and estimating uncertainty by defining prior distributions and approximating posterior distributions over the parameter space [46, 21, 14, 56, 18]. However, modeling parameter distributions across the entire parameter space during fine-tuning introduces significant computational overhead [13, 69]. To address this challenge, recent research has explored combining Bayesian methods with Parameter-Efficient Fine-Tuning (PEFT) techniques to improve the efficiency of uncertainty estimation. Several notable approaches have emerged in this direction. Wang et al. [61] and Balabanov & Linander [4] demonstrate improved performance by training multiple LoRA modules and ensemble their predictions during inference. Taking a different approach, Yang et al. [67] applies a Kronecker-factorized Laplace approximation to fine-tuned LoRA parameters. More recently, BLoB [62] advances the field by simultaneously estimating both the mean and covariance of LLM parameters within a single fine-tuning stage, leading to substantial performance improvements. Our proposed training-free Bayesianization represents a significant departure from these existing methods. Unlike approaches that require re-training [14, 61, 4, 62] or rely on continued training and gradient estimation [67], our method achieves uncertainty estimation without any additional training steps, substantially improving the simplicity and efficiency for Bayesian learning of LLMs.

## D  Proof of Theorems

**Theorem 4.1** (**Equivalent Variational Distribution of the Full Weight $\boldsymbol{W}$ in** TFB). *With the pre-trained weight matrix $\boldsymbol{W}_0 \in \mathbb{R}^{m \times n}$, the low-rank weight update matrix $\{\boldsymbol{B}' \in \mathbb{R}^{m \times r}, \boldsymbol{A}' \in \mathbb{R}^{r \times n}\}$ transformed from the given matrices $\{\boldsymbol{B}, \boldsymbol{A}\}$ following Eqn. 4 and 5, suppose that the variational distribution of $\boldsymbol{A}'$ is Gaussian $q(\boldsymbol{A}'|\boldsymbol{\theta}) = \prod_{ij} \mathcal{N}(A_{ij}|M_{ij}, \Omega_{ij}^2)$, where $\boldsymbol{M} = [M_{ij} = A'_{ij}] \in \mathbb{R}^{r \times n}$ is its mean and $\boldsymbol{\Omega} = [\Omega_{ij}] \in \mathbb{R}^{r \times n}$ is the standard deviation calculated as in Eqn. 7. The equivalent variational distribution $q(\mathrm{vec}(\boldsymbol{W})|\sigma_q)$ defined on the full weight matrix $\boldsymbol{W}$ is*

$$
\begin{aligned}
q(\mathrm{vec}(\boldsymbol{W})|\sigma_q) &= \mathcal{N}(\mathrm{vec}(\boldsymbol{W})|\boldsymbol{\mu}_q, \boldsymbol{\Sigma}_q), \\
\text{where} \quad \boldsymbol{\mu}_q &= \mathrm{vec}(\boldsymbol{W}_0 + \boldsymbol{B}'\boldsymbol{M}), \\
\boldsymbol{\Sigma}_q &= \sigma_q^2 \boldsymbol{I}_n \otimes \begin{bmatrix} \boldsymbol{I}_r & \\ & \boldsymbol{0}_{m-r} \end{bmatrix}.
\end{aligned} \tag{14}
$$

*Proof.* We have the following lemma from BLoB that calculates the covariance matrix of a given low-rank Bayesianization scheme $\{\boldsymbol{B}, \boldsymbol{A}, \boldsymbol{\Omega}\}$ [62].

**Lemma D.1.** *With the pre-trained weight matrix $\boldsymbol{W}_0 \in \mathbb{R}^{m \times n}$ and the low-rank weight update matrix $\boldsymbol{B} \in \mathbb{R}^{m \times r}$, suppose that the variational distribution of the other low-rank update matrix $\boldsymbol{A} \in \mathbb{R}^{r \times n}$ is Gaussian with $q(\boldsymbol{A}|\boldsymbol{\theta} = \{\boldsymbol{M}, \boldsymbol{\Omega}\}) = \prod_{ij} \mathcal{N}(A_{ij}|M_{ij}, \Omega_{ij}^2)$, where $\boldsymbol{M} = [M_{ij}] \in \mathbb{R}^{r \times n}$ and $\boldsymbol{\Omega} = [\Omega_{ij}] \in \mathbb{R}^{r \times n}$ are its mean and standard deviation, respectively. The equivalent variational*

*distribution defined on the full weight matrix $\boldsymbol{W}$ is given by*

$$q(\mathrm{vec}(\boldsymbol{W})|\boldsymbol{B},\boldsymbol{\theta}) = \mathcal{N}(\mathrm{vec}(\boldsymbol{W})|\boldsymbol{\mu}_q, \boldsymbol{\Sigma}_q),$$
$$where \quad \boldsymbol{\mu}_q = \mathrm{vec}(\boldsymbol{W}_0 + \boldsymbol{B}\boldsymbol{M}), \tag{15}$$
$$\boldsymbol{\Sigma}_q = [\boldsymbol{I}_n \otimes \boldsymbol{B}][\mathrm{diag}(\mathrm{vec}(\boldsymbol{\Omega})^2)][\boldsymbol{I}_n \otimes \boldsymbol{B}^\top].$$

Based on the assumption outlined in Eqn. 4, 5, and 7, we have the following properties about $\boldsymbol{B}'$, $\boldsymbol{M}$, and $\boldsymbol{\Omega}$ of TFB:

$$\boldsymbol{B}' = \boldsymbol{U}\,\mathrm{diag}(\boldsymbol{d}), \tag{16}$$
$$\boldsymbol{\Omega} = [1/\boldsymbol{d}, \cdots, 1/\boldsymbol{d}], \tag{17}$$
$$where \ \boldsymbol{U}^\top \boldsymbol{U} = \boldsymbol{I}_r, \boldsymbol{U}\boldsymbol{U}^\top = \begin{bmatrix} \boldsymbol{I}_r & \\ & \boldsymbol{0}_{m-r} \end{bmatrix}. \tag{18}$$

It now can be easily shown that the covariance matrix of TFB is:

$$\boldsymbol{\Sigma}_q = [\boldsymbol{I}_n \otimes \boldsymbol{B}'][\mathrm{diag}(\mathrm{vec}(\boldsymbol{\Omega})^2)][\boldsymbol{I}_n \otimes \boldsymbol{B}'^\top] \tag{19}$$
$$= [\boldsymbol{I}_n \otimes \boldsymbol{B}'][\boldsymbol{I}_n \otimes \mathrm{diag}(1/\boldsymbol{d})^2][\boldsymbol{I}_n \otimes \boldsymbol{B}'^\top] \tag{20}$$
$$= \boldsymbol{I}_n \otimes [\boldsymbol{B}'\,\mathrm{diag}(\sigma_q/\boldsymbol{d})^2 \boldsymbol{B}'^\top] \tag{21}$$
$$= \boldsymbol{I}_n \otimes [\boldsymbol{U}\,\mathrm{diag}(\boldsymbol{d})\,\mathrm{diag}(\sigma_q/\boldsymbol{d})^2\,\mathrm{diag}(\boldsymbol{d})^\top \boldsymbol{U}^\top] \tag{22}$$
$$= \sigma_q^2 \boldsymbol{I}_n \otimes \begin{bmatrix} \boldsymbol{I}_r & \\ & \boldsymbol{0}_{m-r} \end{bmatrix}, \tag{23}$$

which proves that $q(\mathrm{vec}(\boldsymbol{W}))$ is a low-rank isotropic Gaussian distribution. $\qquad\square$

**Proposition D.1.1.** *The function $\mathrm{proj}(\cdot)$ defined in Eqn. 3 projects the full-dimensional isotropic Gaussian to the low-rank subspace of LoRA. It can be formulated as*

$$\mathrm{proj}(\sigma_q^2 \boldsymbol{I}_{mn}) = \boldsymbol{P}(\sigma_q^2 \boldsymbol{I}_{mn}), \tag{24}$$
$$where \quad \boldsymbol{P} = \boldsymbol{I}_n \otimes \begin{bmatrix} \boldsymbol{I}_r & \\ & \boldsymbol{0}_{m-r} \end{bmatrix}. \tag{25}$$

*Proof.* By Theorem 4.1, we have

$$\boldsymbol{P}\boldsymbol{I}_{mn} = \boldsymbol{I}_n \otimes \begin{bmatrix} \boldsymbol{I}_r & \\ & \boldsymbol{0}_{m-r} \end{bmatrix}. \tag{26}$$

Hence it is trivial to have $\boldsymbol{P} = \boldsymbol{I}_n \otimes \begin{bmatrix} \boldsymbol{I}_r & \\ & \boldsymbol{0}_{m-r} \end{bmatrix}$. $\qquad\square$

**Theorem 4.2** (TFB **as Generalized Variational Inference**). *Suppose the evaluation metric $l_\mathcal{D}(\sigma_q)$ defined following Assumption 4.1 is locally convex within the range of $\sigma_q \in [0, \epsilon_0)$. Suppose the approximate distribution of $\boldsymbol{W}$ given $\sigma_q$ is defined following Theorem 4.1. Suppose we have the prior distribution $P(\mathrm{vec}(\boldsymbol{W})) = \mathcal{N}(\mathrm{vec}(\boldsymbol{W})|\boldsymbol{\mu}_p, \boldsymbol{\Sigma}_p)$, where $\boldsymbol{\mu}_p = \boldsymbol{\mu}_q = \mathrm{vec}(\boldsymbol{W}_0 + \boldsymbol{B}'\boldsymbol{M})$, and $\boldsymbol{\Sigma}_p = \sigma_p^2 \boldsymbol{I}$ with $\sigma_p > \epsilon_0$. Then for $\forall \lambda > 0$, $\exists \widetilde{\epsilon}$, s.t. the following two optimization problems (i) Generalized Variational Inference [7, 22, 30]*

$$\min_{\sigma_q} \quad l_\mathcal{D}(\sigma_q) + \lambda\,\mathrm{KL}[q(\boldsymbol{W}|\sigma_q) \parallel P(\boldsymbol{W})], \tag{27}$$

*and (ii) Training-Free Bayesianization ( TFB )*

$$\max \quad \sigma_q$$
$$s.t. \quad l_\mathcal{D}(\sigma_q) \leq \widetilde{\epsilon}, \tag{28}$$

*are equivalent, i.e., the two optimization problems have the same optimal solution, where $\lambda$ is the regularization coefficient of the KL-divergence.*

*Proof.* First we prove the KL divergence term is convex w.r.t. $\sigma_q$. For two Gaussian distributions $q$ and $p$ whose covariance matrices $\boldsymbol{\Sigma}_q \in \mathbb{R}^{d\times d}$ and $\boldsymbol{\Sigma}_p \in \mathbb{R}^{d\times d}$ are both full-rank, with their means as $\boldsymbol{\mu}_q \in \mathbb{R}^d$ and $\boldsymbol{\mu}_p \in \mathbb{R}^d$, we have their KL-divergence as

$$\mathrm{KL}[q\|p] = \tfrac{1}{2}\left[\log\frac{|\boldsymbol{\Sigma}_p|}{|\boldsymbol{\Sigma}_q|} - d + \mathrm{tr}(\boldsymbol{\Sigma}_p^{-1}\boldsymbol{\Sigma}_q) + (\boldsymbol{\mu}_q - \boldsymbol{\mu}_p)^\top \boldsymbol{\Sigma}_p^{-1}(\boldsymbol{\mu}_q - \boldsymbol{\mu}_p)\right]. \tag{29}$$

For `TFB`, to avoid unbounded KL divergence, we project the original assumed Gaussian prior $P$ into the same low-rank sub-space of the posterior $q$. We summarize the prior and variational distribution of the posterior as follows:

$$q(\mathrm{vec}(\boldsymbol{W})|\sigma_q) = \mathcal{N}\left(\mathrm{vec}(\boldsymbol{W})|\boldsymbol{\mu}_q = \mathrm{vec}(\boldsymbol{W}_0 + \boldsymbol{B}'\boldsymbol{M}), \boldsymbol{\Sigma}_q = \sigma_q^2 \boldsymbol{I}_n \otimes \begin{bmatrix}\boldsymbol{I}_r & \\ & \boldsymbol{0}_{m-r}\end{bmatrix}\right),$$

$$P(\mathrm{vec}(\boldsymbol{W})|\sigma_p) = \mathcal{N}\left(\mathrm{vec}(\boldsymbol{W})|\boldsymbol{\mu}_p = \mathrm{vec}(\boldsymbol{W}_0 + \boldsymbol{B}'\boldsymbol{M}), \boldsymbol{\Sigma}_p = \sigma_p^2 \boldsymbol{I}_n \otimes \begin{bmatrix}\boldsymbol{I}_r & \\ & \boldsymbol{0}_{m-r}\end{bmatrix}\right). \tag{30}$$

Substituting Eqn. 30 back into Eqn. 29, we have

$$\mathrm{KL}[q(\mathrm{vec}(\boldsymbol{W})|\sigma_q)\|P(\mathrm{vec}(\boldsymbol{W}|\sigma_p))] = \tfrac{nr}{2}\left[\log(\sigma_p^2) - 1 + \left\{-\log(\sigma_q^2) + \tfrac{\sigma_q^2}{\sigma_p^2}\right\}\right], \tag{31}$$

which is convex w.r.t. $\sigma_q$ and the global minimum of KL is achieved when $\sigma_q = \sigma_p$.

With $\sigma_q \leq \epsilon_0$, the convexity of two terms (KL and $l_{\mathcal{D}}$) holds. Hence we show by the Karush–Kuhn–Tucker theorem [32, 29, 37] that, for any given $\lambda$ there exists $\widetilde{\epsilon}$ such that the following two optimization problems are equivalent:

1. Minimization of generalized variational inference in the **Lagrange-form optimization**

$$\min_{\sigma_q} \quad \mathrm{KL}[q(\mathrm{vec}(\boldsymbol{W})|\sigma_q) \;\|\; P(\mathrm{vec}(\boldsymbol{W})|\sigma_p)] + \tfrac{1}{\lambda}l_{\mathcal{D}}(\sigma_q); \tag{32}$$

2. The **constrained-form optimization** corresponding to Eqn. 32

$$\begin{aligned} \min \quad & \mathrm{KL}[q(\mathrm{vec}(\boldsymbol{W})|\sigma_q) \;\|\; P(\mathrm{vec}(\boldsymbol{W})|\sigma_p)] \\ s.t. \quad & l_{\mathcal{D}}(\sigma_q) \leq \widetilde{\epsilon}. \end{aligned} \tag{33}$$

Since the KL term is monotonically decreasing when $\sigma_q \in [0, \sigma_p)$, and due to the fact that $\sigma_p > \epsilon_0$, the optimization in Eqn. 33 is equivalent to our final Training-Free Bayesianization (TFB):

$$\begin{aligned} \max \quad & \sigma_q \\ s.t. \quad & l_{\mathcal{D}}(\sigma_q) \leq \widetilde{\epsilon}. \end{aligned} \tag{34}$$

$\square$

# E   Implementation Details

## E.1   Datasets

We provide details of the datasets used in this work, as shown in Table 7. The combined dataset consisting of the six commonsense reasoning tasks contains the label set of "[A, B, C, D, E, True, False]".

Table 7: Dataset Statistics. The size of the Anchor Set $\mathcal{D}$ is used in Table 1, 3 and 14.

|  | WG-S | ARC-C | ARC-E | WG-M | OBQA | BoolQ | Combined |
|---|---|---|---|---|---|---|---|
| Size of Label Space | 2 | 5 | 5 | 2 | 4 | 2 | 7 |
| Size of Training Set | 640 | 1,119 | 2,251 | 2,258 | 4,957 | 9,427 | 20,652 |
| Size of Anchor Set $\mathcal{D}$ | 500 (78%) | 500 (45%) | 500 (22%) | 500 (22%) | 500 (10%) | 500 (5%) | 500 (2%) |
| Size of Test Set | 1,267 | 299 | 570 | 1,267 | 500 | 3,270 | 7,173 |

## E.2 Evaluation Metrics

Negative Log-Likelihood (**NLL**) and Expected Calibration Error (**ECE** [45]) are key metrics for uncertainty estimation. For a model $P_{\boldsymbol{\theta}}$ and test dataset $\{\boldsymbol{x}_n, y_n\}_{n=1}^{N}$, **NLL** penalizes models that assign low probabilities to correct labels, and is defined as:

$$\text{NLL} = \tfrac{1}{N} \sum_{n=1}^{N} -\log P_{\boldsymbol{\theta}}(y_n). \tag{35}$$

**ECE** measures the alignment between model confidence and accuracy by binning predictions:

$$\text{ECE} = \sum_{m=1}^{M} \tfrac{|B_m|}{n} \left| \text{acc}(B_m) - \text{conf}(B_m) \right|, \tag{36}$$

where $\text{acc}(B_m) = 1/|B_m| \sum_{i \in B_m} \mathbb{1}(\widehat{y}_i = y_i)$ is the average accuracy and $\text{conf}(B_m) = 1/|B_m| \sum_{i \in B_m} P(\widehat{y}_i)$ is the average confidence in bin $B_m$. We use bin size $|B_m| = 15$ throughout this paper.

## E.3 Searched $\sigma_q$ of TFB

We report the searched $\sigma_q^*$ using Algorithm 1 in Table 8, where the reported values are the mean values of three random seeds.

Table 8: Searched $\sigma_q^*$ of TFB using Algorithm 1.

| Base Model | WG-S | ARC-C | ARC-E | WG-M | OBQA | BoolQ |
|---|---|---|---|---|---|---|
| MLE | 0.004500 | 0.003917 | 0.004500 | 0.004354 | 0.003771 | 0.004063 |
| MAP | 0.004500 | 0.003479 | 0.003188 | 0.004208 | 0.003917 | 0.005083 |
| BLoB-Mean | 0.005813 | 0.005229 | 0.005229 | 0.006250 | 0.006250 | 0.005958 |

## E.4 Bayesianization (Training)

**Shared Configuration.** We report the mean and standard deviation of all experimental results calculated over three random seeds. For all training processes in our experiments, we employ the AdamW optimizer. The learning rate follows a linear decay schedule with a warmup ratio of 0.06 and a maximum value of $2e-4$. The batch size is set to 4, and the maximum sentence length is limited to 300 tokens. The LoRA configuration includes LoRA $\alpha = 16$ and LoRA $r = 8$. PiSSA [41] follows the exact same configuration as the LoRA's. For VeRA [35], due to its characteristic of shared weights across different layers which enables higher-rank setting with the same memory efficiency, we set its rank to $r = 256$ and learning rate to $5e-3$ for the MLE training on the combined dataset.

**Baseline Configuration.** The baseline configuration mainly follows BLoB [62]. MLE follows the standard LoRA implementation. For MAP, we implement it with a weight decay rate of $1e-5$. MCD consists of an ensemble of 10 LoRAs with a dropout rate of $p = 0.1$. For ENS, we fine-tune 3 LoRAs independently and combine them by averaging their logits during evaluation. We implement LAP and apply it to the MAP checkpoints. For BBB and BLoB, we use the default settings from Bayesian-Torch library [36], applying Bayesianization only to the $\boldsymbol{A}$ matrix. During training, the number of samples is set to $K = 1$ for both BBB and BLoB. At test time, we use $N = 10$ samples, matching the configuration of TFB.

**TFB Configuration.** We randomly sample unlabeled training data points to construct the anchor dataset $\mathcal{D} = \{\boldsymbol{x}_i, \widehat{y}_i\}_{i \in [M]}$ where $\widehat{y}_i$ is the pseudo-label generated by the given LoRA adapter before Bayesianization; the anchor dataset size $M = 500$ is fixed for all the datasets. We use NLL as the metric $l$ and set the performance change tolerance $\epsilon$ to 0.3% of relative performance change for all the datasets. To determine the optimal $\sigma_q^*$, we perform a 5-step binary search with the initial range of $[0.001, 0.015]$ using Algorithm 1. Similar to the other baseline methods, the final results of TFB are reported as averages across three random seeds using $\sigma_q^*$.

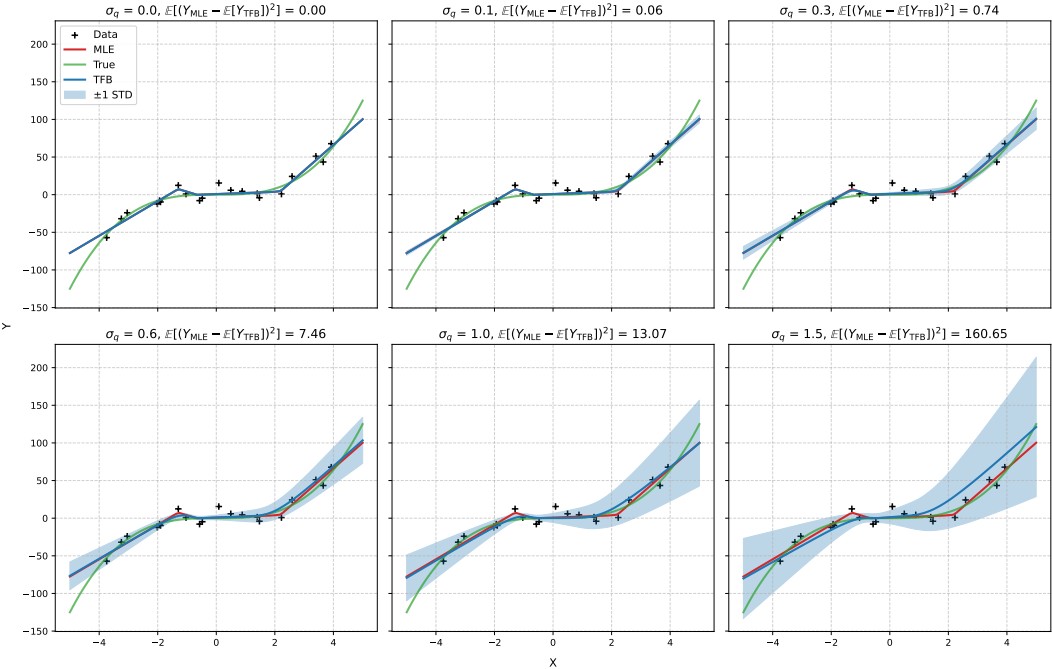

Figure 2: **Uncertainty estimation of** TFB **on a toy regression task.** The true cubic function $y = x^3$ (**GREEN**) and noisy training samples (**GRAY**) are shown alongside predictions from a deterministic MLP baseline (**RED**) and our TFB (**BLUE**). The blue shaded region represents ±1 standard deviation from TFB predictions with $\sigma_q = 1.0$. TFB effectively captures predictive uncertainty, showing low variance in data-dense regions ($x \in [0, 2]$) and increasing uncertainty for inputs outside the training distribution ($x \notin [-4, 4]$), with uncertainty proportional to distance from the training domain.

## F   Additional Experimental Results

We present additional experimental results in this section. Due to space constraints (and large table size), we defer several detailed tables to the end of this section rather than presenting them alongside the corresponding analyses.

### F.1   TFB as a General Bayesian Neural Network: A Visual Study

We demonstrate that TFB functions as a general Bayesian Neural Network (beyond its application to the LoRA adapters of LLMs) by evaluating its uncertainty estimation capabilities on a toy regression dataset, as illustrated in Fig. 2. We follow the regression task framework [21, 57] with the following specifications:

- **Input Features:** $\{\boldsymbol{x}_i \sim U[-4, 4]\}_{i \in [20]}$ uniformly sampled from the interval $[-4, 4]$.
- **True Function:** $y = x^3$ (plotted in **GREEN**).
- **Noisy Labels:** $\boldsymbol{y}_i = \boldsymbol{x}_i^3 + 9 \cdot \epsilon_i$ where $\epsilon_i \sim \mathcal{N}(0, 1)$, representing the true function with added Gaussian noise. These data points appear as **GRAY** crosses.
- **MLE Baseline:** A two-layer MLP with 16 hidden neurons fit to the sampled dataset, providing a deterministic baseline without uncertainty quantification. The MLE predictions are shown in **RED**.
- **MLE Training Configuration:** Adam optimizer [31] with learning rate 0.1, trained for 1000 steps until convergence.
- TFB **Implementation:** We Bayesianize the MLE baseline using TFB with full-rank noise, defining the variational distribution as $q(\boldsymbol{W}|\boldsymbol{W}_0, \sigma_q) = \prod_{ij} \mathcal{N}(W_{ij}|W_{0,ij}, \sigma_q^2)$. For inference, we draw $N = 10$ samples from the variational weight distribution and plot the average prediction as a **BLUE** curve. The shaded region represents ±1 standard deviation of the sampled predictions. We evaluate multiple settings of $\sigma_q \in [0.1, 0.3, 0.6, 1.0, 1.5]$ and report the average squared predictive difference between TFB and MLE in the figure.

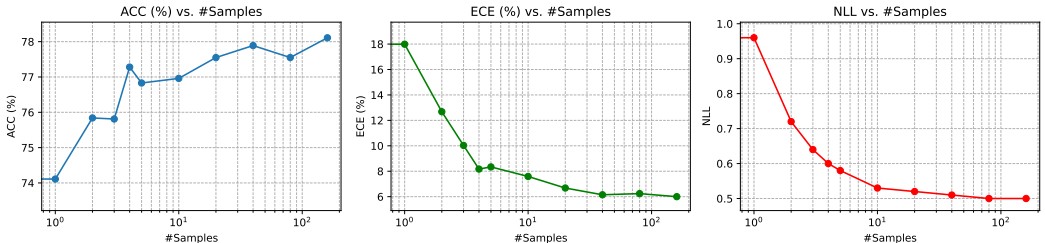

Figure 3: **Performance of test-time scaling for** TFB **with varying numbers of samples** $N$**.** The shaded region denotes the standard deviation across three random seeds. While larger $N$ generally yields better performance, our choice of $N = 10$ provides a favorable trade-off between accuracy and efficiency at test time.

As Fig. 2 demonstrates, with an appropriately calibrated standard deviation of the approximate posterior ($\sigma_q = 1.0$ in this case), TFB effectively captures predictive uncertainty: in regions with dense data sampling ($x \in [0, 2]$), TFB produces low predictive uncertainty; conversely, for inputs outside the training domain ($x \notin [-4, 4]$), the predictive uncertainty increases proportionally with distance from the training distribution.

### F.2 Test-Time Sample Size Analysis

Increasing the number of samples generally yields more accurate estimates of the expected output, thereby improving model performance in terms of ECE and NLL. **We ran additional experiments and report** TFB**'s performance for sample sizes ranging from** $N = 0$ **(reduced to BLoB-Mean) to** $N = 160$ **on the WG-S dataset [50] in Fig. 3.** Each experiment is repeated three times with different random seeds on Llama3.1-8B.

As shown Fig. 3, increasing the number of samples at test time generally improves the alignment between the Monte Carlo estimates and the expectations in theory, leading to better uncertainty estimation. Our choice of $N = 10$ samples in the paper balances performance and test-time efficiency: it achieves significant improvement in ECE and NLL while introducing acceptable computational overhead. The performance gap between this setting and extremely large $N$ is also mild.

### F.3 Anchor Dataset Size Analysis

**We present the performance of BLoB-Mean +** TFB **on the ARC-Easy dataset with varying anchor dataset sizes ranging from 100 to 2000 in Figure 4.** Initially, we hypothesized a negative correlation between the performance variance and the anchor dataset size. However, as shown in the figure, across experiments with three different random seeds, neither the performance variance nor the average performance exhibits a significant correlation with the anchor dataset size. This suggests that TFB is robust to the size of the anchor dataset.

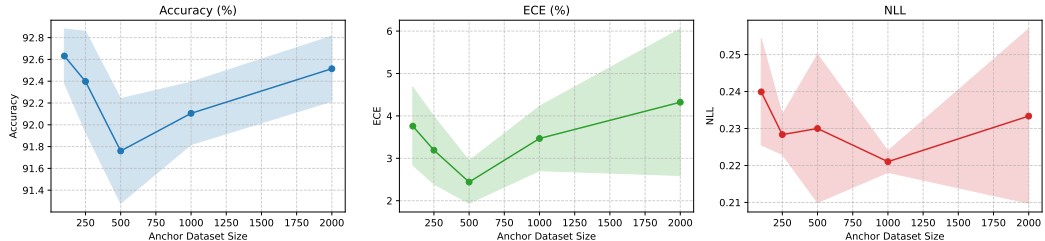

Figure 4: **Performance of BLoB-Mean +** TFB **with different size of anchor dataset on the ARC-Easy dataset.** The shaded area represents the standard deviation of results across three random seeds, indicating that TFB is not sensitive to the size of the anchor dataset.

### F.4 TFB Beyond Training Data as Anchor Dataset

In this section, we designate a portion of the unlabeled testing dataset as the anchor dataset, simulating a scenario where partial user input is accessible. The anchor dataset size remains fixed at 500 across all datasets. As illustrated in Table 10, the performance variation across different data sources for the anchor dataset is minimal, indicating that the choice of data source for the anchor dataset has negligible impact.

### F.5 TFB Beyond the NLL Metric

**We report the additional results of TFB when using Accuracy (ACC) as the evaluation metrics $l$ in Table 9.** In our implementation, we adopt the change ratio of classification results as an accuracy-based evaluation metric ($l_{ACC}$) for the unsupervised anchor dataset. Comparing the two evaluation metrics ($l_{ACC}$ vs $l_{NLL}$) in Table 9, we observe comparable performance across all datasets. In some cases, accuracy-based evaluation ($l_{ACC}$) even yields slightly better results. For instance, BLoB-Mean+TFB achieves lower ECE on several datasets when using $l_{ACC}$. However, we adopt NLL as the primary evaluation metric in Table 1 since it better aligns with our theoretical framework in Theorem 4.2.

### F.6 TFB Beyond the Low-Rank Isotropic Gaussians

In Sec. 5.4, we compare TFB with two alternative Gaussian distribution families that are controlled by a single parameter $\sigma_q$:

- Full-Rank Isotropic Gaussian (**FR**): given $\sigma_q$, the FR's variational distribution of the weight matrix $q(\text{vec}(\boldsymbol{W})) = \mathcal{N}(\text{vec}(\boldsymbol{W})|\boldsymbol{\mu}_q, \boldsymbol{\Sigma}_q)$ where $\boldsymbol{\mu}_q = \boldsymbol{W}_0 + \boldsymbol{BA}$ (same as TFB) and $\boldsymbol{\Sigma}_q = \sigma_q^2 \boldsymbol{I}_{mn}$ is full-rank.
- Constant Standard Deviation Matrix (**C-STD**): given $\sigma_q$, the C-STD's variational distribution of the weight matrix $q(\text{vec}(\boldsymbol{W})) = \mathcal{N}(\text{vec}(\boldsymbol{W})|\boldsymbol{\mu}_q, \boldsymbol{\Sigma}_q)$ where $\boldsymbol{\mu}_q = \boldsymbol{W}_0 + \boldsymbol{BA}$ (same as TFB) and $\boldsymbol{\Sigma}_q = \sigma_q^2 \boldsymbol{I}_n \otimes [\boldsymbol{BB}^\top]$.

C-STD's covariance matrix $\boldsymbol{\Sigma}_q$ is derived through Lemma D.1:

$$\boldsymbol{\Sigma}_q = [\boldsymbol{I}_n \otimes \boldsymbol{B}][\text{diag}(\text{vec}(\boldsymbol{\Omega})^2)][\boldsymbol{I}_n \otimes \boldsymbol{B}^\top] \tag{37}$$

$$= [\boldsymbol{I}_n \otimes \boldsymbol{B}][\sigma_q^2 \boldsymbol{I}_{rn}][\boldsymbol{I}_n \otimes \boldsymbol{B}^\top] \tag{38}$$

$$= \sigma_q^2 [\boldsymbol{I}_n \otimes \boldsymbol{B}][\boldsymbol{I}_n \otimes \boldsymbol{B}^\top] \tag{39}$$

$$= \sigma_q^2 \boldsymbol{I}_n \otimes [\boldsymbol{BB}^\top]. \tag{40}$$

This depends on $\boldsymbol{B}$ and thus varies for equivalent LoRA parameterizations $\boldsymbol{B}, \boldsymbol{A}$ of the same $\Delta\boldsymbol{W}$.

**We report the additional results comparing TFB with other approximate families Gaussians (FR and C-STD as discussed in Sec. 5.4) when using Accuracy as the evaluation metrics $l$ in Table 11.** When the evaluation metric is set to Accuracy, the advantage of TFB becomes more significant compared to the results shown in Table 3. TFB with low-rank isotropic Gaussian as the variational distribution demonstrates superior calibration performance compared to both **FR** and **C-STD** variants while maintaining competitive accuracy. For ECE, TFB achieves better results across most datasets, with notable improvements on in-distribution tasks: 8.78% on WG-S (vs. 12.06% for **FR** and 11.61% for **C-STD**) and 1.28% on BoolQ (vs. 3.26% for **FR** and 2.65% for **C-STD**). Similarly for NLL, TFB consistently outperforms or matches the baseline variants, particularly on WG-S (0.55 vs. 0.63 for **FR** and 0.61 for **C-STD**) while preserving comparable accuracy scores. These results suggest that TFB's approach to variance modeling is more effective than both full-rank isotropic and constant standard deviation alternatives.

### F.7 TFB Beyond the Llama3.1-8B Backbone

**We report the detailed performance of TFB applied to various LLM backbones in Table 12.** While the baseline MLE is typically trained for 2 epochs (shown with each backbone name), we also report results with reduced training (1 epoch) for comparison. Although training with fewer steps (early stopping) can effectively reduce model overconfidence, it typically leads to performance degradation.

The results demonstrate that TFB consistently improves model calibration across different backbones while maintaining competitive accuracy. Specifically, for Llama2-7B, TFB reduces the ECE from 4.50% to 1.24% on the combined dataset while preserving the accuracy (81.32% vs 81.41%). Similar improvements are observed with Llama3-8B, Llama3.1-8B, and Mistral-7B-v0.3, where TFB achieves better calibration than both the full training and early stopping baselines without sacrificing performance, suggesting its effectiveness as a general approach for enhancing LLM calibration.

### F.8 TFB Beyond the Naive LoRA

**We report the detailed performance of TFB applied to various LoRA variants in Table 13.** The baseline models are trained for 2 epochs using pre-trained Llama3.1-8B on the concatenated dataset of six commonsense reasoning tasks. Specifically, we consider the two LoRA variants:

- VeRA [35]: Uses shared low-rank matrices $B$ and $A$ across layers, with layer-specific trainable scalar vector $d$ and bias vector $b$. Concretely, the parameterization of VeRA's updated weight matrix $W$ is modeled as:

$$W = W_0 + \Delta W = W_0 + [\text{diag}(b)]B[\text{diag}(d)]A. \qquad (41)$$

  Hence after the fine-tuning of VeRA, we can easily regroup the weight matrices into $\{B' = [\text{diag}(b)]B[\text{diag}(d)], A' = A\}$, and apply the TFB Bayesianization scheme illustrated in Algorithm 1.

- PiSSA [41]: Employs an alternative initialization scheme while maintaining LoRA's parameterization and training procedure. Hence the TFB process for PiSSA is trivial.

The results in Table 13 show that TFB consistently improves calibration across different LoRA variants while preserving model performance. Notably, when applied to the standard LoRA, TFB significantly reduces the ECE from 4.74% to 1.05% on the combined dataset with minimal impact on accuracy (86.45% vs 86.70%). Similar improvements are observed with VeRA and PiSSA variants, where TFB achieves better calibration (reducing ECE to 1.44% and 1.17% respectively) while maintaining comparable accuracy levels. These results demonstrate that TFB can effectively enhance model calibration across different LoRA architectures without compromising their performance.

### F.9 Improving the Inference-Time Efficiency of TFB

**We report the detailed performance of last-layer TFB (LL TFB) in Table 14**. As indicated in the table, with only $N = 10$ samples, last-layer Bayesianization provides a less effective uncertainty estimation compared to full-model Bayesianization. However, increasing the number of samples to $N = 100$ significantly enhances the posterior estimation, allowing last-layer Bayesianization to achieve better accuracy. This improvement further allows it to outperform the full-model Bayesianization in terms of NLL across most datasets.

### F.10 Additional Results on Llama2-7B

**We report the detailed performance of TFB applied to the Llama2-7B pre-trained weights in Table 15.** The performance change tolerance $\epsilon$ is set adaptively to either 1% or 0.5%, depending on the checkpoint's overfitting characteristics. To determine the optimal $\sigma_q^*$, we conduct parallel experiments with eight values of $\sigma_q \in [0.01, 0.015, 0.02, 0.025, 0.03, 0.035, 0.04, 0.05]$ using a single random seed. We construct an approximate function $\widehat{\sigma}_q(p)$ through piecewise linear interpolation of the observed performance and estimate $\sigma_q^* \approx \widehat{\sigma}_q(p_0 - \epsilon)$. Similar to other baseline methods, the final results of TFB are reported as averages across three random seeds using $\sigma_q^*$.

**In-Distribution (IND) Results.** We observed several key patterns from the IND Datasets results. For example, the MLE baseline shows relatively strong accuracy but suffers from high ECE values (e.g., 29.83% on WG-S), indicating significant overconfidence. This aligns with the common challenge of LLM overconfidence during conventional fine-tuning.

TFB applied to BLoB-Mean demonstrates strong overall performance across the IND datasets, achieving the highest accuracy on several datasets (69.94% on WG-S, 70.72% on ARC-C, and 86.74% on ARC-E). More importantly, it achieves this while maintaining lower ECE values compared to methods like MCD and ENS, suggesting better calibrated predictions. The method also shows strong

NLL performance, with values consistently among the lowest across datasets (0.62 for WG-S, 0.86 for ARC-C).

In summary, TFB consistently enhances the performance of baseline methods (MLE, MAP, and BLoB-Mean) across different evaluation scenarios, with notable improvements in both accuracy and calibration metrics. The improvements are particularly evident in the significant ECE reductions (e.g., from 29.83% to 16.26% for MLE on WG-S) while maintaining or improving accuracy, with the most substantial gains observed when TFB is combined with BLoB-Mean, achieving both the highest accuracy and lowest ECE values across most datasets.

**Out-of-Distribution (OOD) Results.**   The OOD evaluation reveals interesting patterns across both smaller and larger distribution shifts. For smaller shifts (ARC-C and ARC-E), BLoB-Mean with TFB maintains strong performance, achieving 70.38% and 80.16% accuracy respectively, while keeping ECE values low (12.28% and 8.07%). This suggests robust generalization under moderate distribution shifts.

For larger shifts (Chem and Phy datasets), we see a more significant performance degradation across all methods, as expected. However, BLoB-Mean with TFB still maintains competitive performance, achieving 42.67% accuracy on Chem and 30.67% on Phy, while maintaining reasonable calibration metrics. The method's NLL values (1.35 and 1.46 respectively) remain competitive with other approaches, indicating relatively well-calibrated uncertainty estimates even under substantial distribution shifts.

Notable is the consistently strong performance of the BLoB variants (both w/ and w/o TFB) across different metrics and datasets, suggesting that this approach offers a robust framework for both in-distribution and out-of-distribution scenarios. The results demonstrate that the method successfully balances the trade-off between accuracy and calibration, particularly evident in the out-of-distribution scenarios where maintaining both aspects becomes more challenging.

Table 9: **Performance of different methods applied to LoRA on Llama3.1-8B pre-trained weights,** where Accuracy (**ACC**) and Expected Calibration Error (**ECE**) are reported in percentages. **"TF?"** denotes whether a method is **T**raining-**F**ree. The evaluation is done across six common-sense reasoning tasks with a shared hyper-parameter setting after fine-tuning of 5 epochs . We sample $N = 10$ during inference in all sampling-based methods including **BLoB** [62] and TFB. Rows with shading indicate training-free Bayesianization methods that use a pre-trained LoRA as their mean. For TFB, we randomly sample a subset of the training data without labels as the anchor dataset $\mathcal{D}$. For accuracy-based evaluation ($l_{\mathrm{ACC}}$), we set the performance drop tolerance to $\epsilon = 1\%$. For NLL loss ($l_{\mathrm{NLL}}$), we use the same settings as in Table 1. "↑" and "↓" indicate that higher and lower values are preferred, respectively. **Boldface** and underlining denote the best and the second-best performance, respectively.

| Metric | Method | TF? | In-Distribution Datasets | | | | | | Out-of-Distribution Datasets (OBQA→X) | | | |
| | | | | | | | | | Small Shift | | Large Shift | |
| | | | WG-S | ARC-C | ARC-E | WG-M | OBQA | BoolQ | ARC-C | ARC-E | Chem | Phy |
| ACC (↑) | MCD | ✗ | 78.03±0.61 | 81.64±1.79 | 91.37±0.38 | 83.18±0.84 | 87.20±1.02 | 89.93±0.16 | 81.42±1.38 | 87.27±0.84 | 47.92±2.25 | **46.53±0.49** |
| | ENS | ✗ | **78.82±0.52** | 82.55±0.42 | **91.84±0.36** | **83.99±0.74** | 87.37±0.67 | **90.50±0.14** | 79.62±0.57 | 86.56±0.60 | **49.65±3.22** | 44.44±1.96 |
| | LAP | BP | 76.05±0.92 | 79.95±0.42 | 90.73±0.08 | 82.83±0.85 | 87.90±0.20 | 89.36±0.52 | 81.08±1.20 | 87.21±1.20 | 48.26±3.93 | 46.18±1.30 |
| | MLE | - | 77.87±0.54 | 81.08±0.48 | 91.67±0.36 | 82.30±0.53 | 87.90±0.87 | 89.58±0.26 | 81.48±2.41 | 86.83±0.87 | 45.83±0.85 | 42.36±1.77 |
| | + TFB ($l_{\mathrm{ACC}}$) | ✓ | 76.40±0.13 | 82.00±0.33 | 91.39±0.34 | 82.37±0.59 | 88.07±0.52 | 89.66±0.19 | 10.66±1.03 | 6.44±1.14 | 23.59±1.74 | 26.90±4.47 |
| | + TFB ($l_{\mathrm{NLL}}$) | ✓ | 77.44±0.30 | 82.53±1.00 | 91.33±0.37 | 82.53±0.28 | **88.53±0.57** | 89.75±0.25 | 79.76±1.24 | 85.52±0.56 | 44.33±4.03 | 37.00±2.16 |
| | MAP | - | 76.90±0.97 | 81.08±2.48 | 91.61±0.44 | 82.59±0.28 | 85.73±0.19 | 90.09±0.28 | 79.98±0.87 | 86.58±0.79 | 43.40±4.98 | 38.54±3.40 |
| | + TFB ($l_{\mathrm{ACC}}$) | ✓ | 76.00±1.34 | 82.53±1.80 | 91.39±0.09 | 82.19±0.89 | 86.13±0.34 | 89.84±0.15 | 79.73±1.86 | 86.18±1.04 | 42.67±3.30 | 36.00±3.56 |
| | + TFB ($l_{\mathrm{NLL}}$) | ✓ | 76.43±0.72 | 82.80±1.42 | 91.39±0.37 | 82.64±0.58 | 86.00±0.16 | 89.96±0.18 | 80.61±1.24 | 86.30±0.89 | 45.33±2.87 | 35.67±4.11 |
| | BLoB | ✗ | 76.45±0.37 | 82.32±1.15 | 91.14±0.54 | 82.01±0.56 | 87.57±0.21 | 89.65±0.15 | 79.75±0.43 | 87.13±0.00 | 42.71±3.71 | 44.79±6.64 |
| | BLoB-Mean | ✗ | 77.72±0.12 | 82.60±0.60 | 91.64±0.55 | 83.92±0.48 | 88.00±0.80 | 89.86±0.05 | 82.06±1.15 | **88.54±0.31** | 39.93±5.20 | 39.93±4.02 |
| | + TFB ($l_{\mathrm{ACC}}$) | ✓ | 75.28±0.33 | 82.80±0.33 | 91.64±0.15 | 81.84±0.74 | 88.00±0.16 | 89.60±0.35 | **82.93±1.91** | 86.97±0.62 | 36.00±5.66 | 36.00±5.72 |
| | + TFB ($l_{\mathrm{NLL}}$) | ✓ | 77.81±0.36 | **83.33±0.19** | 91.76±0.48 | 83.81±0.39 | 87.80±0.16 | 90.11±0.28 | **82.93±1.54** | 87.64±0.51 | 39.67±7.32 | 37.33±6.65 |
| ECE (↓) | MCD | ✗ | 16.13±0.54 | 13.69±1.11 | 6.73±0.71 | 13.05±0.99 | 9.76±0.71 | 7.95±0.17 | 13.63±1.18 | 9.27±0.60 | 30.91±3.57 | 33.08±1.40 |
| | ENS | ✗ | 14.72±0.17 | 13.45±1.19 | 6.59±0.45 | 11.17±0.92 | 8.17±0.86 | 7.35±0.55 | 11.37±1.82 | 7.21±1.13 | 18.92±6.03 | 26.80±3.23 |
| | LAP | BP | 4.18±0.11 | 9.26±3.08 | 5.27±0.51 | **3.50±0.78** | 8.93±0.34 | **1.93±0.22** | 7.83±1.49 | 7.80±1.99 | **14.49±0.57** | **13.17±2.14** |
| | MLE | - | 17.02±0.46 | 16.35±0.68 | 7.00±0.53 | 13.83±0.65 | 9.77±0.81 | 8.69±0.21 | 14.45±2.19 | 10.78±0.50 | 32.46±2.60 | 38.41±4.44 |
| | + TFB ($l_{\mathrm{ACC}}$) | ✓ | 8.77±0.64 | 9.97±0.29 | 4.32±0.42 | 6.10±0.46 | 5.96±0.76 | 5.96±0.23 | 10.66±1.03 | 6.44±1.14 | 23.59±1.74 | 26.90±4.47 |
| | + TFB ($l_{\mathrm{NLL}}$) | ✓ | 12.98±0.37 | 11.63±0.68 | 5.14±0.14 | 10.01±0.70 | 7.20±0.47 | 7.39±0.26 | 6.54±0.53 | 5.69±1.64 | 14.63±1.46 | 19.68±3.27 |
| | MAP | - | 18.71±0.74 | 15.77±1.60 | 6.62±0.64 | 14.26±0.92 | 12.19±0.55 | 8.40±0.25 | 16.46±0.44 | 11.36±0.58 | 34.79±3.76 | 38.50±2.18 |
| | + TFB ($l_{\mathrm{ACC}}$) | ✓ | 11.84±0.98 | 8.61±2.53 | 5.19±0.43 | 7.51±1.92 | 8.72±0.75 | 5.55±0.64 | 12.19±1.63 | 8.08±1.24 | 27.76±3.44 | 31.91±3.68 |
| | + TFB ($l_{\mathrm{NLL}}$) | ✓ | 14.95±0.65 | 11.27±2.53 | 5.76±0.63 | 10.97±1.19 | 9.70±0.69 | 6.86±0.31 | 13.25±0.95 | 9.22±0.91 | 27.21±2.62 | 35.91±4.12 |
| | BLoB | ✗ | 9.93±0.22 | **5.41±1.17** | 2.70±0.87 | 4.28±0.64 | 2.91±0.92 | 2.58±0.25 | **5.61±0.40** | **2.48±0.43** | 16.67±0.87 | **12.78±4.18** |
| | BLoB-Mean | ✗ | 15.43±0.15 | 12.41±1.52 | 4.91±0.28 | 9.37±1.33 | 6.44±0.15 | 6.26±0.29 | 11.22±0.36 | 6.34±0.71 | 26.65±3.06 | 25.40±5.40 |
| | + TFB ($l_{\mathrm{ACC}}$) | ✓ | **3.04±0.12** | 6.76±1.47 | 4.81±1.18 | 7.42±1.24 | 5.26±0.71 | 3.22±0.36 | 6.55±1.04 | 5.54±1.33 | 17.00±4.71 | 16.65±4.33 |
| | + TFB ($l_{\mathrm{NLL}}$) | ✓ | 8.16±0.48 | 6.48±0.36 | **2.44±0.50** | 3.83±0.43 | 2.67±0.18 | 3.10±0.59 | 6.69±1.63 | 3.61±0.87 | 18.45±6.75 | 20.53±6.27 |
| NLL (↓) | MCD | ✗ | 0.83±0.01 | 0.99±0.10 | 0.45±0.06 | 0.64±0.03 | 0.62±0.08 | 0.49±0.01 | 1.03±0.02 | 0.61±0.03 | 1.91±0.18 | 2.02±0.15 |
| | ENS | ✗ | 0.75±0.01 | 0.80±0.11 | 0.38±0.03 | 0.55±0.02 | 0.45±0.05 | 0.42±0.05 | 0.72±0.07 | 0.44±0.04 | 1.40±0.18 | 1.50±0.13 |
| | LAP | BP | 0.56±0.00 | 1.18±0.02 | 1.04±0.01 | 0.51±0.00 | 0.94±0.00 | 0.43±0.00 | 1.17±0.01 | 1.11±0.00 | **1.27±0.01** | **1.28±0.00** |
| | MLE | - | 0.88±0.04 | 1.20±0.11 | 0.46±0.04 | 0.68±0.01 | 0.61±0.06 | 0.52±0.01 | 1.07±0.06 | 0.72±0.06 | 1.91±0.16 | 2.25±0.21 |
| | + TFB ($l_{\mathrm{ACC}}$) | ✓ | 0.58±0.02 | 0.73±0.02 | 0.29±0.02 | 0.46±0.01 | 0.41±0.03 | 0.36±0.00 | 0.79±0.04 | 0.49±0.05 | 1.52±0.09 | 1.82±0.05 |
| | + TFB ($l_{\mathrm{NLL}}$) | ✓ | 0.68±0.03 | 0.85±0.02 | 0.33±0.03 | 0.53±0.01 | 0.46±0.04 | 0.42±0.00 | 0.66±0.02 | 0.44±0.01 | 1.39±0.11 | 1.49±0.05 |
| | MAP | - | 0.99±0.07 | 1.12±0.23 | 0.46±0.03 | 0.74±0.07 | 0.79±0.02 | 0.52±0.01 | 1.19±0.04 | 0.83±0.06 | 1.97±0.13 | 2.32±0.10 |
| | + TFB ($l_{\mathrm{ACC}}$) | ✓ | 0.65±0.02 | 0.65±0.08 | 0.35±0.03 | 0.48±0.04 | 0.54±0.04 | 0.35±0.02 | 0.88±0.03 | 0.60±0.05 | 1.58±0.12 | 1.95±0.13 |
| | + TFB ($l_{\mathrm{NLL}}$) | ✓ | 0.77±0.06 | 0.80±0.15 | 0.38±0.04 | 0.57±0.05 | 0.61±0.03 | 0.40±0.01 | 0.96±0.08 | 0.66±0.06 | 1.69±0.16 | 2.12±0.08 |
| | BLoB | ✗ | 0.58±0.00 | **0.51±0.03** | 0.23±0.01 | 0.43±0.01 | 0.34±0.01 | **0.26±0.01** | 0.56±0.02 | 0.35±0.02 | 1.34±0.04 | 1.35±0.10 |
| | BLoB-Mean | ✗ | 0.74±0.02 | 0.73±0.04 | 0.29±0.03 | 0.47±0.03 | 0.37±0.02 | 0.32±0.02 | 0.67±0.07 | 0.39±0.03 | 1.53±0.13 | 1.54±0.15 |
| | + TFB ($l_{\mathrm{ACC}}$) | ✓ | **0.52±0.01** | 0.52±0.03 | 0.24±0.01 | 0.45±0.01 | 0.35±0.01 | 0.28±0.00 | 0.55±0.02 | 0.37±0.01 | 1.38±0.10 | 1.41±0.09 |
| | + TFB ($l_{\mathrm{NLL}}$) | ✓ | 0.55±0.01 | 0.53±0.04 | **0.23±0.02** | 0.40±0.01 | 0.33±0.02 | 0.27±0.01 | **0.52±0.05** | 0.35±0.02 | 1.36±0.13 | 1.46±0.11 |

Table 10: **Performance of different methods applied to LoRA on Llama3.1-8B pre-trained weights,** where Accuracy (**ACC**) and Expected Calibration Error (**ECE**) are reported in percentages. ($\mathcal{D}_{\text{train}}$) and ($\mathcal{D}_{\text{test}}$) denote the anchor dataset is randomly sampled from the training dataset and the testing dataset, respectively. **"TF?"** denotes whether a method is **T**raining-**F**ree. The evaluation is done across six common-sense reasoning tasks with a shared hyper-parameter setting after fine-tuning of 5 epochs . We sample $N = 10$ during inference in all sampling-based methods including **BLoB** [62] and TFB. Rows with shading indicate training-free Bayesianization methods that use a pre-trained LoRA as their mean. "↑" and "↓" indicate that higher and lower values are preferred, respectively. **Boldface** and underlining denote the best and the second-best performance, respectively.

| Metric | Method | TF? | In-Distribution Datasets | | | | | | Out-of-Distribution Datasets (OBQA→X) | | | |
| | | | | | | | | | Small Shift | | Large Shift | |
| | | | WG-S | ARC-C | ARC-E | WG-M | OBQA | BoolQ | ARC-C | ARC-E | Chem | Phy |
|---|---|---|---|---|---|---|---|---|---|---|---|---|
| ACC (↑) | MCD | ✗ | 78.03±0.61 | 81.64±1.79 | 91.37±0.38 | 83.18±0.84 | 87.20±1.02 | 89.93±0.16 | 81.42±1.38 | 87.27±0.84 | 47.92±2.25 | **46.53±0.49** |
| | ENS | ✗ | **78.82±0.52** | 82.55±0.42 | 91.84±0.36 | **83.99±0.74** | 87.37±0.67 | **90.50±0.14** | 79.62±0.57 | 86.56±0.60 | **49.65±3.22** | 44.44±1.96 |
| | LAP | BP | 76.05±0.92 | 79.95±0.42 | 90.73±0.08 | 82.83±0.85 | 87.90±0.20 | 89.36±0.52 | 81.08±1.20 | 87.21±1.20 | 48.26±3.93 | 46.18±1.30 |
| | MLE | - | 77.87±0.54 | 81.08±0.48 | 91.67±0.36 | 82.30±0.53 | 87.90±0.87 | 89.58±0.26 | 81.48±2.41 | 86.83±0.87 | 45.83±0.85 | 42.36±1.77 |
| | + TFB ($\mathcal{D}_{\text{train}}$) | ✓ | 77.44±0.30 | 82.53±1.00 | 91.33±0.37 | 82.53±0.56 | **88.53±0.57** | 89.75±0.25 | 79.76±1.24 | 85.52±0.56 | 44.33±4.03 | 37.00±2.16 |
| | + TFB ($\mathcal{D}_{\text{test}}$) | ✓ | 77.07±0.48 | 82.27±0.68 | 91.33±0.45 | 82.51±0.86 | 88.20±0.16 | 89.60±0.12 | 81.82±1.48 | 86.67±0.73 | 47.33±1.70 | 40.33±2.62 |
| | MAP | - | 76.90±0.97 | 81.08±2.48 | 91.61±0.44 | 82.59±0.28 | 85.73±0.19 | 90.09±0.28 | 79.98±0.87 | 86.58±0.79 | 43.40±4.98 | 38.54±3.40 |
| | + TFB ($\mathcal{D}_{\text{train}}$) | ✓ | 76.43±0.72 | 82.80±1.42 | 91.39±0.37 | 82.64±0.58 | 86.00±0.16 | 89.96±0.18 | 80.61±1.24 | 86.30±0.89 | 45.33±2.87 | 35.67±4.11 |
| | + TFB ($\mathcal{D}_{\text{test}}$) | ✓ | 76.21±0.88 | 82.40±1.70 | 91.64±0.74 | 82.29±0.85 | 86.07±0.19 | 89.99±0.13 | 80.85±1.12 | 86.30±0.89 | 44.67±1.70 | 36.00±4.32 |
| | BLoB | ✗ | 76.45±0.37 | 82.32±1.15 | 91.14±0.54 | 82.01±0.56 | 87.57±0.21 | 89.65±0.15 | 79.75±0.43 | 87.13±0.00 | 42.71±3.71 | 44.79±6.64 |
| | BLoB-Mean | ✗ | 77.72±0.12 | 82.60±0.60 | 91.64±0.55 | 83.92±0.48 | 88.00±0.80 | 89.86±0.05 | 82.06±1.15 | **88.54±0.31** | 39.93±5.20 | 39.93±4.02 |
| | + TFB ($\mathcal{D}_{\text{train}}$) | ✓ | 77.81±0.36 | **83.33±0.19** | 91.76±0.48 | 83.81±0.39 | 87.80±0.16 | **90.11±0.28** | **82.93±1.54** | 87.64±0.51 | 39.67±7.32 | 37.33±6.65 |
| | + TFB ($\mathcal{D}_{\text{test}}$) | ✓ | 77.65±0.65 | 83.20±1.18 | **91.94±0.09** | 83.41±0.62 | 87.73±0.38 | 89.71±0.06 | 82.93±1.47 | 87.15±0.34 | 37.67±4.50 | 37.00±6.48 |
| ECE (↓) | MCD | ✗ | 16.13±0.54 | 13.69±1.11 | 6.73±0.71 | 13.05±0.99 | 9.76±0.71 | 7.95±0.17 | 13.63±1.18 | 9.27±0.60 | 30.91±3.57 | 33.08±1.40 |
| | ENS | ✗ | 14.72±0.17 | 13.45±1.19 | 6.59±0.45 | 11.17±0.92 | 8.17±0.86 | 7.35±0.55 | 11.37±1.82 | 7.21±1.13 | 18.92±6.03 | 26.80±3.23 |
| | LAP | BP | 4.18±0.11 | 9.26±3.08 | 5.27±0.51 | 3.50±0.78 | 8.93±0.34 | 1.93±0.22 | 7.83±1.49 | 7.80±1.99 | **14.49±0.57** | 13.17±2.14 |
| | MLE | - | 17.02±0.46 | 16.35±0.68 | 7.00±0.53 | 13.83±0.65 | 9.77±0.81 | 8.69±0.21 | 14.45±2.19 | 10.78±0.50 | 32.46±2.60 | 38.41±4.44 |
| | + TFB ($\mathcal{D}_{\text{train}}$) | ✓ | 12.98±0.37 | 11.63±0.68 | 5.14±0.14 | 10.01±0.70 | 7.20±0.47 | 7.39±0.26 | 6.54±0.53 | 5.69±1.64 | 14.63±1.46 | 19.68±3.27 |
| | + TFB ($\mathcal{D}_{\text{test}}$) | ✓ | 11.22±1.15 | 13.85±0.66 | 5.62±0.86 | 9.20±1.41 | 7.40±0.60 | 6.05±0.44 | 13.41±0.72 | 8.98±0.51 | 29.72±3.01 | 36.22±3.50 |
| | MAP | - | 18.71±0.74 | 15.77±1.60 | 6.62±0.64 | 14.26±0.92 | 12.19±0.55 | 8.40±0.25 | 16.46±0.44 | 11.36±0.58 | 34.79±3.76 | 38.50±2.18 |
| | + TFB ($\mathcal{D}_{\text{train}}$) | ✓ | 14.95±0.65 | 11.27±2.53 | 5.76±0.63 | 10.97±1.19 | 9.70±0.69 | 6.86±0.31 | 13.25±0.95 | 9.22±0.91 | 27.21±2.62 | 35.91±4.12 |
| | + TFB ($\mathcal{D}_{\text{test}}$) | ✓ | 13.08±1.55 | 12.85±2.09 | 5.03±0.44 | 9.28±1.13 | 9.86±0.94 | 6.44±0.37 | 11.07±1.54 | 8.02±1.00 | 24.51±3.76 | 33.25±5.63 |
| | BLoB | ✗ | 9.93±0.22 | **5.41±1.17** | 2.70±0.87 | 4.28±0.64 | 2.91±0.92 | 2.58±0.25 | 5.61±0.40 | **2.48±0.43** | 16.67±0.87 | **12.78±4.18** |
| | BLoB-Mean | ✗ | 15.43±0.15 | 12.41±1.52 | 4.91±0.28 | 9.37±1.33 | 6.44±0.15 | 6.26±0.29 | 11.22±0.38 | 6.34±0.71 | 26.65±3.06 | 25.40±5.40 |
| | + TFB ($\mathcal{D}_{\text{train}}$) | ✓ | 8.16±0.48 | 6.48±0.36 | 2.44±0.50 | 3.83±0.43 | 2.67±0.18 | 3.10±0.59 | 6.69±1.63 | 3.61±0.87 | 18.45±6.75 | 20.53±6.27 |
| | + TFB ($\mathcal{D}_{\text{test}}$) | ✓ | 7.77±0.77 | 6.92±2.27 | 2.98±0.67 | **2.94±0.27** | 3.69±0.57 | **1.48±0.35** | **4.91±1.21** | 3.56±0.47 | 17.15±3.28 | 16.81±5.33 |
| NLL (↓) | MCD | ✗ | 0.83±0.01 | 0.99±0.10 | 0.45±0.06 | 0.64±0.03 | 0.62±0.08 | 0.49±0.01 | 1.03±0.02 | 0.61±0.03 | 1.91±0.18 | 2.02±0.15 |
| | ENS | ✗ | 0.75±0.02 | 0.80±0.11 | 0.38±0.03 | 0.55±0.02 | 0.45±0.05 | 0.42±0.05 | 0.72±0.07 | 0.44±0.03 | 1.40±0.18 | 1.50±0.13 |
| | LAP | BP | 0.56±0.00 | 1.18±0.02 | 1.04±0.01 | 0.51±0.00 | 0.94±0.00 | 0.43±0.00 | 1.17±0.01 | 1.11±0.00 | **1.27±0.01** | **1.28±0.00** |
| | MLE | - | 0.88±0.04 | 1.20±0.11 | 0.46±0.04 | 0.68±0.01 | 0.61±0.06 | 0.52±0.01 | 1.07±0.06 | 0.72±0.06 | 1.91±0.16 | 2.25±0.21 |
| | + TFB ($\mathcal{D}_{\text{train}}$) | ✓ | 0.68±0.03 | 0.85±0.02 | 0.33±0.03 | 0.53±0.01 | 0.46±0.04 | 0.42±0.00 | 0.66±0.04 | 0.44±0.01 | 1.39±0.11 | 1.49±0.05 |
| | + TFB ($\mathcal{D}_{\text{test}}$) | ✓ | 0.63±0.04 | 1.00±0.06 | 0.36±0.05 | 0.51±0.02 | 0.46±0.05 | 0.36±0.01 | 0.83±0.06 | 0.51±0.06 | 1.59±0.15 | 1.94±0.10 |
| | MAP | - | 0.99±0.07 | 1.12±0.23 | 0.46±0.03 | 0.74±0.07 | 0.79±0.02 | 0.52±0.01 | 1.19±0.04 | 0.83±0.06 | 1.97±0.13 | 2.32±0.10 |
| | + TFB ($\mathcal{D}_{\text{train}}$) | ✓ | 0.77±0.05 | 0.80±0.15 | 0.38±0.04 | 0.57±0.05 | 0.61±0.03 | 0.40±0.01 | 0.96±0.08 | 0.66±0.06 | 1.69±0.16 | 2.12±0.08 |
| | + TFB ($\mathcal{D}_{\text{test}}$) | ✓ | 0.72±0.08 | 0.92±0.20 | 0.34±0.03 | 0.51±0.03 | 0.61±0.04 | 0.37±0.02 | 0.98±0.03 | 0.66±0.08 | 1.70±0.18 | 2.12±0.15 |
| | BLoB | ✗ | 0.58±0.00 | **0.51±0.03** | **0.23±0.01** | 0.43±0.01 | **0.34±0.01** | **0.26±0.01** | 0.56±0.02 | **0.35±0.02** | **1.34±0.04** | **1.35±0.10** |
| | BLoB-Mean | ✗ | 0.74±0.02 | 0.73±0.04 | 0.29±0.01 | 0.47±0.01 | 0.37±0.02 | 0.32±0.02 | 0.67±0.07 | 0.39±0.03 | 1.53±0.13 | 1.54±0.15 |
| | + TFB ($\mathcal{D}_{\text{train}}$) | ✓ | 0.55±0.01 | 0.53±0.04 | **0.23±0.02** | 0.40±0.01 | 0.33±0.02 | 0.27±0.01 | 0.52±0.05 | **0.35±0.02** | 1.36±0.13 | 1.46±0.11 |
| | + TFB ($\mathcal{D}_{\text{test}}$) | ✓ | **0.54±0.01** | 0.54±0.04 | 0.24±0.01 | 0.41±0.01 | 0.34±0.01 | 0.27±0.01 | 0.54±0.03 | 0.36±0.02 | 1.38±0.10 | 1.43±0.10 |

Table 11: **Performance of TFB with accuracy-based evaluation metric ($l$=ACC) using different posterior families applied to the mean of BLoB, based on Llama3.1-8B pre-trained weights.** **FR:** Full-rank isotropic Gaussian noises are applied to $\Delta W$; **C-STD:** Standard deviation matrix $\Omega = [\Omega_{ij} = \sigma_q]$ is constant. The evaluation protocol strictly follows Table 1. "↑" and "↓" indicate that higher and lower values are preferred, respectively. **Boldface** and underlining denote the best and the second-best performance, respectively (only for TFB variants).

| Metric | Method | In-Distribution Datasets | | | | | | Out-of-Distribution Datasets (OBQA→X) | | | |
| | | | | | | | | Small Shift | | Large Shift | |
| | | WG-S | ARC-C | ARC-E | WG-M | OBQA | BoolQ | ARC-C | ARC-E | Chem | Phy |
|---|---|---|---|---|---|---|---|---|---|---|---|
| ACC (↑) | BLoB-Mean | 77.72±0.12 | 82.60±0.60 | 91.64±0.55 | 83.92±0.48 | 88.00±0.80 | 89.86±0.05 | 82.06±1.15 | 88.54±0.31 | 39.93±5.20 | 39.93±4.02 |
| | + TFB (FR) | 76.32±0.45 | 82.13±0.38 | 91.82±0.65 | 83.33±0.85 | **88.40±0.16** | 90.03±0.18 | 82.27±1.05 | **88.24±0.37** | **43.33±5.79** | 41.33±3.30 |
| | + TFB (C-STD) | **78.40±0.36** | 82.75±0.16 | **92.38±0.08** | 83.44±0.46 | 88.37±0.50 | **90.21±0.04** | 82.38±0.89 | 87.98±0.30 | 40.00±4.68 | 41.67±4.12 |
| | + TFB (Final) | 75.28±0.33 | 82.80±0.33 | 91.64±0.15 | 81.84±0.74 | 88.00±0.16 | 89.60±0.35 | **82.93±1.91** | 86.97±0.62 | 36.00±5.66 | 36.00±5.72 |
| ECE (↓) | BLoB-Mean | 15.43±0.15 | 12.41±1.52 | 4.91±0.28 | 9.37±1.33 | 6.44±0.15 | 6.26±0.29 | 11.22±0.38 | 6.34±0.71 | 26.65±3.06 | 25.40±5.40 |
| | + TFB (FR) | 11.49±0.35 | **5.74±1.24** | **3.21±0.43** | **6.43±0.70** | **4.22±0.14** | 4.90±0.49 | 8.85±1.74 | **4.18±0.72** | 17.63±4.29 | 22.31±5.67 |
| | + TFB (C-STD) | 14.07±0.22 | 9.85±1.48 | 4.17±0.73 | 8.94±0.50 | 5.48±0.50 | 5.72±0.28 | 9.29±0.46 | 6.13±0.41 | 24.78±3.57 | 24.51±2.65 |
| | + TFB (Final) | **3.04±0.12** | 6.76±1.47 | 4.81±1.18 | 7.42±1.24 | 5.26±0.71 | **3.22±0.36** | **6.55±1.04** | 5.54±1.33 | **17.00±4.71** | **16.65±4.33** |
| NLL (↓) | BLoB-Mean | 0.74±0.02 | 0.73±0.04 | 0.29±0.03 | 0.47±0.03 | 0.37±0.02 | 0.32±0.02 | 0.67±0.07 | 0.39±0.03 | 1.53±0.13 | 1.54±0.15 |
| | + TFB (FR) | 0.61±0.00 | **0.49±0.02** | **0.23±0.03** | **0.43±0.02** | **0.32±0.01** | 0.29±0.02 | 0.61±0.04 | **0.35±0.02** | **1.36±0.11** | 1.51±0.10 |
| | + TFB (C-STD) | 0.69±0.01 | 0.59±0.05 | 0.25±0.01 | 0.46±0.02 | 0.34±0.01 | 0.31±0.01 | 0.68±0.05 | 0.37±0.02 | 1.49±0.13 | 1.45±0.13 |
| | + TFB (Final) | **0.52±0.01** | 0.52±0.03 | 0.24±0.01 | 0.45±0.01 | 0.35±0.01 | **0.28±0.00** | **0.55±0.02** | 0.37±0.01 | 1.38±0.10 | **1.41±0.09** |

Table 12: TFB **Performances with various LLM backbones [54, 55, 12, 26],** where Accuracy (**ACC**) and Expected Calibration Error (**ECE**) are reported in percentages. The MLE training for each different backbone is conducted for **2 epochs** on the concatenated dataset of six commonsense reasoning tasks, with a shared hyperparameter setting; "Fewer Epochs" represents training for **1 epoch**. We use $N = 10$ samples for TFB during inference and rows with shading indicate training-free Bayesianization methods that use a pre-trained LoRA as their mean. "↑" and "↓" indicate that higher and lower values are preferred, respectively. **Boldface** and underlining denote the best and the second-best performance, respectively.

| Metric | Method | Datasets | | | | | | |
|---|---|---|---|---|---|---|---|---|
| | | WG-S | ARC-C | ARC-E | WG-M | OBQA | BoolQ | Combined |
| ACC (↑) | Llama2-7B | **72.30**$_{\pm0.90}$ | 73.24$_{\pm1.34}$ | **87.66**$_{\pm0.81}$ | 72.30$_{\pm0.90}$ | 83.27$_{\pm1.53}$ | **87.84**$_{\pm0.57}$ | **81.41**$_{\pm0.64}$ |
| | + Fewer Epochs | 63.85$_{\pm3.68}$ | 69.23$_{\pm0.33}$ | 86.73$_{\pm0.97}$ | 63.85$_{\pm3.68}$ | 79.67$_{\pm1.55}$ | 86.08$_{\pm0.23}$ | 76.88$_{\pm1.11}$ |
| | + TFB (Ours) | 72.03$_{\pm0.88}$ | **74.36**$_{\pm1.58}$ | 87.31$_{\pm1.14}$ | **72.85**$_{\pm0.96}$ | **83.73**$_{\pm0.70}$ | 87.44$_{\pm0.34}$ | 81.32$_{\pm0.51}$ |
| | Llama3-8B | **81.45**$_{\pm0.00}$ | **84.95**$_{\pm1.53}$ | 92.63$_{\pm0.93}$ | **81.45**$_{\pm0.00}$ | **88.20**$_{\pm0.53}$ | **90.19**$_{\pm0.13}$ | **86.93**$_{\pm0.09}$ |
| | + Fewer Epochs | 79.08$_{\pm1.18}$ | 82.72$_{\pm0.39}$ | 92.22$_{\pm0.83}$ | 79.08$_{\pm1.18}$ | 86.07$_{\pm0.90}$ | 79.94$_{\pm14.97}$ | 82.01$_{\pm5.76}$ |
| | + TFB (Ours) | 81.19$_{\pm0.51}$ | 84.73$_{\pm1.68}$ | **92.98**$_{\pm0.80}$ | 81.11$_{\pm0.55}$ | 87.73$_{\pm0.64}$ | 89.75$_{\pm0.10}$ | 86.61$_{\pm0.20}$ |
| | Llama3.1-8B | **81.24**$_{\pm0.05}$ | 82.72$_{\pm0.19}$ | **92.11**$_{\pm1.05}$ | **81.24**$_{\pm0.05}$ | 87.80$_{\pm2.03}$ | **90.20**$_{\pm0.11}$ | **86.70**$_{\pm0.08}$ |
| | + Fewer Epochs | 78.11$_{\pm0.12}$ | **83.95**$_{\pm1.00}$ | 91.17$_{\pm1.17}$ | 78.11$_{\pm0.12}$ | 85.33$_{\pm0.90}$ | 89.38$_{\pm0.35}$ | 84.96$_{\pm0.22}$ |
| | + TFB (Ours) | 80.66$_{\pm0.70}$ | 82.50$_{\pm0.84}$ | 91.93$_{\pm1.05}$ | 81.22$_{\pm0.83}$ | 87.73$_{\pm1.29}$ | 89.96$_{\pm0.23}$ | 86.45$_{\pm0.33}$ |
| | Mistral-7B-v0.3 | **82.45**$_{\pm0.82}$ | **84.28**$_{\pm1.53}$ | 90.94$_{\pm0.27}$ | **82.45**$_{\pm0.82}$ | 87.73$_{\pm0.31}$ | **89.71**$_{\pm0.48}$ | **86.88**$_{\pm0.51}$ |
| | + Fewer Epochs | 79.72$_{\pm0.00}$ | 83.95$_{\pm0.33}$ | **91.58**$_{\pm0.63}$ | 79.72$_{\pm0.00}$ | 87.53$_{\pm0.31}$ | 89.20$_{\pm0.20}$ | 85.71$_{\pm0.11}$ |
| | + TFB (Ours) | 81.74$_{\pm0.43}$ | 84.06$_{\pm1.68}$ | 90.99$_{\pm0.73}$ | 81.74$_{\pm0.75}$ | **87.93**$_{\pm0.42}$ | 89.71$_{\pm0.32}$ | 86.64$_{\pm0.28}$ |
| ECE (↓) | Llama2-7B | 9.17$_{\pm0.74}$ | 9.37$_{\pm1.27}$ | **2.65**$_{\pm0.16}$ | 9.17$_{\pm0.74}$ | 5.54$_{\pm0.66}$ | **1.59**$_{\pm0.49}$ | 4.50$_{\pm0.37}$ |
| | + Fewer Epochs | **4.83**$_{\pm1.17}$ | **5.67**$_{\pm0.92}$ | 4.46$_{\pm0.23}$ | **4.83**$_{\pm1.17}$ | 4.41$_{\pm0.83}$ | 6.90$_{\pm1.73}$ | 2.00$_{\pm0.34}$ |
| | + TFB (Ours) | 5.44$_{\pm0.80}$ | 6.06$_{\pm1.54}$ | 3.83$_{\pm0.74}$ | 5.50$_{\pm1.55}$ | **3.87**$_{\pm1.15}$ | 2.51$_{\pm0.35}$ | **1.24**$_{\pm0.22}$ |
| | Llama3-8B | 8.49$_{\pm0.14}$ | 6.76$_{\pm1.77}$ | **2.57**$_{\pm0.84}$ | 8.49$_{\pm0.14}$ | 3.84$_{\pm0.37}$ | **1.88**$_{\pm1.18}$ | 4.28$_{\pm0.54}$ |
| | + Fewer Epochs | 4.45$_{\pm0.32}$ | **4.99**$_{\pm2.00}$ | 2.83$_{\pm0.58}$ | 4.45$_{\pm0.32}$ | 3.14$_{\pm0.13}$ | 2.71$_{\pm0.25}$ | 1.79$_{\pm1.16}$ |
| | + TFB (Ours) | **3.47**$_{\pm0.74}$ | 5.58$_{\pm0.58}$ | 4.34$_{\pm1.59}$ | **4.07**$_{\pm0.28}$ | 3.79$_{\pm0.90}$ | 3.49$_{\pm1.42}$ | **1.64**$_{\pm0.64}$ |
| | Llama3.1-8B | 8.58$_{\pm0.56}$ | 8.58$_{\pm0.29}$ | 2.92$_{\pm0.92}$ | 8.58$_{\pm0.56}$ | 3.85$_{\pm1.18}$ | 2.32$_{\pm0.27}$ | 4.74$_{\pm0.28}$ |
| | + Fewer Epochs | 4.76$_{\pm0.91}$ | **4.23**$_{\pm0.95}$ | 3.11$_{\pm0.76}$ | 4.76$_{\pm0.91}$ | 3.99$_{\pm0.93}$ | 3.02$_{\pm0.59}$ | 1.45$_{\pm0.38}$ |
| | + TFB (Ours) | **4.45**$_{\pm0.36}$ | 4.34$_{\pm1.29}$ | 2.97$_{\pm0.26}$ | 4.56$_{\pm0.68}$ | 3.55$_{\pm0.55}$ | 3.16$_{\pm0.45}$ | **1.05**$_{\pm0.06}$ |
| | Mistral-7B-v0.3 | 8.02$_{\pm1.68}$ | 6.98$_{\pm1.18}$ | 4.12$_{\pm0.13}$ | 8.02$_{\pm1.68}$ | 5.99$_{\pm0.48}$ | 3.17$_{\pm0.55}$ | 5.05$_{\pm0.88}$ |
| | + Fewer Epochs | 5.72$_{\pm2.01}$ | 4.74$_{\pm1.31}$ | **2.52**$_{\pm0.79}$ | 5.72$_{\pm2.01}$ | 3.50$_{\pm0.75}$ | 1.70$_{\pm0.47}$ | 2.47$_{\pm1.09}$ |
| | + TFB (Ours) | **4.47**$_{\pm2.00}$ | **4.72**$_{\pm0.83}$ | 2.62$_{\pm0.20}$ | **4.01**$_{\pm1.08}$ | 4.10$_{\pm0.26}$ | **0.97**$_{\pm0.18}$ | **1.68**$_{\pm0.53}$ |
| NLL (↓) | Llama2-7B | 0.58$_{\pm0.01}$ | 0.69$_{\pm0.03}$ | **0.35**$_{\pm0.00}$ | 0.58$_{\pm0.01}$ | 0.48$_{\pm0.03}$ | **0.30**$_{\pm0.00}$ | **0.43**$_{\pm0.00}$ |
| | + Fewer Epochs | 0.64$_{\pm0.03}$ | 0.78$_{\pm0.01}$ | 0.39$_{\pm0.01}$ | 0.64$_{\pm0.03}$ | 0.56$_{\pm0.02}$ | 0.36$_{\pm0.01}$ | 0.50$_{\pm0.01}$ |
| | + TFB (Ours) | **0.56**$_{\pm0.01}$ | **0.68**$_{\pm0.02}$ | 0.35$_{\pm0.02}$ | **0.57**$_{\pm0.01}$ | **0.46**$_{\pm0.03}$ | 0.31$_{\pm0.00}$ | **0.43**$_{\pm0.00}$ |
| | Llama3-8B | 0.48$_{\pm0.01}$ | 0.47$_{\pm0.03}$ | **0.22**$_{\pm0.01}$ | 0.48$_{\pm0.01}$ | **0.35**$_{\pm0.01}$ | **0.25**$_{\pm0.00}$ | **0.34**$_{\pm0.00}$ |
| | + Fewer Epochs | 0.46$_{\pm0.01}$ | 0.48$_{\pm0.01}$ | **0.22**$_{\pm0.02}$ | 0.46$_{\pm0.01}$ | 0.37$_{\pm0.02}$ | 0.41$_{\pm0.20}$ | 0.40$_{\pm0.08}$ |
| | + TFB (Ours) | **0.44**$_{\pm0.01}$ | **0.45**$_{\pm0.02}$ | 0.23$_{\pm0.00}$ | **0.44**$_{\pm0.01}$ | **0.35**$_{\pm0.01}$ | 0.27$_{\pm0.01}$ | **0.34**$_{\pm0.00}$ |
| | Llama3.1-8B | 0.48$_{\pm0.01}$ | 0.53$_{\pm0.01}$ | 0.24$_{\pm0.03}$ | 0.48$_{\pm0.01}$ | 0.33$_{\pm0.03}$ | 0.25$_{\pm0.00}$ | 0.35$_{\pm0.00}$ |
| | + Fewer Epochs | 0.48$_{\pm0.00}$ | **0.45**$_{\pm0.00}$ | 0.23$_{\pm0.01}$ | 0.48$_{\pm0.00}$ | 0.37$_{\pm0.01}$ | 0.27$_{\pm0.00}$ | 0.36$_{\pm0.00}$ |
| | + TFB (Ours) | **0.44**$_{\pm0.01}$ | 0.46$_{\pm0.00}$ | 0.23$_{\pm0.02}$ | **0.44**$_{\pm0.01}$ | 0.33$_{\pm0.02}$ | 0.27$_{\pm0.00}$ | **0.34**$_{\pm0.00}$ |
| | Mistral-7B-v0.3 | 0.46$_{\pm0.04}$ | 0.47$_{\pm0.02}$ | 0.28$_{\pm0.01}$ | 0.46$_{\pm0.04}$ | 0.36$_{\pm0.03}$ | 0.26$_{\pm0.01}$ | 0.35$_{\pm0.02}$ |
| | + Fewer Epochs | 0.47$_{\pm0.02}$ | 0.46$_{\pm0.01}$ | **0.25**$_{\pm0.01}$ | 0.47$_{\pm0.02}$ | **0.35**$_{\pm0.01}$ | 0.26$_{\pm0.00}$ | 0.35$_{\pm0.01}$ |
| | + TFB (Ours) | **0.42**$_{\pm0.02}$ | **0.43**$_{\pm0.02}$ | 0.26$_{\pm0.01}$ | **0.42**$_{\pm0.02}$ | 0.33$_{\pm0.01}$ | 0.26$_{\pm0.01}$ | **0.33**$_{\pm0.01}$ |

Table 13: **Performance of** TFB **when applied to variants of LoRAs [23, 68, 35, 41],** where Accuracy (**ACC**) and Expected Calibration Error (**ECE**) are reported in percentages. The MLE training for each LoRA variant is conducted with pre-trained Llama3.1-8B model for **2 epochs** on the concatenated dataset of six commonsense reasoning tasks, with a shared hyperparameter setting. We set the number of samples to $N = 10$ for TFB during inference and rows with shading indicate training-free Bayesianization methods that use a pre-trained LoRA as their mean. "↑" and "↓" indicate that higher and lower values are preferred, respectively. **Boldface** denotes the best performance.

| Metric | Method | Datasets | | | | | | |
|---|---|---|---|---|---|---|---|---|
| | | WG-S | ARC-C | ARC-E | WG-M | OBQA | BoolQ | Combined |
| ACC (↑) | LoRA | **81.24**±0.05 | **82.72**±0.19 | **92.11**±1.05 | **81.24**±0.05 | **87.80**±2.03 | **90.20**±0.11 | **86.70**±0.08 |
| | + TFB (Ours) | 80.66±0.70 | 82.50±0.84 | 91.93±1.05 | 81.22±0.83 | 87.73±1.29 | 89.96±0.23 | 86.45±0.33 |
| | VeRA | **78.24**±1.03 | **82.39**±2.55 | **90.47**±1.17 | **78.24**±1.03 | **86.13**±0.23 | **89.27**±0.27 | **84.93**±0.50 |
| | + TFB (Ours) | 76.82±0.97 | 81.27±2.34 | 90.35±0.91 | 77.03±1.04 | 86.07±0.64 | 88.99±0.32 | 84.28±0.48 |
| | PiSSA | **81.45**±1.45 | **83.95**±1.77 | 92.22±0.54 | **81.45**±1.45 | **88.40**±0.69 | **90.09**±0.11 | **86.83**±0.51 |
| | + TFB (Ours) | 80.77±1.42 | 82.94±1.21 | **92.40**±0.66 | 81.32±0.78 | 88.13±0.42 | 90.01±0.23 | 86.61±0.43 |
| ECE (↓) | LoRA | 8.58±0.56 | 8.58±0.29 | **2.92**±0.92 | 8.58±0.56 | 3.85±1.18 | **2.32**±0.27 | 4.74±0.28 |
| | + TFB (Ours) | **4.45**±0.36 | **4.34**±1.29 | 2.97±0.26 | **4.56**±0.68 | **3.55**±0.55 | 3.16±0.45 | **1.05**±0.06 |
| | VeRA | 9.54±0.47 | 7.26±2.62 | 3.72±0.86 | 9.54±0.47 | 5.41±0.78 | 2.28±0.40 | 5.11±0.55 |
| | + TFB (Ours) | **5.03**±0.92 | **5.92**±1.53 | **2.80**±0.57 | **5.09**±0.87 | **3.31**±0.84 | **1.78**±0.40 | **1.44**±0.44 |
| | PiSSA | 7.36±0.40 | 8.12±1.28 | 2.83±1.09 | 7.36±0.40 | 3.73±1.07 | 2.59±0.30 | 4.26±0.14 |
| | + TFB (Ours) | **4.59**±0.63 | **4.97**±0.63 | **2.71**±0.65 | **4.37**±0.32 | **2.96**±0.16 | **1.41**±0.64 | **1.17**±0.22 |
| NLL (↓) | LoRA | 0.48±0.01 | 0.53±0.01 | 0.24±0.03 | 0.48±0.01 | **0.33**±0.03 | **0.25**±0.00 | 0.35±0.00 |
| | + TFB (Ours) | **0.44**±0.01 | **0.46**±0.00 | **0.23**±0.02 | **0.44**±0.01 | **0.33**±0.02 | 0.27±0.00 | **0.34**±0.00 |
| | VeRA | 0.54±0.01 | 0.53±0.05 | 0.29±0.03 | 0.54±0.01 | 0.41±0.03 | **0.27**±0.01 | 0.39±0.01 |
| | + TFB (Ours) | **0.51**±0.01 | **0.51**±0.02 | **0.27**±0.02 | **0.50**±0.01 | **0.39**±0.02 | 0.28±0.01 | **0.38**±0.01 |
| | PiSSA | 0.47±0.01 | 0.49±0.02 | **0.23**±0.02 | 0.47±0.01 | **0.32**±0.03 | 0.26±0.00 | 0.35±0.00 |
| | + TFB (Ours) | **0.44**±0.01 | **0.46**±0.02 | **0.23**±0.01 | **0.44**±0.01 | **0.32**±0.02 | **0.26**±0.00 | **0.33**±0.00 |

Table 14: **Performance of Last-Layer** TFB (**LL TFB**) **applied to LoRA on Llama3.1-8B pre-trained weights,** where Accuracy (**ACC**) and Expected Calibration Error (**ECE**) are reported in percentages. The evaluation is done across six common-sense reasoning tasks with a shared hyper-parameter setting after 5 epochs. We sample $N$ times during inference in the sampling-based methods. Rows with shading indicate training-free Bayesianization methods that use a pre-trained LoRA as their mean. "↑" and "↓" indicate that higher and lower values are preferred, respectively. **Boldface** and underlining denote the best and the second-best performance, respectively.

| Metric | Method | #Sample (N) | Datasets | | | | | |
|---|---|---|---|---|---|---|---|---|
| | | | WG-S | ARC-C | ARC-E | WG-M | OBQA | BoolQ |
| ACC (↑) | MLE | - | **77.87**$_{\pm0.54}$ | 81.08$_{\pm0.48}$ | 91.67$_{\pm0.36}$ | 82.30$_{\pm0.53}$ | 87.90$_{\pm0.87}$ | 89.58$_{\pm0.26}$ |
| | + TFB | 10 | 77.44$_{\pm0.30}$ | 82.53$_{\pm1.00}$ | 91.33$_{\pm0.37}$ | 82.53$_{\pm0.56}$ | **88.53**$_{\pm0.57}$ | 89.75$_{\pm0.25}$ |
| | + LL TFB | 10 | 76.96$_{\pm0.46}$ | 82.00$_{\pm0.40}$ | 90.97$_{\pm0.34}$ | 82.67$_{\pm0.49}$ | 87.80$_{\pm1.07}$ | 89.62$_{\pm0.12}$ |
| | + LL TFB | 100 | 77.39$_{\pm0.32}$ | 82.13$_{\pm0.82}$ | 91.33$_{\pm0.37}$ | 82.61$_{\pm0.55}$ | 87.80$_{\pm0.91}$ | 89.66$_{\pm0.28}$ |
| | MAP | - | 76.90$_{\pm0.97}$ | 81.08$_{\pm2.48}$ | 91.61$_{\pm0.44}$ | 82.59$_{\pm0.28}$ | 85.73$_{\pm0.19}$ | 90.09$_{\pm0.28}$ |
| | + TFB (Ours) | 10 | 76.43$_{\pm0.72}$ | 82.80$_{\pm1.42}$ | 91.39$_{\pm0.37}$ | 82.64$_{\pm0.58}$ | 86.00$_{\pm0.16}$ | 89.96$_{\pm0.18}$ |
| | + LL TFB | 10 | 76.35$_{\pm0.89}$ | 83.07$_{\pm1.97}$ | 91.15$_{\pm0.52}$ | 82.27$_{\pm0.53}$ | 85.27$_{\pm0.19}$ | 90.09$_{\pm0.20}$ |
| | + LL TFB | 100 | 76.72$_{\pm0.77}$ | 83.07$_{\pm2.12}$ | 91.15$_{\pm0.60}$ | 82.53$_{\pm0.33}$ | 85.60$_{\pm0.16}$ | 90.02$_{\pm0.14}$ |
| | BLoB | 10 | 76.45$_{\pm0.37}$ | 82.32$_{\pm1.15}$ | 91.14$_{\pm0.54}$ | 82.01$_{\pm0.56}$ | 87.57$_{\pm0.21}$ | 89.65$_{\pm0.15}$ |
| | BLoB-Mean | - | 77.72$_{\pm0.12}$ | 82.60$_{\pm0.60}$ | 91.64$_{\pm0.55}$ | **83.92**$_{\pm0.48}$ | 88.00$_{\pm0.80}$ | 89.86$_{\pm0.05}$ |
| | + TFB (Ours) | 10 | 77.81$_{\pm0.36}$ | **83.33**$_{\pm0.19}$ | **91.76**$_{\pm0.48}$ | 83.81$_{\pm0.39}$ | 87.80$_{\pm0.16}$ | **90.11**$_{\pm0.28}$ |
| | + LL TFB | 10 | 77.57$_{\pm1.02}$ | 82.80$_{\pm0.33}$ | 91.45$_{\pm0.54}$ | 83.23$_{\pm0.57}$ | 88.33$_{\pm0.09}$ | 89.85$_{\pm0.13}$ |
| | + LL TFB | 100 | 77.60$_{\pm0.62}$ | **83.33**$_{\pm0.82}$ | 91.39$_{\pm0.60}$ | 83.63$_{\pm0.62}$ | 87.60$_{\pm0.43}$ | 90.03$_{\pm0.03}$ |
| ECE (↓) | MLE | - | 17.02$_{\pm0.46}$ | 16.35$_{\pm0.68}$ | 7.00$_{\pm0.53}$ | 13.83$_{\pm0.65}$ | 9.77$_{\pm0.81}$ | 8.69$_{\pm0.21}$ |
| | + TFB (Ours) | 10 | 12.98$_{\pm0.37}$ | 11.63$_{\pm0.68}$ | 5.14$_{\pm0.14}$ | 10.01$_{\pm0.70}$ | 7.20$_{\pm0.47}$ | 7.39$_{\pm0.26}$ |
| | + LL TFB | 10 | 14.42$_{\pm0.41}$ | 13.86$_{\pm0.45}$ | 6.92$_{\pm0.62}$ | 10.32$_{\pm0.90}$ | 8.56$_{\pm0.96}$ | 7.52$_{\pm0.12}$ |
| | + LL TFB | 100 | 13.45$_{\pm0.30}$ | 13.17$_{\pm0.62}$ | 6.84$_{\pm0.67}$ | 10.76$_{\pm0.88}$ | 8.68$_{\pm0.60}$ | 7.46$_{\pm0.10}$ |
| | MAP | - | 18.71$_{\pm0.74}$ | 15.77$_{\pm1.60}$ | 6.62$_{\pm0.64}$ | 14.26$_{\pm0.92}$ | 12.19$_{\pm0.55}$ | 8.40$_{\pm0.25}$ |
| | + TFB (Ours) | 10 | 14.95$_{\pm0.65}$ | 11.27$_{\pm2.53}$ | 5.76$_{\pm0.63}$ | 10.97$_{\pm1.19}$ | 9.70$_{\pm0.69}$ | 6.86$_{\pm0.31}$ |
| | + LL TFB | 10 | 16.03$_{\pm0.64}$ | 12.72$_{\pm1.33}$ | 6.54$_{\pm0.68}$ | 12.06$_{\pm1.09}$ | 11.36$_{\pm0.34}$ | 7.51$_{\pm0.23}$ |
| | + LL TFB | 100 | 15.56$_{\pm0.97}$ | 12.84$_{\pm2.17}$ | 6.38$_{\pm0.66}$ | 11.80$_{\pm1.14}$ | 11.22$_{\pm0.38}$ | 7.30$_{\pm0.41}$ |
| | BLoB | 10 | 9.93$_{\pm0.22}$ | **5.41**$_{\pm1.17}$ | 2.70$_{\pm0.87}$ | 4.28$_{\pm0.64}$ | 2.91$_{\pm0.92}$ | 2.58$_{\pm0.25}$ |
| | BLoB-Mean | - | 15.43$_{\pm0.15}$ | 12.41$_{\pm1.52}$ | 4.91$_{\pm0.28}$ | 9.37$_{\pm1.33}$ | 6.44$_{\pm0.15}$ | 6.26$_{\pm0.29}$ |
| | + TFB (Ours) | 10 | **8.16**$_{\pm0.48}$ | 6.48$_{\pm0.36}$ | **2.44**$_{\pm0.50}$ | 3.83$_{\pm0.43}$ | **2.67**$_{\pm0.18}$ | 3.10$_{\pm0.59}$ |
| | + LL TFB | 10 | 9.68$_{\pm0.70}$ | 7.20$_{\pm0.91}$ | 3.01$_{\pm0.66}$ | 3.94$_{\pm0.78}$ | 3.33$_{\pm0.93}$ | 2.96$_{\pm0.30}$ |
| | + LL TFB | 100 | 8.88$_{\pm0.32}$ | 6.47$_{\pm1.55}$ | 2.84$_{\pm0.50}$ | **3.40**$_{\pm0.82}$ | 3.70$_{\pm0.27}$ | **2.51**$_{\pm0.46}$ |
| NLL (↓) | MLE | - | 0.88$_{\pm0.04}$ | 1.20$_{\pm0.11}$ | 0.46$_{\pm0.04}$ | 0.68$_{\pm0.01}$ | 0.61$_{\pm0.06}$ | 0.52$_{\pm0.01}$ |
| | + TFB (Ours) | 10 | 0.68$_{\pm0.03}$ | 0.85$_{\pm0.02}$ | 0.33$_{\pm0.03}$ | 0.53$_{\pm0.01}$ | 0.46$_{\pm0.04}$ | 0.42$_{\pm0.00}$ |
| | + LL TFB | 10 | 0.70$_{\pm0.02}$ | 0.96$_{\pm0.12}$ | 0.41$_{\pm0.06}$ | 0.53$_{\pm0.02}$ | 0.50$_{\pm0.06}$ | 0.42$_{\pm0.01}$ |
| | + LL TFB | 100 | 0.66$_{\pm0.02}$ | 0.84$_{\pm0.08}$ | 0.39$_{\pm0.07}$ | 0.53$_{\pm0.02}$ | 0.49$_{\pm0.05}$ | 0.40$_{\pm0.00}$ |
| | MAP | - | 0.99$_{\pm0.07}$ | 1.12$_{\pm0.23}$ | 0.46$_{\pm0.03}$ | 0.74$_{\pm0.07}$ | 0.79$_{\pm0.02}$ | 0.52$_{\pm0.01}$ |
| | + TFB (Ours) | 10 | 0.77$_{\pm0.05}$ | 0.80$_{\pm0.15}$ | 0.38$_{\pm0.03}$ | 0.57$_{\pm0.05}$ | 0.61$_{\pm0.03}$ | 0.40$_{\pm0.01}$ |
| | + LL TFB | 10 | 0.80$_{\pm0.07}$ | 0.88$_{\pm0.19}$ | 0.43$_{\pm0.02}$ | 0.60$_{\pm0.05}$ | 0.65$_{\pm0.01}$ | 0.43$_{\pm0.02}$ |
| | + LL TFB | 100 | 0.77$_{\pm0.06}$ | 0.86$_{\pm0.18}$ | 0.41$_{\pm0.02}$ | 0.57$_{\pm0.04}$ | 0.63$_{\pm0.02}$ | 0.40$_{\pm0.03}$ |
| | BLoB | 10 | 0.58$_{\pm0.00}$ | **0.51**$_{\pm0.03}$ | **0.23**$_{\pm0.01}$ | 0.43$_{\pm0.01}$ | 0.34$_{\pm0.01}$ | **0.26**$_{\pm0.01}$ |
| | BLoB-Mean | - | 0.74$_{\pm0.02}$ | 0.73$_{\pm0.04}$ | 0.29$_{\pm0.03}$ | 0.47$_{\pm0.03}$ | 0.37$_{\pm0.02}$ | 0.32$_{\pm0.02}$ |
| | + TFB | 10 | 0.55$_{\pm0.01}$ | 0.53$_{\pm0.04}$ | **0.23**$_{\pm0.02}$ | 0.40$_{\pm0.01}$ | 0.33$_{\pm0.02}$ | 0.27$_{\pm0.01}$ |
| | + LL TFB | 10 | 0.56$_{\pm0.02}$ | 0.60$_{\pm0.05}$ | 0.26$_{\pm0.02}$ | 0.41$_{\pm0.01}$ | 0.33$_{\pm0.01}$ | 0.27$_{\pm0.01}$ |
| | + LL TFB | 100 | **0.53**$_{\pm0.01}$ | 0.54$_{\pm0.04}$ | 0.24$_{\pm0.01}$ | **0.39**$_{\pm0.01}$ | **0.31**$_{\pm0.01}$ | **0.26**$_{\pm0.01}$ |

Table 15: **Performance of different methods applied to LoRA on Llama2-7B pre-trained weights,** where Accuracy (**ACC**) and Expected Calibration Error (**ECE**) are reported in percentages. **"TF?"** denotes whether a method is **T**raining-**F**ree. The evaluation is done across six common-sense reasoning tasks with a shared hyper-parameter setting after 5,000 gradient steps. We sample $N = 10$ during inference in all sampling-based methods including **BLoB [62]** and TFB. Rows with shading indicate training-free Bayesianization methods that use pre-trained LoRA as their mean. For TFB, the anchor dataset $\mathcal{D}$ is set to a randomly sampled subset of the original training set, the performance evaluation metric $l$ is set to accuracy, and the performance drop tolerance is set adaptively to 1% or 0.5% based on whether the given mean overfits. "↑" and "↓" indicate that higher and lower values are preferred, respectively. **Boldface** and underlining denote the best and the second-best performance, respectively.

| Metric | Method | TF? | In-Distribution Datasets | | | | | | Out-of-Distribution Datasets (OBQA→X) | | | |
| | | | | | | | | | Small Shift | | Large Shift | |
| | | | WG-S | ARC-C | ARC-E | WG-M | OBQA | BoolQ | ARC-C | ARC-E | Chem | Phy |
|---|---|---|---|---|---|---|---|---|---|---|---|---|
| ACC (↑) | MCD | ✗ | 69.46±0.62 | 68.69±1.30 | 86.21±0.46 | **76.45±0.04** | 81.72±0.10 | 87.29±0.13 | 69.03±0.70 | 76.00±1.58 | 42.71±0.01 | 29.17±4.54 |
| | ENS | ✗ | 69.57±0.66 | 66.20±2.01 | 84.40±0.81 | 75.32±0.21 | 81.38±0.91 | 87.09±0.11 | 67.34±0.70 | 75.18±2.03 | 43.75±1.04 | 30.56±2.62 |
| | BBB | ✗ | 56.54±7.87 | 68.13±1.27 | 85.86±0.74 | 73.63±2.44 | 82.06±0.59 | 87.21±0.22 | 67.25±1.18 | 75.83±0.75 | 42.36±0.49 | 30.21±2.25 |
| | LAP | BP | 69.20±1.50 | 66.78±0.69 | 80.05±0.22 | 75.55±0.36 | 82.12±0.67 | 86.95±0.09 | 69.14±1.15 | 74.94±0.96 | 44.10±1.30 | 31.60±0.49 |
| | MLE | - | 68.99±0.58 | 69.10±2.84 | 85.65±0.92 | 74.53±0.66 | 81.52±0.25 | 86.53±0.28 | 66.20±0.87 | 75.12±0.85 | 40.62±2.25 | 28.82±1.30 |
| | + TFB (Ours) | ✓ | 69.83±1.02 | 68.13±1.03 | 86.21±0.90 | 75.95±0.34 | 82.80±0.35 | 87.66±0.35 | 69.93±2.11 | 78.87±1.06 | 34.67±3.51 | 31.00±2.00 |
| | MAP | - | 68.62±0.71 | 67.59±0.40 | 86.55±0.55 | 75.61±0.71 | 81.38±0.65 | 86.50±0.41 | 69.59±0.33 | 75.47±0.73 | 44.79±0.00 | 28.47±1.20 |
| | + TFB (Ours) | ✓ | 69.17±1.08 | 67.68±1.73 | 85.86±0.37 | 75.87±0.40 | 83.07±0.61 | **87.74±0.23** | 69.37±2.54 | 78.76±0.87 | 34.33±5.51 | 31.00±1.00 |
| | BLoB | ✗ | 68.80±0.53 | 67.59±0.43 | 86.37±0.34 | 73.26±1.36 | 81.99±1.48 | 86.58±0.18 | 67.71±1.13 | 76.37±0.80 | 44.79±1.47 | 31.60±2.73 |
| | BLoB-Mean | ✗ | **72.15±0.17** | 69.56±0.91 | 86.31±0.37 | 75.47±1.36 | 82.53±0.74 | 86.69±0.08 | 69.93±1.20 | 76.88±0.41 | 41.67±2.25 | **31.94±1.77** |
| | + TFB (Ours) | ✓ | 69.94±1.68 | **70.72±2.25** | **86.74±0.97** | 73.13±2.38 | **83.13±0.76** | 86.36±0.26 | **70.38±1.03** | **80.16±0.71** | 42.67±1.15 | 30.67±1.53 |
| ECE (↓) | MCD | ✗ | 27.98±0.44 | 27.53±0.80 | 12.20±0.56 | 19.55±0.47 | 13.10±0.11 | 3.46±0.16 | 19.54±0.33 | 15.32±1.16 | 17.9±0.63 | 29.53±4.20 |
| | ENS | ✗ | 28.52±0.55 | 29.16±2.37 | 12.57±0.58 | 20.86±0.43 | 15.34±0.27 | 9.61±0.24 | 7.59±1.43 | 6.44±0.83 | 12.04±4.57 | 17.52±1.28 |
| | BBB | ✗ | 21.81±12.95 | 26.23±1.47 | 12.28±0.58 | 15.76±4.71 | 11.38±1.07 | 3.74±0.10 | 19.90±0.66 | 13.41±0.85 | 15.67±1.23 | 26.10±4.76 |
| | LAP | BP | **4.15±1.12** | 16.25±2.61 | 33.29±0.57 | 7.40±0.27 | 8.70±1.77 | 1.30±0.33 | **5.84±0.64** | 8.51±1.06 | 10.76±3.41 | **13.91±0.90** |
| | MLE | - | 29.83±0.58 | 29.00±1.97 | 13.12±1.39 | 20.62±0.74 | 12.55±0.46 | 3.18±0.09 | 22.20±0.39 | 16.47±0.86 | 21.72±0.30 | 29.60±1.29 |
| | + TFB (Ours) | ✓ | 16.26±0.36 | 6.93±1.43 | 5.82±0.87 | 8.78±0.84 | 4.60±0.62 | 2.30±0.50 | 8.47±2.04 | **4.64±0.75** | 15.87±5.17 | 16.77±4.10 |
| | MAP | - | 29.76±0.87 | 29.42±0.68 | 12.07±0.55 | 23.07±0.14 | 13.26±0.82 | 3.16±0.23 | 19.31±1.46 | 15.68±0.51 | 17.55±1.95 | 30.25±2.18 |
| | + TFB (Ours) | ✓ | 11.72±0.56 | 6.07±1.89 | 6.99±0.96 | 5.21±0.86 | 3.82±0.60 | 2.65±0.30 | 8.39±0.75 | 4.86±1.03 | 16.11±3.22 | 16.35±2.94 |
| | BLoB | ✗ | 8.98±0.58 | 10.81±1.29 | 4.54±0.90 | **3.98±1.04** | **3.64±0.54** | **1.24±0.33** | 9.55±0.40 | 5.48±1.27 | **9.77±1.35** | 18.29±1.35 |
| | BLoB-Mean | ✗ | 18.63±0.31 | 22.51±0.93 | 9.64±0.60 | 11.58±1.24 | 8.65±0.98 | 2.88±0.07 | 14.00±1.02 | 10.70±0.39 | 15.05±0.77 | 22.90±2.27 |
| | + TFB (Ours) | ✓ | 6.33±1.04 | **5.77±0.32** | **3.03±0.43** | 4.07±1.65 | 5.94±0.46 | 5.37±0.44 | 12.28±1.24 | 8.07±1.01 | 12.36±1.73 | 22.02±0.30 |
| NLL (↓) | MCD | ✗ | 2.79±0.53 | 2.67±0.15 | 1.00±0.14 | 1.02±0.03 | 0.77±0.03 | 0.31±0.00 | 1.08±0.01 | 0.88±0.03 | 1.59±0.07 | 1.67±0.05 |
| | ENS | ✗ | 2.71±0.46 | 2.46±0.22 | 0.82±0.03 | 1.25±0.03 | 1.06±0.04 | 0.57±0.02 | 0.86±0.01 | 0.69±0.03 | **1.28±0.00** | 1.39±0.03 |
| | BBB | ✗ | 1.40±0.55 | 2.23±0.04 | 0.91±0.06 | 0.84±0.15 | 0.66±0.05 | 0.31±0.00 | 1.06±0.01 | 0.79±0.02 | 1.49±0.05 | 1.62±0.06 |
| | LAP | BP | **0.60±0.00** | 1.03±0.04 | 0.88±0.00 | 0.57±0.01 | 0.52±0.01 | 0.31±0.00 | 0.81±0.00 | 0.70±0.02 | 1.35±0.03 | 1.36±0.01 |
| | MLE | - | 3.17±0.37 | 2.85±0.27 | 1.17±0.13 | 0.95±0.07 | 0.73±0.03 | 0.32±0.00 | 1.16±0.00 | 0.92±0.03 | 1.56±0.06 | 1.66±0.05 |
| | + TFB (Ours) | ✓ | 0.86±0.06 | 0.98±0.02 | 0.48±0.04 | 0.59±0.01 | 0.54±0.02 | 0.30±0.00 | 0.87±0.03 | 0.70±0.05 | 1.46±0.03 | 1.43±0.05 |
| | MAP | - | 2.46±0.34 | 2.66±0.11 | 0.90±0.05 | 1.62±0.29 | 0.75±0.00 | 0.33±0.00 | 1.10±0.07 | 0.93±0.04 | 1.55±0.06 | 1.65±0.03 |
| | + TFB (Ours) | ✓ | 0.72±0.03 | 0.96±0.03 | 0.50±0.04 | 0.55±0.01 | 0.53±0.02 | **0.30±0.00** | 0.87±0.02 | 0.71±0.05 | 1.46±0.02 | 1.42±0.05 |
| | BLoB | ✗ | 0.63±0.01 | **0.84±0.00** | **0.41±0.02** | **0.54±0.01** | 0.49±0.01 | 0.31±0.00 | **0.83±0.01** | **0.60±0.01** | 1.38±0.01 | 1.46±0.02 |
| | BLoB-Mean | ✗ | 0.79±0.01 | 1.27±0.02 | 0.57±0.03 | 0.60±0.03 | 0.56±0.00 | 0.32±0.01 | 0.89±0.02 | 0.67±0.02 | 1.44±0.00 | 1.53±0.02 |
| | + TFB (Ours) | ✓ | 0.62±0.03 | 0.86±0.01 | 0.42±0.03 | 0.56±0.03 | 0.50±0.01 | 0.34±0.00 | 0.84±0.03 | 0.61±0.01 | 1.35±0.01 | 1.46±0.06 |

