# OpenReview forum: "Training-Free Bayesianization for Low-Rank Adapters of Large Language Models"
_NeurIPS.cc/2025/Conference — NeurIPS 2025 poster_

### Official Review · Reviewer_3U1X · 2025-06-27

**Clarity:** 2
**Significance:** 2
**Originality:** 2
**Rating:** 3
**Confidence:** 4

**Summary:**

This paper proposes Training-Free Bayesianization (TFB), which makes pre-trained LoRA adapters into Bayesian adapters. It provides a practical way of doing uncertainty quantification in LLMs, and the paper shows that TFB's variance maximization is equivalent to generalized variational inference with some mild conditions. The paper also shows superiority in empirical results.

**Questions:**

- line 21: "its internal internal" is it a typo?
- line 86: When you denote matrix as $\boldsymbol{X}=\left[\boldsymbol{x}_1, \cdots, \boldsymbol{x}_n\right] \in \mathbb{R}^{m \times n}$ with comma separted, I don't think your definition for notation of the vector, $\operatorname{vec}(\boldsymbol{X})=\left[\boldsymbol{x}_1^{\top}, \boldsymbol{x}_2^{\top}, \cdots, \boldsymbol{x}_n^{\top}\right]^{\top} \in \mathbb{R}^{(m n) \times 1}$, could be confusing
- line 93: What is "the" LoRA layer you are referring to here? I think it would be better to mention what that layer is.
Is there any difference between "LoRA", "LoRA adapter", and "low-rank adapter"? The authors seem to use different expressions that refer to the same thing.
-  line 100: What is the "prior" here? Are $B, \theta$ observations? It is hard to understand what "Bayesianized low-rank adapter's posterior" is. Why do you call it Bayesian at all?
Also, I guess since the authors used $q$ to denote the approximate posterior(?) of $\operatorname{vec}(\boldsymbol{W})$, should I also understand $q\left(A_{i j}\right)$ to be the approximate posterior of $A_{i j}$?
- line 126: Is it correct that the mean of the posterior distribution of a random variable is the random variable itself? ( $\boldsymbol{M}=\boldsymbol{A}^{\prime}$ )
- line 186: What is $P_{\boldsymbol{\theta}}(y \mid \boldsymbol{x})$ and what is $P\left(\boldsymbol{y} \mid \boldsymbol{x}, \boldsymbol{W}_n\right)$? Is the latter a true probability density function?
- line 187: Does $\theta$ include $B$ as well? or is it just $\{M, \Omega\}$ as it was first defined in line 101
- line 188: It is not valid to justify the selection of such a small number of Monte Carlo samples ( $N=10$ ) by referring to previous work. Do the authors acknowledge possible errors due to the gap between $P_{\boldsymbol{\theta}}(y \mid \boldsymbol{x})$ and the empirical average?
- line 210: Missing \prime for A on the right-hand side? $q\left(\boldsymbol{A}^{\prime} \mid \boldsymbol{\theta}\right)=\prod_{i j} \mathcal{N}\left(A_{i j} \mid M_{i j}, \Omega_{i j}^2\right)$
- In table 1, is the standard deviation calculated with $N=10$ samples?

**Ethical Concerns:**

["NO or VERY MINOR ethics concerns only"]

**Final Justification:**

After the rebuttal discussion, I would like to keep my score as it is since there are still things to be revised in the manuscript for an acceptance.

**Limitations:**

yes

**Quality:**

2

**Strengths And Weaknesses:**

Strength
- This paper brings a novel way to change pretrained LoRA adapters into a Bayesian setting
- It also provides theoretical grounds for showing the equivalence between TFB's variance maximization and generalized variational inference with mild conditions
- It covers a decent amount of dataset/benchmark models to compare in the empirical analysis

Weakness
- Some of the arguments are not clear, and the presentation could be improved. Also, it is a little bit confusing why they call this Bayesianization at the first place.
- Please refer to the questions for more details

---

> ### Author Rebuttal · Authors · 2025-07-28
>
> We thank you for your constructive feedback.
> We are glad you find our approach `"novel"`, with `"theoretical grounds"` under `"mild conditions"`, and that our experiments cover `"a decent amount of dataset/benchmark models"` and show `"superiority"`. Below we address your comments one by one in detail.
>
> **W1.1 Some of the arguments are not clear, and the presentation could be improved.**
>
> We are sorry for the confusion and will include all clarifications (W1.2-Q10 below) in the revision.
>
> **W1.2 Also, it is a little bit confusing why they call this Bayesianization at the first place.**
>
> We apologize for the confusion. We follow prior work to use "Bayesianization" as the noun form of "Bayesianize" [1, 2]. This term aptly describes TFB’s process of transforming a pretrained, deterministic, non-Bayesian LoRA into a Bayesian model.
>
>
> **Q1 \& Q9. line 21: "its internal internal" is it a typo? \& line 210: Missing \prime for A on the right-hand side?**
>
> We are sorry for the typos and will fix them in the revision.
>
> **Q2. line 93: When you denote matrix as ... with comma separted, I don't think your definition for notation of the vector ... could be confusing**
>
> Here in matrix $X=[x_1, \cdots, x_n]\in \mathbb{R}^{m\times n}$, the column vectors $x\_i\in \mathbb{R}^{m\times 1}$ are separated by commas, making $x\_i^\top \in \mathbb{R}^{1\times m}$ a *row* vector.
> Therefore $[x_1^\top, \cdots, x_n^\top]\in \mathbb{R}^{1\times (mn)}$ is a long *row* vector, and transposing this long row vector leads to a long *column* vector $[x_1^\top, \cdots, x_n^\top]^\top \in \mathbb{R}^{(mn)\times 1}$.
>
> **Q3. line 93: What is "the" LoRA layer you are referring to here? I think it would be better to mention what that layer is. Is there any difference between "LoRA", "LoRA adapter", and "low-rank adapter"? The authors seem to use different expressions that refer to the same thing.**
>
> We apologize for the confusion.
> Ideally, `"LoRA adapter"` and `"LoRA"` refer specifically to the two low-rank matrices $\\{B, A\\}$, while `"LoRA layer"` refers to the LoRA adapter together with the pre-trained weights $W_0$. In Line 93, the LoRA layer refers to the operation $BAh$ in Eqn. (1), where z is the output.
>
> In this paper, we have occasionally `"LoRA"`, `"LoRA layer"`, and `"LoRA adapter"` interchangeablly.
> We will clarify these distinctions and use the terms more consistently in the revision, as you suggested.
>
>
>
> **Q4. line 100: What is the "prior" here? Are $B, \theta$ observations? It is hard to understand what "Bayesianized low-rank adapter's posterior" is. Why do you call it Bayesian at all? Also, I guess since the authors used $q$ to denote the approximate posterior(?) of $\text{vec}(W)$, should I also understand $q(A_{ij})$ to be the approximate posterior of $A_{ij}$?**
>
> **Prior.** The prior of TFB is a Gaussian distribution with mean equal to the pre-trained weights $W_0$ and covariance $\sigma_p I$, where $\sigma_p$ is larger than the local convexity range $\epsilon_0$. See `Theorem 4.2` and `Eq. 30` for details.
>
> **Approximate Posterior $q(A_{ij})$.** Here in Eq. 2, we introduce LoRA Layer with Asymmetric Bayesianization [2], where $B$ and $\theta=\\{M, \Omega\\}$ are the parameterization, but not observations. As you mentioned, it is appropriate to understand $q(A)$ as the approximate posterior of the LoRA component $A$, and consequently, Eq. 2 is the **equivalent approximate posterior** of the full random weight matrix $W$.
>
> **Why Call It Bayesian.** The reason why the preceding work of BLoB [2] calls it Bayesian is that they define a prior distribution $P(W)$ on the full-weight matrix. They then optimize the ELBO from scratch in a similar way as described in Eq. 12. This ELBO contains the data-likelihood term $\mathbb{E}_{q(W|\theta)}[\log P(D|W)]$ and the KL-divergence term between the approximate posterior and the prior $KL[q(W|\theta)\|P(W)]$.
>
>
> **Q5. line 126: Is it correct that the mean of the posterior distribution of a random variable is the random variable itself? ($M=A^\prime$)**
>
> We apologize for the overloaded definition of $A'$.
> Based on your advice, we have revised it to:
>
> "We convert the deterministic matrix $A'$ into a variational distribution with the mean matrix $M=A'$ and the standard deviation matrix $\Omega$."
>
>
> **Q6. line 186: What is $P_\theta(y|x)$ and what is $P(y|x,W_n)$? Is the latter a true probability density function?**
>
>
> Typically, a Bayesian neural network's output is the expected output $E_{q(W|\theta)}[P(y|x,W)]$, where $x$ is the input, $y$ is the output, and $q(W|\theta)$ is the approximate posterior of the LLM parameters $W$.
>
> In practice, we approximate this expectation through Monte Carlo sampling: $E_{q(W|\theta)}[P(y|x,W)]\approx \frac{1}{N}\sum_{n=1}^{N} P(y|x, W_n)$, where $W_n\sim q(W|\theta)$ is the $n$-th sample drawn from the approximate posterior distribution, $N$ is the total number of samples, and $P(y|x,W_n)$ is the probability distribution on the output variable, depending on the sampled LLM parameters $W_n$.
>
>
> **Q7. line 187: Does $\theta$ include $B$ as well? or is it just $M,\Omega$ as it was first defined in line 101**
>
> You are right. $\theta=\\{M,\Omega\\}$, as the set of parameters modeling the approximate posterior of the weight matrices, does **not** include $B$. This is because the matrix $B$ is kept deterministic and the sampling is performed only over the random variable $A$ (whose mean and standard deviation are $M$ and $\Omega$, respectively).
>
>
> **Q8. line 188: It is not valid to justify the selection of such a small number of Monte Carlo samples ($N=10$) by referring to previous work. Do the authors acknowledge possible errors due to the gap between $P_\theta(y|x)$ and the empirical average?**
>
> This is a good point. Increasing the number of samples generally yields more accurate estimates of the expected output, thereby improving model performance in terms of ECE and NLL. This observation is supported by previous work [2] and partially by our analysis in `Table 5, Section 5.6`.
>
> Inspired by your comments, we ran additional experiments and report TFB's performance for sample sizes ranging from $N=0$ to $N=160$ on the WG-S dataset. Each experiment was repeated three times with different random seeds on Llama3.1-8B, and the results are reported as `mean (std)` in the table below:
>
> **Table A. Influence of the number of samples at test time for TFB.**
>
> | | **N=0** | N=1 | N=2| N=3| N=4| N=5| N=10  | N=20| N=40 | N=80 | N=160 |
> |-|-|-|---|---|---|---|---|---|---|---|---|
> | **ACC ($\uparrow$)**   | 77.72 (0.12) | 74.11 (0.65) | 75.84 (0.62) | 75.81 (0.73) | 77.28 (0.52) | 76.83 (0.53) | 76.96 (0.36) | 77.55 (0.25) | 77.89 (0.38) | 77.55 (0.08) | 78.11 (0.20) |
> | **ECE ($\downarrow$)** | 15.43 (0.15) | 17.99 (1.06) | 12.68 (0.58) | 10.03 (0.67) | 8.17 (0.79)  | 8.34 (0.22)  | 7.59 (0.34)  | 6.68 (0.27)  | 6.15 (0.54)  | 6.24 (0.66)  | 6.00 (0.23)  |
> | **NLL ($\downarrow$)** | 0.74 (0.02) | 0.96 (0.02)  | 0.72 (0.01)  | 0.64 (0.01)  | 0.60 (0.01)  | 0.58 (0.01)  | 0.53 (0.01)  | 0.52 (0.01)  | 0.51 (0.01)  | 0.50 (0.01)  | 0.50 (0.01)  |
>
> As shown in the table, increasing the number of samples at test time generally improves the alignment between the Monte Carlo estimates and expectation in theory, leading to better uncertainty estimation.
>
> Our choice of $N=10$ samples in the paper balances performance and test-time efficiency: it achieves significant improvement in ECE and NLL while introducing acceptable computational overhead. The performance gap between this setting and extremely large $N$ is also mild.
>
>
> **Q10. In table 1, is the standard deviation calculated with $N=10$ samples?**
>
> For all baselines and our TFB, we run all experiments with $3$ different random seeds and report the standard deviation. $N=10$ is the number of samples we use for Monte Carlo estimation of the Bayesian Model Averaging.
>
>
> **References**
>
> [1] Bacharach, Michael, and Susan Hurley. "Issues and advances in the foundations of decision theory." Foundations of decision theory (1991): 1-38.
>
> [2] Wang, Yibin, et al. "Blob: Bayesian low-rank adaptation by backpropagation for large language models." Advances in Neural Information Processing Systems 37 (2024): 67758-67794.

---

> > ### Comment · Reviewer_3U1X · 2025-08-02
> >
> > Thanks for the rebuttal response. Can you provide more details on the experiments you conducted above? Which table should I see to compare the numbers to? I guess the N=10 results you provided have to match some numbers from the table in the manuscript?

---

> > > ### Author Response · Authors · 2025-08-02
> > >
> > > Thank you for the follow-up comment and for keeping the communication channel open.
> > >
> > > The follow-up results shown in `Table A` above were obtained under the same experimental setting as `Column "WG-S", Table 1` in the paper. The slight difference in the TFB results for $N=10$ (ECE = 7.59±0.34, NLL = 0.53±0.01) compared to the main table (ECE = 8.16±0.48, NLL = 0.55±0.01) arises because we used three different random seeds in this experiment when scaling up the number of samples at test time. It is worth noting that the result reported in `Table A` is actually slightly better than the one in the original paper. This demonstrates that we are not cherry-picking results to show larger improvements but are faithfully reporting them.
> > >
> > > We hope these additional experiments and explanations address your question.

---

> ### Author Response · Authors · 2025-08-09
>
> Dear Reviewer 3U1X,
>
> With the author–reviewer discussion closing in less than a day, we would greatly appreciate it if you could let us know whether our rebuttal has fully addressed your concerns or if further clarification would be helpful. We remain committed to responding to any follow-up questions. Thank you again for your time and thoughtful feedback!
>
> Best regards,
>
> TFB Authors

---

### Official Review · Reviewer_3SzA · 2025-06-30

**Clarity:** 3
**Significance:** 3
**Originality:** 4
**Rating:** 5
**Confidence:** 2

**Summary:**

The paper considers uncertainty estimation for Large Language Models (LLMs), specifically low-rank adapters for LLMs. It offers a simple approach to do with low-rank matrix found by an existed low-rank adapter, then treating it as matrix of isotropic Gaussians with the same variance. This variance is optimised (as an example binary search is used in the paper). Then predictions are averaged with weights sampled with these Gaussians. Despite simplicity of this approach it is shown to be theoretically grounded and equivalent to generalised variational inference.

**Questions:**

From the practical point of view, why do you claim TFB to be training free if it still involves optimisation of the variance (which requires a training dataset)?

**Ethical Concerns:**

["NO or VERY MINOR ethics concerns only"]

**Final Justification:**

After reading all other reviews and authors' responses I am still voting to accept the paper. I was quite impressed how many additional experimental results the authors' were able to include during the rebuttal process, all in favour of their method.

**Limitations:**

The authors discuss both the limitations of their work and societal impact of their work.

**Quality:**

3

**Strengths And Weaknesses:**

**Strengths:**

*Quality*: The main claims of the paper are supported with good quality experiments and also grounded theoretically linking the proposed approach with generalised variational inference.

*Clarity*: The paper is well written and easy to follow, after reading it though I have one confusion, please see below.

*Significance*: With the spread of LLM usage uncertainty estimation is very important yet it still remains an open problem, therefore this paper aiming to address it can have a significant impact.

*Originality*: The idea appears to be novel.

**Weaknesses:**

*Clarity and originality*: The paper emphasises that it offers a training-free method, which makes it unique amongst existing methods. However, I am finding myself confused why it is claimed to be training-free. It still requires to find the variance. I can get that on a higher level it is not training, but from the practical point of view, isn’t it just the same as training?

*Originality*: The paper is generally well put into the existing literature but after a quick search I have found this work, for example, Onal, Emre, et al. "Gaussian stochastic weight averaging for Bayesian low-rank adaptation of large language models." arXiv preprint arXiv:2405.03425 (2024) - that seems to be related.


Other comments/suggestions:
1. Line 241. Missing reference for Meta-Llama-3.1-8B
2. Line 257. The acronym PEFT is not introduced (only introduced later in the Appendix)
3. Figure 1 is unclear. It has a lot of elements that needs to be described. For example, what fire symbols mean, why $W_0$ is faded for TFB but still contributes to $z$, what different colours mean, etc.
4. Lemma D.1 is placed in a weird place, namely before the theorem, while the proof of the theorem says “We have the *following* lemma” as if it should be after this text. Moreover, as the reader reads the text and encounters the lemma first, it stands there very confusingly without a proof or reference.
5. Figure 2. In the first two plots apparently red and blue lines are inseparable. Maybe making one of the lines dashed would be better for visibility. At least text explanation is needed.


Minor:
1. Lines 344-345. “Beyesianization” -> ‘Bayesianization”

---

> ### Author Rebuttal · Authors · 2025-07-28
>
> Thank you for your constructive feedback. We are glad that you find the problem we solve `"very important"`, our paper `"can have a significant impact"`, our ideas `"novel"`, our main claims `"supported with good quality experiments"`, our proposed approach `"grounded theoretically"` `"despite simplicity"`, and our paper `"well written and easy to follow"`. Below we address your comments one by one in detail.
>
> **W1. The paper emphasises that it offers a training-free method, which makes it unique amongst existing methods. However, I am finding myself confused why it is claimed to be training-free. It still requires to find the variance. I can get that on a higher level it is not training, but from the practical point of view, isn’t it just the same as training?**
>
> We are sorry for the confusion. By "training-free", we meant to claim that the variance maximization framework proposed in TFB **does not require any gradient estimation with backpropagation**. This makes TFB more *stable*, more *plug-and-play*, and *easier to use*, especially for researchers and practitioners without a Bayesian background.
>
> From another perspective, most training-free frameworks for LLM inference introduce a few hyperparameters. By convention, the community does not consider the process of tuning these hyperparameters as part of "training." In TFB’s case, only a single (hyper)parameter, $\sigma$, is introduced, accompanied by a principled guideline for selecting its value. Hence, we describe our algorithm as "training-free".
>
> We will include this clarification in the revision as suggested.
>
> **W2. The paper is generally well put into the existing literature but after a quick search I have found this work, for example, Onal, Emre, et al. "Gaussian stochastic weight averaging for Bayesian low-rank adaptation of large language models." arXiv preprint arXiv:2405.03425 (2024) - that seems to be related.**
>
> Thanks for bringing this work to our attention.
>
> After careful comparison, we find the performance of SWAG [1] close to or worse than our two included baselines LAP and BLoB. This is why we chose LAP and BLoB as they are more representative Bayesian low-rank adaptation works.
>
> We will be sure to cite and discuss SWAG in our revision.
>
> **Q1. Line 241. Missing reference for Meta-Llama-3.1-8B**
>
> Thanks for mentioning this. We will add the reference to Llama-3.1-8B in the revision.
>
> **Q2. Line 257. The acronym PEFT is not introduced (only introduced later in the Appendix)**
>
> Thank you. We will explain the acronym of PEFT at its first appearance in the paper.
>
> **Q3. Figure 1 is unclear. It has a lot of elements that needs to be described. For example, what fire symbols mean, why $W_0$ is faded for TFB but still contributes to $z$, what different colours mean, etc.**
>
> Thank you for the helpful suggestion.
> In Figure 1, we use the fire symbol in orange to denote the trainable parameters during the Bayesianization process, the snowflake symbol in blue to denote frozen parameters, and the sample symbol in green to denote Bayesian components (i.e., the approximate posterior). The faded $W_0$ is simply for visual clarity, as it overlaps with the arrow. We will add a legend to Figure 1 to make these distinctions clearer, as suggested.
>
> **Q4. Lemma D.1 is placed in a weird place, namely before the theorem, while the proof of the theorem says “We have the following lemma” as if it should be after this text. Moreover, as the reader reads the text and encounters the lemma first, it stands there very confusingly without a proof or reference.**
>
> Thank you for the great advice. We will move Lemma D.1 into the proof of the Theorem 4.1 to make it more coherent and accessible to readers.
>
> **Q5. Figure 2. In the first two plots apparently red and blue lines are inseparable. Maybe making one of the lines dashed would be better for visibility. At least text explanation is needed.**
>
> This is a good idea. We will change the MLE curve (red) to a dashed line and add explanation in the caption to make it visually clear in the revision.
>
> **Q6. Lines 344-345. “Beyesianization” -> ‘Bayesianization”**
>
> Thank you. We will fix the typo.
>
> **References**
>
> [1] Onal, Emre, et al. "Gaussian stochastic weight averaging for Bayesian low-rank adaptation of large language models." arXiv preprint arXiv:2405.03425 (2024)

---

> > ### Comment · Reviewer_3SzA · 2025-08-04
> >
> > Thank you for your response. Thank you for clarifying about the term "training-free". I am not too familiar with this area, I can see now the difference. Thank you.
> >
> > I don't have any further questions.

---

> > > ### Author Response · Authors · 2025-08-06
> > > **Thank you.**
> > >
> > > Thank you very much for your engagement in our discussion. We’re glad that our response has fully addressed your questions, and we will be sure to include the above clarifications in our revision.

---

### Official Review · Reviewer_y21w · 2025-07-02

**Clarity:** 4
**Significance:** 3
**Originality:** 3
**Rating:** 5
**Confidence:** 5

**Summary:**

The paper proposes a training-free approach for Bayesianizing low-rank adapters (LoRA) for large language models (LLM). Bayesianizing low-rank adapters is crucial in estimating uncertainty in LLM responses, therefore of utmost practical importance. The proposed method, TFB, models the posterior over LoRA weights using low-rank isotropic Gaussian distributions, and automatically selects the maximum variance that preserves task performance using an anchor dataset. The authors prove that TFB is theoretically equivalent to generalized variational inference under mild assumptions. Empirical results across multiple LLM backbones and datasets show that TFB improves uncertainty calibration (lower ECE, NLL) while preserving accuracy and significantly reducing complexity compared to prior Bayesian LoRA methods.

**Questions:**

1. Can you clarify what you mean by "more compact" (line 107, 130)?
2. Can you move limitations to the main text?
3. How do we select an appropriate anchor dataset for a given model and downstream task?

**Ethical Concerns:**

["NO or VERY MINOR ethics concerns only"]

**Final Justification:**

The authors have provided additional experimental results, highlighting the effectiveness of their method against the baselines. The additional analyses results were also insightful.

**Limitations:**

Yes

**Quality:**

3

**Strengths And Weaknesses:**

**Strength**

1. The overall writing of the paper is very good, quite easy to follow.
2. TFB works on trained LoRA adapters without retraining or any gradient updates, making it highly practical and scalable.
3. The theoretical results provided in the paper are solid and relevant.
4. Experimental setup is pretty rigorous. TFB is shown to work well across different LLMs (LLaMA, Mistral) and LoRA variants (VeRA, PiSSA).

**Weaknesses**

1. Asymptotic inference-time complexity of TFB is not provided in the paper.
2. TFB is sensitive to tolerance level $\epsilon$. Some experiments should be conducted to understand the extent of the sensitivity.
3. TFB requires access to an anchor dataset to search for posterior variance, which might not always be available in zero-shot settings. Also, robustness of TFB across different anchor dataset should be studied.
4. Restricting to isotropic low-rank Gaussians simplifies inference but might limit modeling capacity compared to full-covariance Bayesian methods. Can we see some ablation to understand the trade-off?
5. I am not fully convinced with the performance of TFB. Although its obvious that TFB improves uncertainty estimation (NLL and ECE) over MLE, MAP and BloB-mean, it seems to be performing poorly than LAP and BloB in many instances (as reported in Table 1). Statistical significance tests should be conducted.
6. Can you compare your method with MonteCLoRA [1]? MonteCLoRA seems to be another posterior estimation method, works reliably well on LoRA. It can work in both ad-hoc and post-hoc settings, making it a strong competitor of TFB.

**References**

[1] https://arxiv.org/pdf/2411.04358

---

> ### Author Rebuttal · Authors · 2025-07-28
>
> Thank you for your constructive feedback. We are glad that you find our proposed TFB `"highly practical and scalable,"` our theoretical results `"solid and relevant,"` our experimental setup `"rigorous,"` showing TFB to `"work well"` across different LLMs and LoRA variants, and our paper's writing `"good"` and `"easy to follow."` Below we address your comments one by one in detail.
>
> **W1. Asymptotic inference-time complexity of TFB is not provided in the paper.**
>
> For each forward pass, TFB has the same computational complexity as low-rank adaptation (LoRA), which is minimal, particularly when the rank is low.
>
> The inference-time complexity of TFB matches that of other Bayesian LLMs based on weight posterior approximation (e.g., BLoB, LoRA ensemble) and introduces an additional $O(N)$ factor compared to standard non-Bayesian counterparts, where $N$ is the number of Monte Carlo samples used at test time.
>
> Regarding the "asymptotic analysis," could you please clarify whether you are referring to the asymptotic behavior as the number of samples increases during test time? If so, because TFB employs Bayesian Model Averaging (BMA) via simple Monte Carlo estimation, it is an unbiased estimator of the true expectation, and its asymptotic behavior follows directly from the Central Limit Theorem (CLT).
>
>
> **W2. TFB is sensitive to tolerance level. Some experiments should be conducted to understand the extent of the sensitivity.**
>
> Thank you for mentioning this. Actually, TFB's effectiveness is **not heavily dependent** on $\epsilon$: as demonstrated in `Table A`, we set $\epsilon$ to a consistent value corresponding to a relative performance change of 0.3\% across **all datasets.** This demonstrates TFB's insensitivity to $\epsilon$ and its universal applicability, while suggesting that further improvements are possible by tailoring $\epsilon$ to specific datasets.
>
> Furthermore, inspired by your comment, we conducted additional experiments to analyze the sensitivity of $\epsilon$. Each experiment was repeated three times with different random seeds, and the `mean (std)` is reported in the table below. The findings show that, within a reasonable range, variations in $\epsilon$ have minimal impact on the final performance in terms of ECE and NLL, thereby demonstrating the robustness of our chosen $\epsilon$ setting.
>
> **Table A. Sensitivity Analysis on the choice of $\epsilon$.**
>
> |$\epsilon$|0.001|0.002|0.003|0.004|0.005|
> |-|-|-|-|-|-|
> |**ACC($\uparrow$)**|92.18 (0.26)|92.00 (0.30)|91.76 (0.48)|91.76 (0.34)|91.76 (0.34)|
> |**ECE($\downarrow$)**|3.70 (0.50)|3.16 (0.85)|2.44 (0.50)|2.72 (0.55)|2.98 (0.66)|
> |**NLL($\downarrow$)**|0.24 (0.01)|0.23 (0.01)|0.23 (0.02)|0.23 (0.01)|0.23 (0.01)|
>
> **W3. TFB requires access to an anchor dataset to search for posterior variance, which might not always be available in zero-shot settings. Also, robustness of TFB across different anchor dataset should be studied.**
>
> Thanks for mentioning this. Actually, in practice, a natural choice for the anchor dataset is the (subset of) training set itself, which is *usually available*.
>
> Besides, in this paper, we explicitly identify and address the most significant factors of an anchor dataset:
>
> + **Training vs. Validation Set as Anchor:** We thoroughly analyze these choices in `Appendix F.4` and `Table 10`.
> + **Anchor Dataset Size:**
>   + We intentionally use an extremely small anchor dataset (500 samples) throughout all experiments, creating a challenging scenario that disadvantages TFB compared to baselines.
>   + We thoroughly study the performance brought by different anchor dataset sizes in `Appendix F.3` and `Figure 3`, showing robust performance across different sizes.
> + **Supervised vs. Unsupervised Anchor:** In `Section 3.3`, we choose the most restrictive setting where TFB operates without supervision, a scenario that existing calibration methods cannot handle.
>
> It is arguably challenging for any Bayesian method to function in a true “zero-shot” scenario, since Bayesian inference fundamentally relies on updating beliefs based on observed data. Our TFB is no exception.
>
> We will include the discussion above in the revision as suggested.
>
>
> **W4. Restricting to isotropic low-rank Gaussians simplifies inference but might limit modeling capacity compared to full-covariance Bayesian methods. Can we see some ablation to understand the trade-off?**
>
> This is a good question. Modeling the approximate posterior as a Gaussian with **full covariance** is an appealing idea. However, it expands the distribution family to better approximate the true weight posterior; consequently, it imposes prohibitive computational and memory costs on the Bayesianization process, especially for LLMs.
>
> Consider a standard scenario where the pretrained weight matrix $W_0 \in \mathbb{R}^{4096\times 4096}$ is paired with a pretrained rank-8 LoRA $\{B\in \mathbb{R}^{4096\times 8}, A\in \mathbb{R}^{8\times 4096}\}$. Modeling just one low-rank matrix with a full-covariance Gaussian requires $\frac{1}{2}(8\times 4096)^2$ parameters (accounting for symmetry), which is $32\times$ more memory consumption than the non-Bayesian method.
>
> In `Section 5.3 (TFB Beyond the Low-Rank Isotropic Gaussians)`, we investigate a **full-rank** Gaussian (FR) with a diagonal covariance matrix, an alternative to the **full-covariance** Gaussian.
> The results demonstrate that our current design of low-rank isotropic Gaussians as approximate posterior achieves both greater parameter-efficiency and superior performance, attaining the best average rank on ACC and ECE, and the second-best on NLL.
> As `Reviewer BHzf` also noted, this ablation study indicates TFB's choice of low-rank isotropic Gaussians represents `"probably the best tradeoff performance / compute wise."`
>
> **W5. I am not fully convinced with the performance of TFB. Although its obvious that TFB improves uncertainty estimation (NLL and ECE) over MLE, MAP and BloB-mean, it seems to be performing poorly than LAP and BloB in many instances (as reported in Table 1). Statistical significance tests should be conducted.**
>
> Thank you for mentioning this. In terms of the significance of TFB's performance, **the following table shows the rank ("1" being the best) of ECE and NLL from our main experiments (`Table 1`)** to provide a straightforward comparison between TFB and BLoB (the best-performing baseline). It shows TFB's overall superior performance compared to baselines.
>
> **Table B. Rank (Rk) comparison between TFB and the strongest baseline (BLoB).**
>
> |Metric|Method|WG-S|ARC-C|ARC-E|WG-M|OBQA|BoolQ|Avg. Rk.|
> |-|-|-|-|-|-|-|-|-|
> |**ECE**|**BLoB**|3|**1**|2|3|2|**2**|2.16|
> ||**TFB**|**2**|2|**1**|**2**|**1**|3|**1.83**|
> |**NLL**|**BLoB**|3|**1**|**1**|2|2|**1**|1.67|
> ||**TFB**|**1**|2|**1**|**1**|**1**|2|**1.33**|
>
> It is also important to note that the primary purpose of TFB is **NOT** to propose a new state-of-the-art method that tops the current leaderboard. Rather, our empirical evaluation is designed mainly to demonstrate that TFB achieves comparable performance to the baselines in a `"theoretically sound"`, `"empirically simple"` and efficient way. Besides, TFB achieves the current performances **despite several inherent disadvantages:**
> + it operates **without supervision**,
> + uses an **extremely small** anchor dataset,
> + requires **no gradient estimation**,
> + and employs **unified settings** across all LoRA checkpoints (MLE, MAP, and BLoB-Mean) throughout our experiments.
>
> **W6. Can you compare your method with MonteCLoRA [1]? MonteCLoRA seems to be another posterior estimation method, works reliably well on LoRA. It can work in both ad-hoc and post-hoc settings, making it a strong competitor of TFB.**
>
> Thank you for bringing this work to our attention. We will cite and discuss it in the revision.
>
> We noticed that the current implementation of MonteCLoRA does not support decoder-only transformer architecture such as LlaMA. As noted in `Section 1, line 30`, reimplementing and tuning a Bayesian LLM requires careful consideration. Although we were unable to include MonteCLoRA’s results during the rebuttal phase, we will revisit this work and incorporate a discussion of the results in the revised version.
>
>
> **Q1. Can you clarify what you mean by "more compact" (line 107, 130)?**
>
> We apologize for the confusion. By `"compact"`, we mean `"parameter-efficient"`. We will clarify this in the revision.
>
> **Q2. Can you move limitations to the main text?**
>
> This is a good suggestion. We will move the `Limitations` to the main text in the revision.
>
> **Q3. How do we select an appropriate anchor dataset for a given model and downstream task?**
>
> Please refer to our **Response to W3**.

---

> > ### Comment · Reviewer_y21w · 2025-08-03
> > **Response to Author Responses**
> >
> > Thank you for providing the additional information.
> >
> > MonteCLoRA very much works with decoder-only models and in fact evaluated with LLaMA model in their paper. As mentioned above, design-wise MonteCLoRA is one of the strongest baseline, therefore, it is crucial that you evaluate your method against this baseline.

---

> > > ### Author Response · Authors · 2025-08-03
> > >
> > > We sincerely thank you for your follow-up comments and for keeping the communication channel open. We also appreciate you pointing out that the authors of MonteCLoRA released the official implementation a few days ago. Based on their code, we have integrated MonteCLoRA into our codebase and successfully reproduced its results.
> > >
> > > To ensure a fair comparison with the other baselines and our TFB, we set MonteCLoRA's $\alpha = 16$ and rank to $r=8$. All other configurations strictly followed the original paper’s settings, and we trained MonteCLoRA for 5 epochs on the same six commonsense reasoning datasets used in our work. The table below compares our TFB, applied to both the MLE and BLoB-Mean (B-M) checkpoints, with MonteCLoRA:
> > >
> > > **Table A. Performance of MonteCLoRA and TFB (Ours) on Llama3.1-8B.**
> > >
> > > | Metric | Method | WG-S | ARC-C | ARC-E | WG-M | OBQA | BoolQ |
> > > |--------|--------|------|--------|--------|------|------|-------|
> > > | **ACC($\uparrow$)** | **MonteCLoRA** | 69.20 (0.18) | 78.38 (0.89) | 90.79 (0.62) | 74.79 (0.23) | 84.13 (0.31) | 89.17 (0.30) |
> > > | | **TFB (Ours, MLE)** | 77.44 (0.30) | 82.53 (1.00) | 91.33 (0.37) | 82.53 (0.56) | **88.53 (0.57)** | 89.75 (0.25) |
> > > | | **TFB (Ours, B-M)** | **77.81 (0.36)** | **83.33 (0.19)** | **91.76 (0.48)** | **83.81 (0.39)** | 87.80 (0.16) | **90.11 (0.28)** |
> > > | **ECE($\downarrow$)** | **MonteCLoRA** | 18.29 (0.27) | 12.22 (0.75) | 7.23 (0.71) | 15.97 (0.45) | 9.79 (0.07) | 7.09 (0.52) |
> > > | | **TFB (Ours, MLE)** | 12.98 (0.37) | 11.63 (0.68) | 5.14 (0.14) | 10.01 (0.70) | 7.20 (0.47) | 7.39 (0.26) |
> > > | | **TFB (Ours, B-M)** | **8.16 (0.48)** | **6.48 (0.36)** | **2.44 (0.50)** | **3.83 (0.43)** | **2.67 (0.18)** | **3.10 (0.59)** |
> > > | **NLL($\downarrow$)** | **MonteCLoRA** | 0.82 (0.02) | 0.71 (0.03) | 0.51 (0.04) | 0.74 (0.02) | 0.55 (0.02) | 0.36 (0.02) |
> > > | | **TFB (Ours, MLE)** | 0.68 (0.03) | 0.85 (0.02) | 0.33 (0.03) | 0.53 (0.01) | 0.46 (0.04) | 0.42 (0.00) |
> > > | | **TFB (Ours, B-M)** | **0.55 (0.01)** | **0.53 (0.04)** | **0.23 (0.02)** | **0.40 (0.01)** | **0.33 (0.02)** | **0.27 (0.01)** |
> > >
> > > As shown in `Table A`, our TFB consistently outperforms MonteCLoRA in both predictive performance and calibration, achieving higher ACC and lower ECE and NLL.
> > >
> > > We will incorporate this discussion and comparison with MonteCLoRA, an important baseline, in our revision.

---

> > > > ### Comment · Reviewer_y21w · 2025-08-04
> > > > **Response to Authors**
> > > >
> > > > Thank you for providing the additional experimental results. I have no further questions.
> > > >
> > > > I have increased my score accordingly. Good job and best of luck!

---

> > > > > ### Author Response · Authors · 2025-08-06
> > > > > **Thank you.**
> > > > >
> > > > > Thank you very much for the encouraging comments and for acknowledging our contribution. We are glad that our response has been helpful and convincing. We will make sure to include the discussion above in our revision.

---

### Official Review · Reviewer_BHzf · 2025-07-04

**Clarity:** 3
**Significance:** 2
**Originality:** 3
**Rating:** 4
**Confidence:** 3

**Summary:**

This paper proposes to “bayesianize” lora finetuning of LLMs by using as the variational posterior the lora mean and a grid searched variance term on the training data. The loss used to search the variance term is the variance that keeps the maximum functional change in loss underneath a pre-specified threshold. Experiments are done on a pretty wide variety of LLM style benchmarks.

**Questions:**

Generically how is epsilon chosen here as I imagine that’s a clearly important hyperparameter? If this has to be chosen on a reference task, how do we ensure that this reference task represents the downstream usage tasks?

What is the inference computational cost as compared to standard LLM inference?

-	It’s not super clear to me what calibration error on textual generation / a text bot should actually mean. I think I understand when and how accuracy will be better (even from just the variational mean), e.g. we get more questions right.

Why is the Laplace approximation often better than MAP / MLE in Table 1?

**Ethical Concerns:**

["NO or VERY MINOR ethics concerns only"]

**Final Justification:**

I thank the authors for responding during the rebuttal period. Overall, this is a reasonable paper demonstrating that bayesian uncertainty quantification can improve llm benchmarks.

**Quality:**

3

**Strengths And Weaknesses:**

Strengths:

-	I appreciate the simplicity of the approach, as well as the toy demonstration in Figure 2 in the appendix.

-	I like the ablation studies with other versions of the approach; indeed, it seems that the structure you chose is probably the best tradeoff performance / compute wise.

-	I also like that you can just tune the variance on a pretrained Lora model, thereby getting this for a few passes through the finetuning data.

Weaknesses:

-	Choosing the prior mean the same as the posterior mean in Thm 4.2 is definitely putting the cart before the horse. Under those conditions, pretty much anything is variational inference. The overselling of this result is one reason why i'm voting weak accept currently; this is a pretty restrictive set of assumptions (not a "relatively weak" set).

-	No comparison to something like simply turning up the generation temperature to increase output uncertainty and then using many samples in the Table 1 benchmarks.

-	I appreciate the attempt to phrase the method as Bayesian, but that’s really a strong overstatement. The “meta” prior here is “any weight distribution that doesn’t change the function too much”, which is distinctly not Bayesian, but more a post facto way of justifying some sort of uncertainty quantification.

---

> ### Author Rebuttal · Authors · 2025-07-28
>
> Thank you for your constructive feedback.
> We are glad that you `"appreciate the simplicity"` of our approach and like our ablation studies, acknowledging that the structure we chose is `"the best tradeoff performance / compute,"` and that our experiments cover `"a pretty wide variety of LLM style benchmarks."` Below we address your comments one by one in detail.
>
> **W1. Choosing the prior mean the same as the posterior mean in Thm 4.2 ... Under those conditions, pretty much anything is variational inference ... this is a pretty restrictive set of assumptions (not a "relatively weak" set).**
>
> Thank you for mentioning this. The goal of our theoretical analysis is to identify the conditions under which the proposed TFB is equivalent to variational inference, thereby providing a solid theoretical foundation for our novel yet practically simple variance maximization framework.
>
> The assumption that the posterior shares the same mean as the prior (i.e., the off-the-shelf LoRA adapter) reflects the practical design of our algorithm: in a training-free framework, estimating or modifying the parameter mean would be highly challenging. As noted in the `Remark` (lines 234–237), this assumption is also consistent with the core premise of other Bayesian methods, such as the Laplace Approximation, where the MAP point estimate is effectively used as the posterior mean.
>
> **W2. No comparison to something like simply turning up the generation temperature to increase output uncertainty and then using many samples in the Table 1 benchmarks.**
>
> This is an interesting point. Following your suggestion, we have implemented this simple baseline, which is equivalent to MC Temperature Scaling (MCTS) [1], using the LoRA weights from BLoB-mean and MAP. To ensure a fair comparison, we strictly followed the hyperparameter settings in [1] and performed 10 samples at test time (same as TFB). Each experiment was repeated three times with different random seeds, and the results are reported as `mean (std)` in the table below across six datasets.
>
> **Table A. Performance of MC Temperature Scaling (MCTS) and TFB (Ours) applied to LoRA of BLoB-mean weights on Llama3.1-8B.**
>
> |Metric|Method|WG-S|ARC-C|ARC-E|WG-M|OBQA|BoolQ|
> |-|-|-|-|-|-|-|-|
> |**ACC($\uparrow$)**|**MCTS**|77.71 (0.34)|**83.60 (0.98)**|**91.94 (0.31)**|83.28 (0.88)|**88.07 (0.62)**|90.06 (0.12)|
> ||**TFB (Ours)**|**77.81 (0.36)**|83.33 (0.19)|91.76 (0.48)|**83.81 (0.39)**|87.80 (0.16)|**90.11 (0.28)**|
> |**ECE($\downarrow$)**|**MCTS**|18.59 (0.35)|13.41 (1.19)|6.45 (0.09)|13.28 (0.74)|8.91 (0.49)|7.95 (0.05)|
> ||**TFB (Ours)**|**8.16 (0.48)**|**6.48 (0.36)**|**2.44 (0.50)**|**3.83 (0.43)**|**2.67 (0.18)**|**3.10 (0.59)**|
> |**NLL($\downarrow$)**|**MCTS**|0.97 (0.01)|0.81 (0.08)|0.37 (0.02)|0.62 (0.04)|0.46 (0.01)|0.42 (0.02)|
> ||**TFB (Ours)**|**0.55 (0.01)**|**0.53 (0.04)**|**0.23 (0.02)**|**0.40 (0.01)**|**0.33 (0.02)**|**0.27 (0.01)**|
>
> **Table B. Performance of MC Temperature Scaling (MCTS) and TFB (Ours) applied to LoRA of MAP weights on Llama3.1-8B.**
>
> |Metric|Method|WG-S|ARC-C|ARC-E|WG-M|OBQA|BoolQ|
> |-|-|-|-|-|-|-|-|
> |**ACC($\uparrow$)**|**MCTS**|**76.46 (1.33)**|82.38 (1.44)|**91.85 (0.59)**|**82.70 (0.58)**|85.73 (0.25)|**90.07 (0.18)**|
> ||**TFB (Ours)**|76.43 (0.72)|**82.80 (1.42)**|91.39 (0.37)|82.64 (0.58)|**86.00 (0.16)**|89.96 (0.18)|
> |**ECE($\downarrow$)**|**MCTS**|21.28 (0.99)|16.13 (1.60)|7.41 (0.67)|15.91 (0.62)|13.11 (0.15)|9.18 (0.21)|
> ||**TFB (Ours)**|**14.95 (0.65)**|**11.27 (2.53)**|**5.76 (0.63)**|**10.97 (1.19)**|**9.70 (0.69)**|**6.86 (0.31)**|
> |**NLL($\downarrow$)**|**MCTS**|1.30 (0.06)|1.40 (0.30)|0.59 (0.04)|0.95 (0.08)|0.98 (0.03)|0.66 (0.01)|
> ||**TFB (Ours)**|**0.77 (0.05)**|**0.80 (0.15)**|**0.38 (0.03)**|**0.57 (0.05)**|**0.61 (0.03)**|**0.40 (0.01)**|
>
> We can see that our TFB outperforms MCTS in most cases, especially in terms of ECE and NLL, which are the focus of our methods. This demonstrates TFB's effectiveness. We will include the results and discussion in the revision as suggested.
>
> **W3. I appreciate the attempt to phrase the method as Bayesian, but that’s really a strong overstatement. The “meta” prior here is “any weight distribution that doesn’t change the function too much”, which is distinctly not Bayesian, but more a post facto way of justifying some sort of uncertainty quantification.**
>
> Thank you for your constructive feedback and your insightful suggestion on connecting TFB to uncertainty quantification methods.
> Unlike typical Bayesian methods that require **training from scratch**, an approach that can be prohibitively expensive for large language models (LLMs), we focus on the problem of Bayesian **fine-tuning**. In this context, it is natural to use the pretrained weights as the mean of both the prior and the posterior so that the "weight distribution doesn't change the function too much," as you mentioned.
>
> To further demonstrate the non-triviality of TFB’s approximate posterior weight distribution $q(W)$ (as defined in Eq. 15), we compute the Signal-to-Noise Ratio (SNR) of each layer:
> $$\text{SNR}(\mathcal{N}(\mu_q, \Sigma_q)) = \frac{\|\|\mu_q\|\|^2}{\text{tr}(\Sigma_q)}.$$
> A high SNR indicates that the distribution is dominated by the Gaussian mean, effectively nullifying the need for distribution modeling.
>
>
> **Table C. Square root of the Signal-to-Noise Ratio ($\sqrt{\text{SNR}}$) for the first 12 Query and Value layers of TFB on the WG-S dataset, applied to BLoB-Mean using Llama3.1-8B pre-trained weights.**
>
> | Layer |1|2|3|4|5|6| 7| 8| 9| 10| 11| 12|
> | - | - | - | - | - | - | - | - | - | - | - | - | - |
> | **Query** | 0.2468 | 0.2700| 0.2480 | 0.2600 | 0.2330 | 0.2291 | 0.3523| 0.2857 | 0.2511 | 0.2363 | 0.2697 | 0.2314 |
> | **Value** | 0.1062 | 0.1423 | 0.1238 | 0.1245 | 0.1612 | 0.1226 | 0.1325 | 0.1569 | 0.1867 | 0.1378 | 0.1416 | 0.1472 |
>
> As shown in the tables above, the $\sqrt{\text{SNR}}$ values of the TFB layers, both the Query weights and the Value weights, are significantly less than 1. This highlights the non-trivial nature of the searched low-rank weight noises, indicating that the approximate posterior of TFB constitutes a true Bayesian neural network rather than a simple noise-injection-based uncertainty quantification method.
>
> We will add this discussion to the revision.
>
> **Q1. Generically how is epsilon chosen here as I imagine that’s a clearly important hyperparameter? If this has to be chosen on a reference task, how do we ensure that this reference task represents the downstream usage tasks?**
>
> Thank you for mentioning this. As noted in `lines 173–181,` the choice of $\epsilon$ can be determined by considering various factors in real-world production environments. Scenarios such as medical diagnosis may require extremely strict standards; in such cases, $\epsilon$ should be set to a smaller value. In our paper, we adopt a unified $\epsilon$ value across all domains, which consistently yields universal improvements.
>
> Moreover, inspired by your comment, we conducted additional experiments to analyze the sensitivity of $\epsilon$. Each experiment was repeated three times with different random seeds, and the `mean (std)` is reported in the table below. The results show that, within a reasonable range, variations in $\epsilon$ have minimal impact on the final performance in terms of ECE and NLL, thereby demonstrating the robustness of our chosen $\epsilon$ setting.
>
> **Table D. Sensitivity Analysis on the choice of $\epsilon$.**
>
> |$\epsilon$|0.001|0.002|0.003|0.004|0.005|
> |-|-|-|-|-|-|
> |**ACC($\uparrow$)**|92.18 (0.26)|92.00 (0.30)|91.76 (0.48)|91.76 (0.34)|91.76 (0.34)|
> |**ECE($\downarrow$)**|3.70 (0.50)|3.16 (0.85)|2.44 (0.50)|2.72 (0.55)|2.98 (0.66)|
> |**NLL($\downarrow$)**|0.24 (0.01)|0.23 (0.01)|0.23 (0.02)|0.23 (0.01)|0.23 (0.01)|
>
> **Q2. What is the inference computational cost as compared to standard LLM inference?**
>
> As a Bayesian method that relies on Monte Carlo sampling for expectation estimation, TFB is $N\times$ more computationally expensive than standard LLM inference, where $N$ represents the number of samples during test time. While this is a general limitation shared by all Bayesian Model Averaging methods, and improving test-time efficiency is not the primary focus of TFB, we explicitly demonstrate in `Section 5.6` that applying Bayesianization only to the last layer can partially mitigate the inference-time computational overhead.
>
> **Q3. It’s not super clear to me what calibration error on textual generation / a text bot should actually mean.**
>
> This is an excellent question. TFB serves as a simple, effective, and theoretically grounded Bayesian fine-tuning algorithm for different downstream tasks. The label spaces of these tasks (as detailed in `Appendix E.1` and `Table 6`) consist of fixed options (e.g., "A, B, C, D, E") or binary judgments ("True" and "False"). By indexing and normalizing the first token of each option, we transform the original short-form natural language generation output into a probability distribution, making the evaluation of calibration straightforward.
>
> **Q4. Why is the Laplace approximation often better than MAP / MLE in Table 1?**
>
> Thanks for mentioning this. This outcome is actually expected. LAP approximates the weight posterior via Hessian estimation, transforming the point-estimate LLM into a Bayesian LLM through a post-training process. As a result, it yields more calibrated predictions than standard estimation methods, as evidenced by the improved NLL and ECE metrics. We will include the clarification above in the revision as suggested.
>
> **References**
>
> [1] Cecere, Nicola, et al. "Monte Carlo Temperature: a robust sampling strategy for LLM's uncertainty quantification methods." arXiv preprint arXiv:2502.18389 (2025).

---

> > ### Comment · Reviewer_BHzf · 2025-08-04
> >
> > Thanks for the answers to my questions. I'm still pretty happy with my accept rating here. As a practical way to squeeze a bit of uncertainty quantification and robustness to LoRA, this seems like a reasonable enough approach.
> >
> > > this assumption is also consistent with the core premise of other **approximate** Bayesian methods, such as the Laplace Approximation, where the MAP point estimate is effectively used as the posterior mean.
> >
> > This is a good point that should be broadcast in the main text. I'm not totally sure that your approach even needs the theoretical justification then as it just adds to extra mathiness.
> >
> > >  The label spaces of these tasks ... consist of fixed options (e.g., "A, B, C, D, E") or binary judgments .... By indexing and normalizing the first token of each option, we transform the original short-form natural language generation output into a probability distribution, making the evaluation of calibration straightforwad
> >
> > Is the idea then that we're 60% for `A`, 10% for B, etc..? why can't just increasing the temperature threshold to provide sampling diversity also serve as a reasonable baseline?
> >
> > > In this context, it is natural to use the pretrained weights as the mean of both the prior and the posterior
> >
> > In Bayesian continual learning, it's certainly natural to use the pretrained weights as the mean of the prior, but generally not the posterior. At the most basic setting, I'm imagining updating a gaussian process to see a few new data points, where the GP prior mean is the current posterior on that setting, and then after updating the data, we again update the posterior. Or in the NN land the EWC (https://arxiv.org/pdf/1612.00796) line of work that at least updates the posterior...
> >
> > > signal to noise ratio
> >
> > This is a good experiment, thanks for showing it. I tend to think that adding "any" amount of epsilon white noise will produce reasonable results here as well however.

---

> ### Author Response · Authors · 2025-08-06
>
> We sincerely appreciate your encouraging words and thoughtful follow-up comments. Below, we address your questions in detail.
>
> > I'm not totally sure that your approach even needs the theoretical justification then as it just adds to extra mathiness.
>
> We thank you for your kind words.
> We agree that the main value of our work lies in its practical side, as its *simple implementation, minimal hyperparameter requirements, and broad applicability* can benefit much of the community, particularly those without a Bayesian background.
>
> On the theoretical side, our work connects the practical approach of injecting noise into network weights for calibration and uncertainty estimation with variational inference as a Bayesian technique. While the underlying concept and proof are relatively straightforward and do not rely on sophisticated tricks, we believe this formulation can offer important insights to the community.
>
> > Why can't just increasing the temperature threshold to provide sampling diversity also serve as a reasonable baseline?
>
> We are sorry for the confusion in the earlier response. Following your suggestion, we included the baseline of Monte Carlo Temperature Scaling (MCTS) [1] in `W2`, which is essentially equivalent to "turning up the generation temperature to increase output uncertainty." Specifically, as described in [1], we uniformly sampled 10 temperature values from the range $[0.1, 1]$, generated the output probability distribution for each temperature, and then averaged these distributions to obtain the final prediction. We strictly adhered to the hyperparameter settings in [1] and performed 10 sampling runs at test time (the same as TFB). Experiments were conducted using the LoRA weights from both BLoB-mean and MAP, with results shown in `Table A` and `Table B`, respectively.
>
> > In Bayesian continual learning, it's certainly natural to use the pretrained weights as the mean of the prior, but generally not the posterior...
>
> We agree that in Bayesian CL, using the previous MAP estimate as both the prior and (only) the initialization for the next stage’s approximate posterior is a common design choice. However, in our scenario, we aim to find an approximate distribution without gradient estimation, which makes updating the distribution mean risky and challenging, as the loss landscape may be sharp and sensitive to weight changes. Under this constraint, we chose to fix the mean of the posterior, i.e., using the pretrained weights as both the prior and the mean of the approximate posterior.
>
> > This is a good experiment, thanks for showing it. I tend to think that adding "any" amount of epsilon white noise will produce reasonable results here as well however.
>
> That is a good point. To show that not *any* noise will yield reasonable results, in `Section 5.3 (TFB Beyond the Low-Rank Isotropic Gaussians)` we evaluated two other straightforward designs, including one that adds full-rank white noise. The ablation study shows that TFB’s final design achieves the best overall performance while also offering advantages such as efficiency and invariance to equivalent but differently parameterized LoRA adapters.
>
> To further demonstrate that TFB does not merely reshape the output distribution in a trivial way like Temperature Scaling (TS), which preserves the token order during decoding, and to highlight the non-triviality of TFB’s approximate posterior weight distribution $q(W)$, we measured the classification "flip ratio" of TFB and BLoB relative to BLoB-mean, their common weight mean.
> The flip ratio is defined as the proportion of cases where the final classification changes; for TS, this value is 0.
> Each experiment was repeated three times with different random seeds, and the results are reported as `mean (std)` across six datasets in the table below.
>
> **Table E. Flip Ratio of BLoB and TFB (Ours) compared to BLoB-Mean on Llama3.1-8B pre-trained weights.**
>
> | Metric   | Method     | WG-S        | ARC-C       | ARC-E       | WG-M        | OBQA        | BoolQ       |
> |------------------|------------|-------------|-------------|-------------|-------------|-------------|-------------|
> |   **Flip Ratio**   | BLoB  | 6.68 (0.56)  | 6.37 (2.48)  | 4.72 (0.53)  | 6.57 (0.68)  | 5.28 (0.43)  | 3.79 (0.23)  |
> |    | TFB (Ours)   | 6.21 (0.21)  | 7.60 (1.70)  | 3.09 (0.15)  | 4.99 (0.70)  | 4.00 (0.82)  | 2.09 (0.22)  |
>
> The results show that TFB and BLoB alter the final classification outcomes to a comparable extent, verifying that TFB functions as a genuine Bayesian method rather than merely a calibration technique.
>
> **References**
>
> [1] Cecere, Nicola, et al. "Monte Carlo Temperature: a robust sampling strategy for LLM's uncertainty quantification methods." arXiv preprint arXiv:2502.18389 (2025).

---

### Note · Authors · 2025-08-11

We sincerely thank all reviewers for their time and constructive feedback. We are glad that the reviewers find

- the problem we studied as `"very important"` [*3SzA*] and potentially `"significant impact"` [*3SzA*];
- our proposed approach as `"novel"` [*3SzA, 3U1X*], `"grounded theoretically"` [*3SzA, 3U1X*], with `"theoretical grounds"` under `"mild conditions"` [*3U1X*], `"highly practical and scalable"` [*y21w*], and offering `"the best tradeoff performance / compute"` [*BHzf*];
- our experimental setup as `"rigorous"` [*y21w*], `"supported with good quality experiments"` [*3SzA*], `"covering a pretty wide variety of LLM style benchmarks"` [*BHzf*], `"covering a decent amount of dataset/benchmark models"` [*3U1X*], and demonstrating `"superiority"` [*3U1X*];
- our paper as `"well-written"` [*3SzA*], `"good"` [*y21w*], and `"easy to follow"` [*y21w, 3SzA*].

Regarding the raised concerns, we have addressed them in the rebuttal with new analyses and evidence:

- Reviewer BHzf: **Added a comparison to the new baseline Monte Carlo Temperature Scaling (MCTS)** in terms of performance, calibration, and prediction flip ratio, **demonstrating the non-triviality of TFB.** The reviewer appeared to appreciate and acknowledge these additional experiments and analyses.
- Reviewer y21w: Provided further experiments showing the **insensitivity of the noise scale $\sigma\_q$, the significance of TFB, and its superiority over the new baseline MonteCLoRA.** The reviewer acknowledged our clarifications and indicated an intention to raise the score.
- Reviewer 3SzA: **Clarified our main "training-free" claim for TFB.** The reviewer recognized this clarification and maintained a positive rating.
- Reviewer 3U1X: **Added experiments on scaling the number of TFB samples at test time, showing the current setting as a reasonable trade-off between performance and efficiency.** The reviewer seemed satisfied and raised no further questions.

We will integrate these clarifications, ablations, and results into the revision, and will release all code and supplementary materials for full reproducibility.

We also thank the AC and SAC for their guidance and oversight throughout the review process.

---

### Decision · Program_Chairs · 2025-09-17

**Decision:**

Accept (poster)

**Comment:**

This paper proposes a method to derive Bayesian-style uncertainty quantification for the low-rank adaptation of LLMs, without additional training. Under a Gaussian model of the updates, the method consists of searching for a maximal variance solution for the posterior. The paper theoretically characterizes this approach, using the framework of variational inference and performs an extensive empirical study. The reviewers appreciated the timeliness and practicality of the approach.

Some key points of discussion were about exactly how Bayesian the approach is, considering the strong assumptions that, e.g., limit the prior; the paper should be more transparent about these limitations and in what way its Bayesian formulation is an approximation. More convincing baselines were also requested by the reviewers, and some were provided during the discussion phase; these should be included. Some clarification on the fact that "training-free" does not mean no data use, but rather no gradient-based optimization, should also be part of the paper. The paper is notationally heavy, impeding understanding by those not as versed in the field; a glossary and guidance early in the paper could alleviate this.